# ARGORA: Orchestrated Argumentation for Causally Grounded LLM Reasoning and Decision Making

## Abstract

Existing multi-expert LLM systems gather diverse perspectives but combine them through simple aggregation, obscuring which arguments drove the final decision. We introduce **ARGORA**, a framework that organizes multi-expert discussions into explicit argumentation graphs showing which arguments support or attack each other. By casting these graphs as causal models, ARGORA can systematically remove individual arguments and recompute outcomes, identifying which reasoning chains were necessary and whether decisions would change under targeted modifications. We further introduce a correction mechanism that aligns internal reasoning with external judgments when they disagree. Across diverse benchmarks and an open-ended use case, ARGORA achieves competitive accuracy and demonstrates corrective behavior: when experts initially disagree, the framework resolves disputes toward correct answers more often than it introduces new errors, while providing causal diagnostics of decisive arguments.

## 1. Introduction

Large Language Models (LLMs) have improved at multi-step reasoning, with prompting methods such as Chain-of-Thought (CoT) generating steps that can boost performance on difficult tasks (Wei et al., 2022b). Yet even strong LLMs remain prone to confident but incorrect reasoning and hallucinated statements, motivating a growing line of work on verification and self-checking protocols (Huang et al., 2025; Dhuliawala et al., 2024). In high-stakes settings, plausibility is not enough: we need robust decision procedures and mechanisms that attribute decisions to specific parts of the reasoning chain.

A promising direction is to replace single-model prompting with *multi-LLM debates*: multiple model instances or agents can generate diverse viewpoints, critique one another, and aggregate conclusions, often improving reliability (Du et al., 2024; Liang et al., 2024; Long et al., 2024). However, most multi-agent protocols only expose transient natural-language exchanges and make decisions via a judge model or a simple aggregation rule. This makes it difficult to maintain a formally interpretable representation of the discussion's logical structure and dependencies, or to diagnose which reasoning chains drove the final outcome.

In parallel, recent work has begun integrating formal argumentation with LLMs by building explicit argument graphs, where statements support or attack one another and are evaluated using *quantitative semantics*, a formal method that maps statements to real-valued argument strengths (Freedman et al., 2025; Zhu et al., 2025; Görür et al., 2025). This provides a clearer structure than raw dialogue. However, existing approaches often use the argumentation layer mainly as a *scoring interface* for decision making (especially in verification-centric settings where the claim is fixed in advance). While the graph can be inspected, it does not directly answer diagnostic questions such as: *Which* reasoning chains were necessary for the conclusion? *Which* claims are most responsible for the outcome? And *how* would the decision change under targeted structural corrections?

To address these limitations, we introduce **ARGORA**, an *orchestrated argumentation-driven discussion* framework for decision making with LLM experts. ARGORA structures multi-expert exchanges into a family of quantitative bipolar argumentation frameworks (QBAFs). We further interpret QBAF evaluation as a deterministic structural causal model (SCM). Using counterfactual interventions, we quantify the causal necessity of specific reasoning chains for the resulting consensus. Finally, building on this causal view, we introduce an observation-aligned counterfactual override policy that selectively reconciles internal argumentative conclusions with an auxiliary external judgment (*LLM-as-a-judge*) when the two disagree.

[1]Anonymous Institution, Anonymous City, Anonymous Region, Anonymous Country. Correspondence to: Anonymous Author <anon.email@domain.com>.

Preliminary work. Under review by the International Conference on Machine Learning (ICML). Do not distribute.

## 2. Related Works

**Multi-LLM Discussion and Debate Frameworks** Multi-expert prompting frameworks coordinate several LLMs in a debate setting. Long et al. (2024) simulated multiple expert personas, while Li et al. (2024) utilize domain-specific experts until consensus or convergence. Du et al. (2024) introduce an iterative proposal-and-debate process, Liang et al. (2024) develop Multi-Agent Debate (MAD) with debaters and a judge, and Chen et al. (2024b) propose RECONCILE with confidence-weighted voting. While these can improve performance and reliability, they lack explicit formal representations of the debate process and do not support targeted interventions on specific arguments.

**Argumentation Frameworks with LLMs** Recent work combines formal Argumentation Frameworks (AFs) with LLMs by constructing explicit argument graphs with symbolic semantics. Freedman et al. (2025) (ArgLLM) prompt LLMs to generate arguments organized into Quantitative Bipolar Argumentation Frameworks (QBAFs) under gradual semantics for claim verification (Baroni et al., 2019; Kampik et al., 2024a). Extensions include retrieval-augmented and multi-agent settings: ArgRAG builds QBAFs over retrieved evidence for fact verification, and recent forecasting work combines QBAFs from multiple agents to aggregate evidence for explainable judgmental forecasting (Zhu et al., 2025; Görür et al., 2025). While these architectures automate argument generation and provide interpretable reasoning representations, they focus on verifying a single, pre-specified claim rather than generating and adjudicating between multiple competing candidate explanations. In contrast, ARGORA integrates argument generation, multi-expert discussion, and selection across competing candidates into a single end-to-end pipeline.

**Causal Explanations and XAI with LLMs** Explainable AI (XAI) aims to clarify model decision factors, and recent frameworks treat causal structure explicitly. Dalal et al. (2024) propose IBE-Eval, using causal reasoning principles to score and select among competing LLM-generated explanations. Chu et al. (2024) use causal analysis to mitigate confounding bias effects, enabling the extraction of unbiased steering representations for controllable generation. While IBE-Eval treats causality as an external evaluation criterion and LLMGuardrail embeds causal interventions in the generation process, both apply causal methods to single-model outputs. In contrast, ARGORA makes the causal structure intrinsic to multi-expert argumentation, where the final decision is computed from argument interactions, and counterfactual interventions enable diagnosis of how specific reasoning chains contribute to consensus.

Appendix B provides additional analyses for motivating the ARGORA framework.

## 3. Preliminaries

ARGORA builds on quantitative bipolar argumentation frameworks (QBAFs) (Baroni et al., 2018) and a causal interpretation of their evaluation. To preserve space in the main paper, we only define the minimal requirements. Appendix C collects full formal preliminaries and conventions used in this work.

### 3.1. QBAFs and modular evaluation

A QBAF is a tuple $Q = \langle \mathcal{A}, R^-, R^+, w \rangle$ with a set of arguments $\mathcal{A}$, relations $R^-, R^+ \subseteq \mathcal{A} \times \mathcal{A}$, and base scores $w : \mathcal{A} \to [0,1]$. We write $R := R^- \cup R^+$. We treat *arguments* as nodes connected by *support* and *attack* edges. In ARGORA, the graph structure that we use for QBAF is in the form of a rooted directed tree with edges oriented *child→parent*.

**Definition 3.1.** Let $Q = \langle \mathcal{A}, R^-, R^+, w \rangle$ be a QBAF and write $R := R^- \cup R^+$. Assume $(\mathcal{A}, R)$ is a finite directed tree with a unique root and that every non-root node has a unique parent. All edges are oriented from child to parent. For any non-root $a \in \mathcal{A}$, its unique parent is the node $\mathbf{Pa}(a)$ such that $(a \to \mathbf{Pa}(a)) \in R$. For any $a \in \mathcal{A}$, its children are

$$\mathbf{Ch}(a) := \{ c \in \mathcal{A} : (c \to a) \in R \}.$$

For any edge $(c \to a) \in R$, define its *polarity sign*

$$\epsilon(c, a) := \begin{cases} +1, & (c \to a) \in R^+, \\ -1, & (c \to a) \in R^-. \end{cases}$$

If $\mathbf{Ch}(a) = \{c_1, \ldots, c_n\}$ is any fixed ordering, define the polarity vector $\pi(a) := (\epsilon(c_1, a), \ldots, \epsilon(c_n, a)) \in \{-1, +1\}^n$.

A quantitative semantics assigns *final strengths* $\sigma : \mathcal{A} \to [0,1]$. We use *modular semantics* (aggregation–influence) characterized by a single node-local update rule.

**Definition 3.2** (Modular semantics)**.** A *modular semantics* is specified by a pair $(\alpha, \iota)$, consisting of (i) aggregation maps $\alpha_\pi : [0,1]^n \to \mathbb{R}$ indexed by polarity vectors $\pi \in \{-1, +1\}^n$ and (ii) influence maps $\iota_b : \mathbb{R} \to [0,1]$ indexed by base scores (strengths) $b \in [0,1]$, such that the strength function $\sigma$ satisfies, for every $a \in \mathcal{A}$ with $\mathbf{Ch}(a) = \{c_1, \ldots, c_n\}$ and polarity vector $\pi(a)$,

$$\sigma(a) = \iota_{w(a)}\big(\alpha_{\pi(a)}\big(\sigma(c_1), \ldots, \sigma(c_n)\big)\big). \quad (1)$$

For leaves ($n = 0$), we assume normalization so that $\sigma(a) = w(a)$.

### 3.2. Counterfactuals via SCM casting

Our counterfactual explanations ask how the final outcome changes under *edge-local edits* to the argumentation tree

(e.g., removing a single support/attack edge). We formalize these edits by casting the node-local update equations (Eq. (1)) as a deterministic structural causal model (SCM), so that an edge deletion corresponds to a counterfactual intervention. Formal SCM definitions and conventions are given in Appendix C.3.

# 4. ARGORA

At a high level, given a user-specified topic $t$, ARGORA orchestrates a structured discussion among multiple *expert* LLM instances under the coordination of a central *Orchestrator*. The resulting expert interactions are compiled into quantitative bipolar argumentation frameworks (QBAFs), which are evaluated under a chosen quantitative semantics to produce an initial argumentative consensus. To support explainability and causal diagnosis beyond aggregate scores, the induced QBAFs are cast as deterministic structural causal models (SCMs), enabling counterfactual explanation via edge-local interventions. In parallel, ARGORA computes an external observational consensus using the Orchestrator in an *LLM-as-a-judge* role, which serves as an external override signal when the argumentative decision is structurally unstable or misaligned. The final output consists not only of a consensus decision, but also causal explanations and diagnostic artifacts characterizing the robustness of that decision. A detailed illustrative overview of the full pipeline is provided in Appendix A.

## 4.1. Expert Discussion and QBAF Generation

For the discussion phase, each expert outputs a main argument $m$ (its direct answer on topic $t$) and supplementary arguments that support or attack within-round reasoning. For each main argument $m$, ARGORA constructs a bounded-depth rooted-tree QBAF with root $m$. During this phase, the Orchestrator assigns each argument node $a$ a base score $w(a) \in [0, 1]$ (Def. 3.2). Once QBAFs have been constructed for all main arguments, ARGORA evaluates all QBAFs under the chosen quantitative semantics and performs counterfactual analysis on the induced SCMs. Full construction details are deferred to Appendix D.

## 4.2. Redundancy Control via Contextual Orthogonality

Multi-expert discussions frequently produce paraphrases and near-duplicate statements. If inserted naively, these redundant argument nodes can (i) rapidly inflate the effective context length for both the experts and the orchestrator, and (ii) reduce the efficacy of the argumentation framework—often causing strength saturation under modular semantics and diminishing the informativeness of counterfactual explanation queries. To control this redundancy, we introduce a *contextual orthogonality* criterion during graph expansion: a candidate argument is retained only if its statement text is sufficiently distinct from the discussion context already represented in the graph.

Concretely, at depth level $\ell$, we consider a candidate set $\mathcal{A}_{\text{cand}}^{(\ell)}$ and compute embedding similarity between statement texts using cosine similarity $\text{sim}(u, v) = \cos(\phi(u), \phi(v))$ with a sentence encoder $\phi$. We apply a two-stage filter with threshold $\rho_{\text{sim}}$. Stage 1 (parent-level orthogonality) prunes candidates whose maximum similarity to existing nodes at the parent level $\mathcal{A}^{(\ell-1)}$ exceeds $\rho_{\text{sim}}$, while Stage 2 (sibling orthogonality) greedily selects from the remaining candidates to avoid retaining multiple similar arguments attached to the same parent. If Stage 1 rejects all candidates, we retain a single fallback candidate with the smallest parent-level similarity to avoid an empty set of nodes. We provide additional details in Appendix D.4.

## 4.3. Causally Grounded Explainability through Counterfactual Interventions

Argumentative multi-LLM systems improve accuracy on a range of tasks (Kenton et al., 2024; Zhang et al., 2024; Chen et al., 2024a), but leave open the question of *why* a particular chain of reasons prevailed. Outcomes often rely on judging procedures such as majority voting which limits interpretability, and most LLM explainability work focuses on token- or feature-level attributions rather than structured causal reasoning over argument graphs (Zhao et al., 2024). To address this gap, ARGORA equips its QBAF layer with a causally grounded explanation mechanism based on SCMs and edge-local interventions.

**Proposition 4.1.** *Let $Q_m = \langle \mathcal{A}_m, R_m^-, R_m^+, w_m \rangle$ be a QBAF (for a main argument $m$) and fix a modular semantics $(\alpha, \iota)$ (Def. 3.2). Let $\sigma : \mathcal{A}_m \to [0, 1]$ be the (strength) function satisfying the corresponding modular update equations on $Q_m$, and define the evaluation system $\mathcal{E}_m := \langle Q_m, \sigma \rangle$, with evaluated root strength $\sigma(m)$.*

*Assume the structural conditions in App. C.4. Then $\sigma$ is well defined and unique, and $\mathcal{E}_m$ induces a deterministic SCM $\mathcal{M}_m = (\mathbf{V}, \mathbf{U}, \mathcal{F})$ with $\mathbf{V} = \{ v_a \mid a \in \mathcal{A}_m \}$, $\mathbf{U} = \{ u_a \mid a \in \mathcal{A}_m \}$, and exogenous instantiations $u_a := w_m(a)$ for all $a \in \mathcal{A}_m$. For each $a \in \mathcal{A}_m$, let $\mathbf{Ch}(a) = \{c_1, \ldots, c_n\}$ and let $\pi(a) \in \{-1, +1\}^n$ be the corresponding polarity vector (Def. 3.2). The structural assignments are, for every $v_a \in \mathbf{V}$,*

$$v_a = f_{v_a}(u_a, \mathbf{Pa}(v_a)) := \iota_{u_a}\big(\alpha_{\pi(a)}(v_{c_1}, \ldots, v_{c_n})\big),$$

*where $\mathbf{Pa}(v_a) = \{ v_c \in \mathbf{V} \mid c \in \mathbf{Ch}(a) \}$. The proof of this is given in App. C.5.*

In the induced SCM, we use a single primitive: an *edge-local* intervention that deletes the unique outgoing edge from a non-root node $x$ to its parent. This blocks the influence channel from the subtree rooted at $x$ to the root,

while keeping node-local mechanisms and base scores unchanged, and matches the edge/path intervention perspectives in causal inference (Avin et al., 2005; Pearl, 2009; Richardson & Robins, 2013; Shpitser & Tchetgen, 2016).

**Definition 4.2.** Let $Q_m = \langle \mathcal{A}_m, R_m^-, R_m^+, w_m \rangle$ be a rooted directed tree (child→parent) with root $m$, and write $R_m := R_m^- \cup R_m^+$. Fix a modular semantics $(\alpha, \iota)$ (Def. 3.2) and let $\sigma$ be the unique strength function on $Q_m$ (Prop. 4.1). For any non-root $x \in \mathcal{A}_m \setminus \{m\}$ with parent $p := \mathbf{Pa}(x)$, the *edge-local intervention at $x$* (denoted $\ominus x$) deletes the unique outgoing influence edge $(x \to p)$ and yields the intervened QBAF

$$Q_m^{\ominus x} := \langle \mathcal{A}_m, R_m^- \setminus \{(x \to p)\}, R_m^+ \setminus \{(x \to p)\}, w_m \rangle.$$

Let $\sigma^{\ominus x}$ be the unique strength function on $Q_m^{\ominus x}$ under the *same* $(\alpha, \iota)$. The counterfactual root strength is $\sigma^{\ominus x}(m)$.

Edge-local interventions provide *mechanistic* explainability in the sense that they formally trace how argumentative influences determine the decision *within* the explicit decision mechanism specified by the QBAF-induced SCM (Prop. 4.1). The resulting explanation is therefore a *verifiable causal attribution* rather than a post-hoc textual rationale. Human-centered evaluation (e.g., whether such explanations improve user understanding or trust) is orthogonal and left to future work.

**Definition 4.3** (Root-level edge-local impact). Let $\sigma(m)$ be the factual root strength for $Q_m$ and let $\sigma^{\ominus x}(m)$ be the counterfactual root strength for $Q_m^{\ominus x}$ (Def. 4.2). Define

$$\Delta_{\text{edge}}(x; m) := \sigma(m) - \sigma^{\ominus x}(m).$$

We interpret $\Delta_{\text{edge}}(x; m) > 0$ as net support of the root, $\Delta_{\text{edge}}(x; m) < 0$ as net attack, and $|\Delta_{\text{edge}}(x; m)|$ as influence magnitude.

**Definition 4.4** (Counterfactual explanation queries). Let $\mathbf{Ch}(m)$ be the direct children of $m$ (Def. 3.1), and let $\text{leaf}(Q_m) := \{ a \in \mathcal{A}_m : \mathbf{Ch}(a) = \emptyset \}$ be the leaves of $Q_m$. For each $a \in \text{leaf}(Q_m)$, let $P(a)$ be the unique directed path from $a$ to $m$. Define:

$$a_{\text{dir}}^\star \in \arg\max_{a \in \mathbf{Ch}(m)} |\Delta_{\text{edge}}(a; m)|,$$

$$a_{\text{leaf}}^\star \in \arg\max_{a \in \text{leaf}(Q_m)} |\Delta_{\text{edge}}(a; m)|,$$

$$a_{\text{all}}^\star \in \arg\max_{a \in \mathcal{A}_m \setminus \{m\}} |\Delta_{\text{edge}}(a; m)|.$$

Let $P^\star := P(a_{\text{leaf}}^\star)$. We call $a_{\text{dir}}^\star$ the most influential direct child (of the main argument), $P^\star$ the most decisive argument chain, and $a_{\text{all}}^\star$ the most influential argument node.

Taken together, these queries explain the decision made by $Q_m$ by identifying edge-local impacts at three perspectives (children, leaf-to-root chains, and nodes). Positive $\Delta_{\text{edge}}$ indicates net support and negative indicates net attack (Def. 4.3).

### 4.4. Final Consensus Generation and Observation-Aligned Counterfactual Override

The previous section provides causally-grounded explanations of *why* ARGORA prefers a particular main argument: we can identify which edges are necessary and how the decision would change under targeted perturbations. However, these explanations are entirely *internal* to the QBAF-induced SCM. They answer "how did the system arrive at this decision?" but not "is this decision aligned with external reality?" When the internal winner $m^\star$ is potentially misaligned with what a competent external judge would decide, our counterfactual diagnostics can identify arguments that drove the error in retrospect, but the framework has no mechanism to detect that the internal consensus may be brittle.

To address this challenge, we construct a structurally independent observational consensus by re-using the orchestrator as an *LLM-as-a-judge* with *no access to QBAF scores or graph structure*. Given only the main arguments and a compressed discussion transcript, it outputs raw confidence scores $\widehat{\sigma}(m)$ for each main argument $m$, derived from textual evidence aggregation rather than quantitative semantics.

**Definition 4.5** (Baseline and observational consensus). Let $\mathcal{A}^{\text{main}}$ be the set of main arguments and assume $\sigma(m) > 0$ and $\widehat{\sigma}(m) > 0$ for all $m \in \mathcal{A}^{\text{main}}$. Define the normalized consensus distributions

$$p_{\text{QBAF}}(m) := \frac{\sigma(m)}{\sum_{m' \in \mathcal{A}^{\text{main}}} \sigma(m')},$$

$$p_{\text{obs}}(m) := \frac{\widehat{\sigma}(m)}{\sum_{m' \in \mathcal{A}^{\text{main}}} \widehat{\sigma}(m')}.$$

Let $m^\star \in \arg\max_{m \in \mathcal{A}^{\text{main}}} p_{\text{QBAF}}(m)$ be the *baseline winner* and $m^{\text{obs}} \in \arg\max_{m \in \mathcal{A}^{\text{main}}} p_{\text{obs}}(m)$ be the *observational winner*.

When the internal and external assessments disagree, we can use counterfactual interventions to reconcile them. However, such reconciliation involves a fundamental trade-off: aligning with the external signal requires perturbing the internal SCM, potentially undermining the causal structure that makes the system interpretable. We formalize this through a cost-regularized objective with a *winner-confidence gate* that inhibits overrides when the baseline winner is already more confident than the observational winner—a safety mechanism preventing over-correction when disagreement is attributable to external noise rather than internal error.

**Definition 4.6** (Observation-aligned counterfactual override). Let $\mathcal{I}_{\text{cand}}$ be a finite candidate family of intervention configurations containing $\emptyset$, where each configuration $\mathcal{I} = (I_m)_{m \in \mathcal{A}^{\text{main}}}$ specifies edge-local interventions on a set $I_m \subseteq \mathcal{A}_m \setminus \{m\}$ for each main argument $m$ (Def. 4.2). For any $\mathcal{I}$, let $p_{\text{QBAF}}^{\mathcal{I}}$ denote the QBAF consensus distri-

bution after applying $\mathcal{I}$ and re-evaluating under the same semantics, and let $m^{\mathcal{I}} \in \arg\max_{m \in \mathcal{A}^{\mathrm{main}}} p_{\mathrm{QBAF}}^{\mathcal{I}}(m)$.

Fix $\lambda \geq 0$ and define the alignment–vs.–perturbation objective

$$J(\mathcal{I}) := \mathrm{JS}\big(p_{\mathrm{QBAF}}^{\mathcal{I}} \,\|\, p_{\mathrm{obs}}\big) + \lambda\, C(\mathcal{I}),$$

where $\mathrm{JS}(\cdot\|\cdot)$ is the Jensen–Shannon divergence and

$$C(\mathcal{I}) := \frac{\sum_{m \in \mathcal{A}^{\mathrm{main}}} \sum_{a \in \mathcal{A}_m} \big|\sigma^{Q_m}(a) - \sigma^{Q_m^{\mathcal{I}}}(a)\big|}{\sum_{m \in \mathcal{A}^{\mathrm{main}}} |\mathcal{A}_m|}$$

is the normalized internal state perturbation cost.

Fix a threshold $\tau \geq 0$. Define the eligible set

$$\mathcal{I}_{\mathrm{align}} := \{\emptyset\} \cup \Big\{ \mathcal{I} \in \mathcal{I}_{\mathrm{cand}} :$$

$$\underbrace{p_{\mathrm{obs}}(m^{\mathrm{obs}}) - p_{\mathrm{QBAF}}(m^{\star}) \geq \tau}_{\text{winner-confidence gate}} \wedge\, m^{\mathcal{I}} = m^{\mathrm{obs}} \Big\}.$$

Given $\hat{\mathcal{I}} \in \arg\min_{\mathcal{I} \in \mathcal{I}_{\mathrm{align}}} J(\mathcal{I})$, set $m^{\mathrm{final}} := m^{\star}$ unless $\hat{\mathcal{I}} \neq \emptyset$ and $J(\hat{\mathcal{I}}) < J(\emptyset)$, in which case set $m^{\mathrm{final}} := m^{\hat{\mathcal{I}}}$.

In our implementation, we restrict $\mathcal{I}_{\mathrm{cand}}$ to single-edge interventions ($|\mathcal{I}| = 1$) for computational efficiency. This mechanism demonstrates how external feedback can be systematically integrated into a causal framework: ARGORA overrides the baseline winner only when minimal structural perturbation both matches the observational signal and improves the alignment trade-off, with the confidence gate preventing harmful corrections when internal consensus is already reliable.

Upon completion, ARGORA automatically generates a detailed technical consensus report providing a comprehensive, verifiable record of all computed quantities—base scores, QBAF structures, evaluated strengths, counterfactual results, and override decisions (Appendix F).

### 4.5. Implementation Details

ARGORA is fully implemented in Python. The core implementation comprises of approximately 9.7K lines of code. All source code, evaluation scripts, and processed datasets used in this study are publicly available at: https://anonymous.4open.science/r/ARGORA/.

## 5. Evaluation

We evaluate ARGORA on three key dimensions for a holistic evaluation of the reasoning process and design choices: (1) **Reasoning Efficacy**, measuring standard accuracy against baselines; (2) **Argumentative Utility**, quantifying the system's ability to correct prior biases through the discussion process; and (3) **Structural Stability**, assessing the robustness of decisions through the induced

SCM. We also evaluate the observation-alignment override mechanism as a selective, confidence-gated correction step, reporting both its effectiveness and risks of over-correction. In addition, while ARGORA is designed to work with any decision-making tasks, the main evaluation is performed with benchmarks containing discrete ground-truth labels for a quantitative measurement of decision quality and causal diagnostics. We therefore complement our evaluation through a domain-specific use case for a qualitative assessment of explainability (Section 6).

### 5.1. Evaluation Methodology

To quantify the contribution of the argumentation layer, we evaluate how QBAF aggregation changes the identity and correctness of the winning main argument, comparing the *prior winner* under initial scores to the *final winner* after QBAF evaluation. We additionally report paired significance tests for these within-instance transitions in Appendix G.

To quantify **Argumentative Utility**, we track how the argumentation process alters initial beliefs. Let $\{(t_i, y_i)\}_{i=1}^N$ be evaluation task instances $t_i$ with ground-truth label $y_i \in \mathcal{Y}$. Let $L : \mathcal{A}^{\mathrm{main}} \to \mathcal{Y}$ be a mapping function that extracts the predicted label from a main argument. Let $\mathcal{A}^{\mathrm{main}(i)}$ denote the set of main arguments for instance $i$. For each instance $i$, let $m_i^{(0)} \in \arg\max_m w(m)$ denote the *prior winner* (highest initial base score), and $m_i^* \in \arg\max_m \sigma(m)$ denote the *final winner* after QBAF evaluation (Def. 4.5).

We first define the four *transition counts* that summarize how correctness changes from the label of the prior winner to the final winner:

$$p_i := \mathbb{I}\Big[L(m_i^{(0)}) = y_i\Big], \qquad q_i := \mathbb{I}[L(m_i^*) = y_i].$$

$$\begin{bmatrix} n_{+\to+} & n_{+\to-} \\ n_{-\to+} & n_{-\to-} \end{bmatrix} = \sum_{i=1}^N \begin{bmatrix} p_i q_i & p_i(1-q_i) \\ (1-p_i)q_i & (1-p_i)(1-q_i) \end{bmatrix}$$

Here, $\mathbb{I}[\cdot]$ denotes the Boolean indicator function:

$$\mathbb{I}[P] := \begin{cases} 1, & \text{if } P \text{ is true,} \\ 0, & \text{otherwise.} \end{cases}$$

These counts form an exhaustive partition of the dataset, hence $n_{+\to+} + n_{-\to+} + n_{+\to-} + n_{-\to-} = N$.

The *Argumentative Utility Matrix* rates are the normalized transition counts:

$$\underbrace{\mathrm{TRR} = \frac{n_{+\to+}}{N}}_{\text{Truth Retention Rate}}, \qquad \underbrace{\mathrm{PRR} = \frac{n_{-\to+}}{N}}_{\text{Positive Reversal Rate}},$$

$$\underbrace{\mathrm{NRR} = \frac{n_{+\to-}}{N}}_{\text{Negative Reversal Rate}}, \qquad \underbrace{\mathrm{EPR} = \frac{n_{-\to-}}{N}}_{\text{Error Persistence Rate}}.$$

**Net Reversal Efficiency.** With argumentative utility, the goal is to assess whether ARGORA can correct more errors than it introduces. However, measuring this effect requires careful consideration of when the argumentation mechanism is operative. When all main arguments unanimously predict the same label—i.e., $L(m) = y$ for all $m \in \mathcal{A}^{\mathrm{main}}$—then $L(m^{(0)}) = L(m^*)$ necessarily holds, and no reversal (positive or negative) is possible regardless of the argumentation dynamics. Such instances are informationally inert with respect to reversals.

To isolate the contribution of the argumentation layer, we condition on the *disagreement set*: the subset of instances where at least two main arguments predict different labels. Formally, let

$$\mathcal{N}_{\mathrm{disagree}} := \big\{ i \in \{1, \ldots, N\} : \exists m, m' \in \mathcal{A}^{\mathrm{main}(i)} \\ \text{s.t. } L(m) \neq L(m') \big\}$$

denote this set. The *Net Reversal Efficiency* (NRE) is our primary indicator of positive **Argumentative Utility**:

$$\mathrm{NRE} = \frac{n_{-\to+} - n_{+\to-}}{|\mathcal{N}_{\mathrm{disagree}}|}.$$

A positive NRE indicates that, among instances where experts genuinely disagree, the argumentation layer resolves disputes toward correct answers more often than it diverts from them. We also report the statistical significance tests for NRE in Appendix G.

To evaluate **Structural Stability**, we introduce the **Correctness Margin**. Recall that $L : \mathcal{A}^{\mathrm{main}} \to \mathcal{Y}$ extracts the predicted label from a main argument, and note that multiple main arguments may map to the same label (i.e., $L(m_1) = L(m_2)$ is possible for $m_1 \neq m_2$). For a given instance $i$ with ground-truth label $y_i \in \mathcal{Y}$, define the strongest correct-label and strongest incorrect-label argument strengths as:

$$\sigma_{\mathrm{correct}}^{(i)} := \max\{\sigma(m) : m \in \mathcal{A}^{\mathrm{main}(i)}, \ L(m) = y_i\},$$

$$\sigma_{\mathrm{wrong}}^{(i)} := \max\{\sigma(m) : m \in \mathcal{A}^{\mathrm{main}(i)}, \ L(m) \neq y_i\}.$$

We define $\mathcal{N}_{\mathrm{valid}} := \big\{i \in \{1, \ldots, N\} : \exists m, m' \in \mathcal{A}^{\mathrm{main}(i)} \text{ s.t. } L(m) = y_i \wedge L(m') \neq y_i\big\}$ to be the set in which at least one main argument predicts the correct label and at least one predicts an incorrect label. The **Correctness Margin** is:

$$\Delta_{\mathrm{correct}} := \frac{1}{|\mathcal{N}_{\mathrm{valid}}|} \sum_{i \in \mathcal{N}_{\mathrm{valid}}} \big(\sigma_{\mathrm{correct}}^{(i)} - \sigma_{\mathrm{wrong}}^{(i)}\big).$$

A positive $\Delta_{\mathrm{correct}}$ indicates that ARGORA's argumentation structure systematically assigns higher strength to correct-label main arguments than to incorrect-label ones.

**5.2. Evaluation Benchmarks**

Our goal is to demonstrate that ARGORA is broadly compatible with diverse task formats and domains, rather than being specialized to a single benchmark family. Accordingly, we evaluate on a suite spanning: (i) broad multi-domain knowledge and reasoning, (ii) domain-specific professional QA (medicine and science), (iii) robustness to common misconceptions and imitative falsehoods, and (iv) long-context, multi-step narrative reasoning. To this end, we use the following set of benchmark datasets: **MMLU-Pro** (Wang et al., 2024b): harder MMLU with *ten* choices and reasoning-focused curation; **MedQA** (Jin et al., 2021): clinically grounded medical MCQA; **TruthfulQA** (Lin et al., 2022): truthfulness under misconception-inducing prompts; **GPQA Diamond** (Rein et al., 2024): graduate-level science QA subset with strong expert agreement; **MuSR** (Sprague et al., 2024): long-context multi-step narrative reasoning (murder mysteries, object placement, team assignment). In all of our evaluations, we randomly shuffle the datasets on a fixed random seed for reproducibility.

**5.3. Baselines and Evaluation Settings**

ARGORA integrates (i) multi-expert discussion, (ii) deterministic aggregation via QBAF modular semantics, and (iii) a counterfactual intervention-driven override process. Closest related systems (Liang et al., 2024; Freedman et al., 2025) differ substantially in their intermediate representations and outputs (e.g., free-form debate transcripts or generative synthesis), which makes a strictly fair end-to-end comparison difficult without re-implementing and re-tuning each method under equal settings.

We therefore focus on controlled baselines that isolate the effect of ARGORA's orchestration and aggregation from improvements attributable to a stronger backbone model. Our primary baseline is a *single-model* instance of the same base LLM used for ARGORA's experts, prompted to answer each question directly ("*select the best answer.*"). We additionally evaluate an identical CoT-like task prompt used in ARGORA for the single-model baseline.

We also include a compute-matched *majority-vote ensemble* baseline akin to self-consistency (Wang et al., 2022). For each instance, we run the single-model baseline for $n$ independent trials (matching the number of experts in ARGORA) and select the majority answer; ties are broken by an *LLM-as-a-judge*. Table 3 summarizes the main design differences between ARGORA and related multi-expert reasoning frameworks.

Finally, the hyperparameters used for our evaluation are given in Table 6 in the Appendix; in all experiments, temperature is set to 0.

*Table 1.* Evaluation of ARGORA across datasets (all variants use the `GPT-4o-mini` model). Baseline majority vote (MV$_3$) uses 3 independent samples. ARGORA uses 3 experts; we report both DF-QuAD and the best-performing semantics ("Best") under the same conditions for reference. If the best-performing semantics coincides with DF-QuAD, we replace the entire "Best" block with the marker ≡ to indicate identical semantics (and thus identical entries) to the DF-QuAD block. Metrics: Acc = accuracy; NRE (Net Reversal Efficiency), marked as + or − depending on polarity; $\Delta_{\text{correct}}$ = correctness margin. We mark the best-performing and second best-performing variants by **boldface** and underlining among the baselines and ARGORA (post-override) under DF-QuAD (ties may occur). Finally, we denote the post-override accuracy change relative to its pre-override value with ↑ / ↓; "−" indicates no change.

| Dataset | N | Baselines | | | | ARGORA (*pre-override*) | | | | | | ARGORA (*post-override*) | |
| --- | --- | --- | --- | --- | --- | --- | --- | --- | --- | --- | --- | --- | --- |
| | | Direct 1× | Direct MV$_3$ | ARGORA-like (modified CoT) 1× | ARGORA-like (modified CoT) MV$_3$ | DF-QuAD | | | Best | | | DF-QuAD | Best |
| | | Acc | Acc | Acc | Acc | Acc | NRE | $\Delta_{\text{correct}}$ | Acc | NRE | $\Delta_{\text{correct}}$ | Acc | Acc |
| MMLU-Pro | 500 | 0.604 | 0.604 | 0.602 | 0.618 | 0.638 | + | +0.041 | 0.640 | + | +0.039 (REB) | **0.636** (↓) | 0.642 (↑) |
| MedQA | 500 | 0.796 | **0.822** | 0.812 | 0.812 | 0.812 | + | +0.186 | | ≡ | | 0.816 (↑) | ≡ |
| TruthfulQA | 500 | 0.792 | 0.758 | 0.826 | 0.824 | 0.882 | + | +0.269 | 0.886 | + | +0.172 (SDQ) | **0.882** (−) | 0.886 (−) |
| GPQA Diamond | 198 | 0.394 | 0.399 | 0.429 | **0.459** | 0.450 | + | +0.045 | 0.480 | + | +0.043 (QE) | 0.455 (↑) | 0.480 (−) |
| MuSR (Murder Mystery) | 250 | 0.612 | 0.604 | 0.604 | 0.632 | 0.664 | + | +0.021 | | ≡ | | **0.664** (−) | ≡ |
| MuSR (Object Placement) | 256 | **0.523** | 0.511 | 0.503 | 0.511 | 0.523 | − | +0.003 | 0.532 | 0 | +0.008 (EBT) | **0.523** (−) | 0.551 (↑) |
| MuSR (Team Allocation) | 250 | **0.536** | 0.516 | 0.548 | 0.532 | 0.564 | + | −0.047 | | ≡ | | **0.564** (−) | ≡ |

## 5.4. Results and Discussion

Table 1 summarizes the main results across seven benchmarks using `GPT-4o-mini` throughout, with 3 experts for ARGORA and a compute-matched majority-vote baseline (MV$_3$). We use DF-QuAD as our default semantics and report per-benchmark best semantics only for reference; since this selection uses ground-truth labels, it is an oracle upper bound rather than a deployable strategy. Overall, ARGORA improves accuracy over both direct prompting and majority vote baselines on most datasets, with substantial gains on general reasoning and truthfulness benchmarks. On the three MuSR subsets, ARGORA consistently matches or exceeds the baselines, with the largest post-override boost (under the "Best" semantics) appearing on the Object Placement subtask. MedQA is the main exception where the majority-vote baseline is already strong, and ARGORA remains competitive while still providing additional mechanistic diagnostics beyond accuracy.

ARGORA's argumentation layer is often *corrective* when experts genuinely disagree: NRE is positive on almost all benchmarks. We also observe mostly positive correctness margins $\Delta_{\text{correct}}$ (e.g., MedQA +0.186, TruthfulQA +0.269 pre-override), indicating that QBAF strengths tend to separate correct-label main arguments from incorrect-label ones.

Two notable caveats are MuSR's Object Placement and Team Allocation subtasks; However, additional evaluations conducted in Appendix G suggest that these results are largely due to base-model limitations rather than the argumentation layer. We also report further analyses (significance tests, override statistics, additional models, ablations) to complement our evaluations in Appendix G and Appendix H.

## 6. Use Case Evaluation

To complement our counterfactual diagnostics with a qualitative assessment of explainability, we study a cybersecurity scenario in which credibility judgments must be explicitly localized within a narrative. Models are tasked with distinguishing a *fabricated* ransomware incident report from a genuine breach report. We use this case study to qualitatively examine whether the inner discussion process of ARGORA highlights its final decision and produces an accessible, interpretable consensus.

### 6.1. Setup

We evaluate three systems: (1) ARGORA, composed of 3 experts, each using Qwen/Qwen3-14B (Yang et al., 2025) instruction-tuned models from the official HuggingFace repository, served via vLLM in an OpenAI-compatible API mode; (2) GPT-OSS 120B (Agarwal et al., 2025), an open-weight model from the official HuggingFace repository (`openai/gpt-oss-120b`), served via vLLM in an OpenAI-compatible API mode with the default checkpoint; and (3) Gemini 2.5 Pro (Comanici et al., 2025), a proprietary model accessed via the official Google Generative Language API.

All systems are provided with the same synthetic incident report describing a fictitious company in the industrial sector. The incident was constructed using a generation pipeline designed to produce plausible threat-intelligence narratives while embedding controlled inconsistencies, including OPSEC violations, timeline irregularities, and missing forensic signals. The absence of indicators or leaked artifacts from the real world has been verified by human reviewers.

The task is framed as a binary classification problem (*real*

*Table 2.* Reasoning comparison across models. All criteria are derived from a single synthetic incident shared by all systems.

| Criterion | ARGORA | OSS 120B | Gemini 2.5 |
|---|---|---|---|
| OPSEC anomaly detected | ✓ | ✗ | ✗ |
| Timeline inconsistency | ✓ | ✗ | ∼ |
| Missing forensic artifacts | ✓ | ✗ | ✗ |
| TTP mismatch flagged | ✓ | ✗ | ✗ |
| Final verdict | **Fabricated** | Real | Real |

vs. *fabricated*), with successful performance additionally requiring anomaly criterion-driven explanations. All criteria in Table 2 correspond to this single synthetic scenario and were evaluated on identical inputs.

### 6.2. Model Outputs

**ARGORA (Qwen-3 14B, 3 experts).** The system correctly classified the incident as *fabricated*. During the discussion, experts uncovered multiple inconsistencies that were individually subtle but collectively indicative of a lack of forensic coherence. Rather than relying on any single anomaly, agents iteratively challenged and refined the assumptions of each other, exposing deeper structural issues in the narrative. In particular, the consensus decision was supported by explicit criterion-level explanations as follows:

- **OPSEC anomaly.** The system identified the ransomware portal domain "*horizonhelpdesk45n1lk.onion*" as embedding the victim's name, a violation of BlackSuit's OPSEC norms that strongly suggests deliberate mimicry rather than authentic infrastructure.
- **Timeline inconsistency.** The analysis flagged the attacker's public claim occurring on the same date as system restoration as implausible for a double-extortion campaign, where disclosure is typically delayed to preserve leverage.
- **Missing forensic artifacts.** The absence of verifiable encryption artifacts or exfiltration evidence (e.g., hashes or leak-site material) was treated as incompatible with a genuine ransomware incident.
- **TTP mismatch.** Deviations such as the use of a custom Python-based exfiltration mechanism conflicted with tooling commonly associated with BlackSuit, reinforcing the hypothesis of misattribution or a staged false flag.

Through this process, the experts converged on the interpretation that the report closely mimicked the stylistic conventions of real industrial ransomware disclosures, but lacking the tightly interconnected technical signals characteristic of genuine breaches.

For comparison, we also evaluated a single-agent baseline using the same Qwen-3 14B instruction-tuned model. Under identical inputs and prompts, the single-agent configuration failed to surface any of the four embedded forensic inconsistencies and consequently classified the incident as *real*. This highlights that the gains of ARGORA arise from coordinated multi-expert reasoning rather than from the underlying model alone.

**GPT-OSS 120B and Gemini 2.5 Pro.** Both models classified the incident as *real*, emphasizing narrative coherence, sector-specific detail, and plausible operational uncertainty. However, neither model integrated cross-dimensional anomalies or questioned whether the available forensic evidence was jointly sufficient to support a genuine attack. While Gemini briefly noted a minor timeline ambiguity, this signal was not treated as decisive and did not affect the final classification (marked using ∼ symbol in Table 2).

This case illustrates how debate-driven orchestration of expert perspectives, rather than scale alone, can provide robustness in adversarial, detail-sensitive domains such as cybersecurity. For completeness and reproducibility, we provide the full incident specification, representative model outputs (including counterfactual "what-if" reasoning), and a detailed breakdown of the discussion process in Appendix J to complement the formal counterfactual definitions.

## 7. Conclusion

In this paper, we presented ARGORA, an orchestrated multi-expert decision-making framework that builds structured argumentation graphs and casts their evaluation as a structural causal model for decision-level explanations. This enables edge-local counterfactual interventions that identify influential argumentative channels. In addition, ARGORA introduces an observation-aligned counterfactual override policy that performs cost-regularized self-correction when the internally induced consensus is misaligned with an external observational judgment. Across diverse benchmarks and a case study, ARGORA is competitive in performance while providing argumentative utility, structural stability, and mechanistic explainability not typically available in existing multi-LLM discussion frameworks. Future research directions include richer intervention families beyond single-edge edits and human-centered evaluation of causal explanations.

## Impact Statement

This paper presents work whose goal is to advance the field of Machine Learning, more specifically, the mechanistic interpretability and reliability of multi-expert decision-making via structured argumentation and causal interventions. There are many potential societal consequences of our work, none which we feel must be specifically highlighted here.

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

# A. An Illustrative Overview of ARGORA

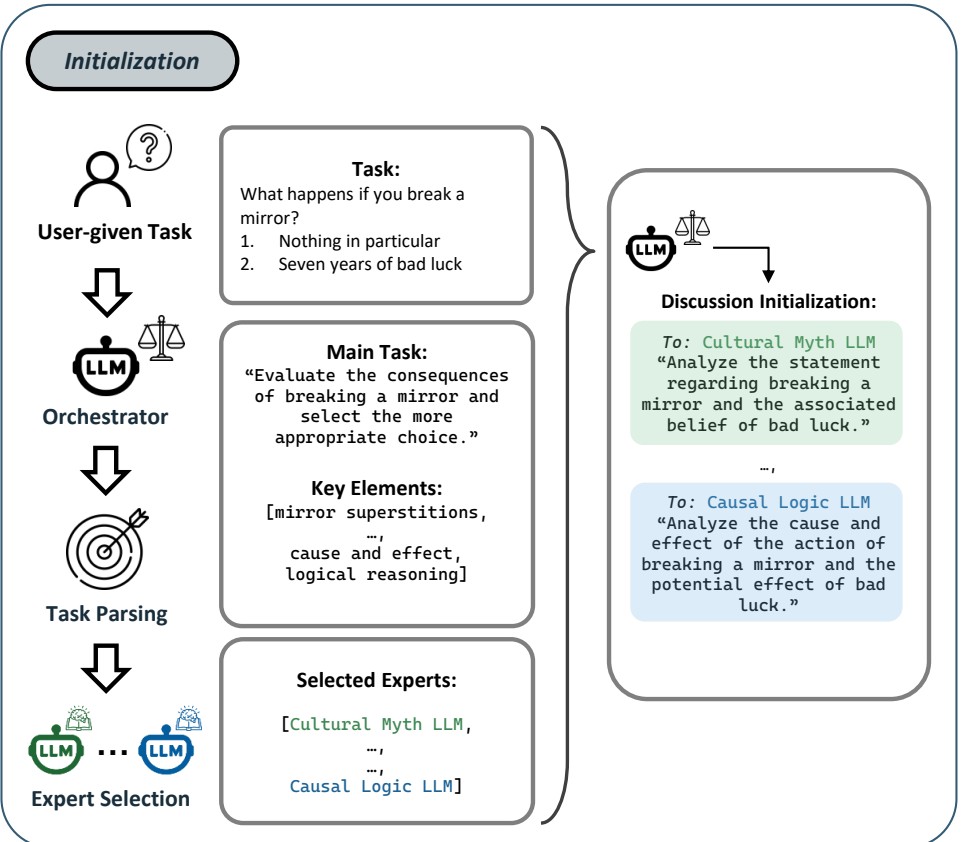

*Figure 1.* **ARGORA overview (Initialization).** Pre-discussion parsing, key-element extraction, expert selection, and prompt generation (Sec. D.1, Prompt. I.1.4).

We provide an end-to-end illustrated guide to ARGORA in Figs. 1–3, using a toy myth-buster-like topic (also referred to as *Task* in the figure) for illustration. All components are identical in the benchmark evaluations; only task prompts and expert instantiations differ.

**Initialization.**   As shown in Fig. 1, ARGORA begins with a pre-discussion initialization phase in which the Orchestrator parses the user input topic into a main task and extracts key elements that will guide the subsequent discussion (Sec. D.1). These outputs determine (i) which specialized expert LLM instances are selected and (ii) how per-expert discussion prompts are generated. In particular, the Orchestrator uses the topic, the parsed main task, and the extracted key elements to synthesize a tailored prompt for each expert (Prompt. I.1.4), ensuring that experts receive distinct roles and guidance aligned with the same underlying decision problem.

**Expert discussion and QBAF construction.**   Fig. 2 illustrates the discussion phase. Conditioned on the Orchestrator-generated prompts, experts first produce an initial main argument and then expand it hierarchically up to three levels (Algs. 3–5) by generating supplementary arguments that either support or attack the main argument. ARGORA incrementally constructs one QBAF per main argument as supplementary arguments are generated. Each QBAF is rooted at its corresponding main argument and encodes support and attack relations among the generated arguments. To prevent redundancy and maintain argumentative diversity, ARGORA applies contextual orthogonality pruning to remove near-duplicate or semantically overlapping arguments, promoting a diverse set of argumentative contributions (Sec. D.4). Each retained argument is assigned a base score $w$ derived from the Orchestrator (Sec. D.8) and evaluated under the selected modular semantics to yield final argument strengths $\sigma$ across the constructed QBAFs (Alg. 7). The final strengths of the main arguments are compared to determine the argumentative winner $m^\star$ (Def. 4.5).

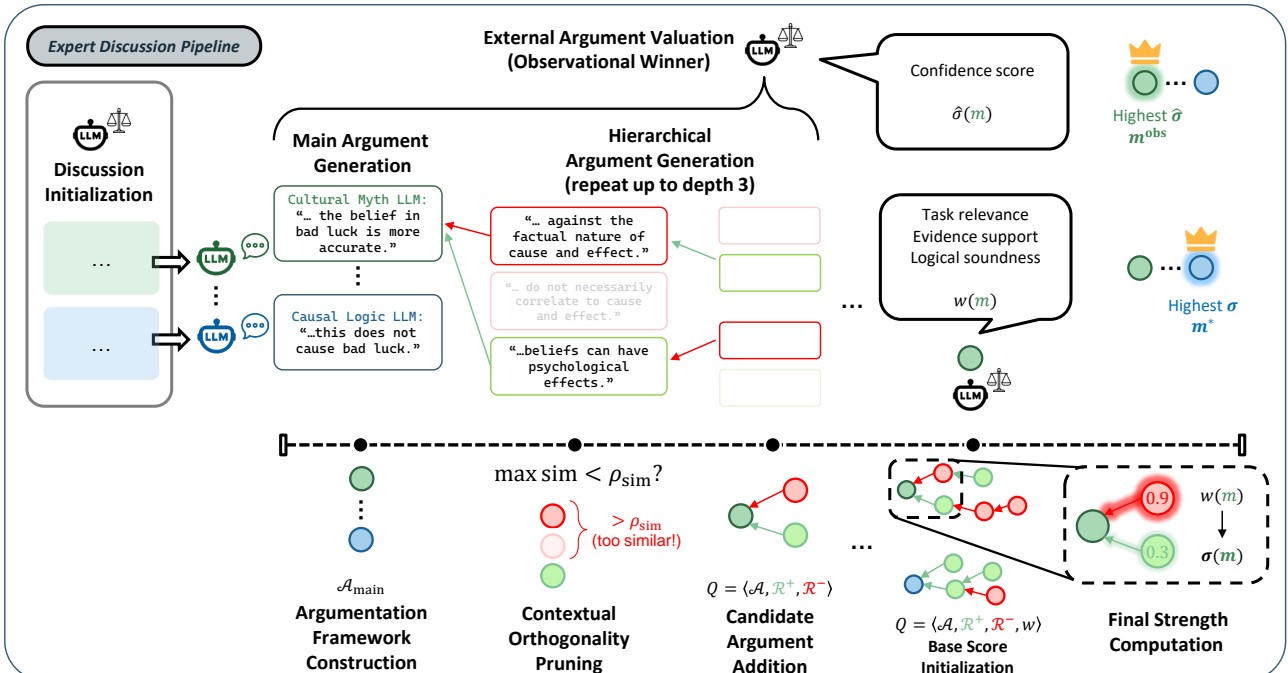

*Figure 2.* **ARGORA overview (Expert discussion).** Hierarchical argument elicitation, QBAF construction/pruning, and semantics-based aggregation (Alg. 3–5, Sec. D.4, Sec. D.8, Alg. 7).

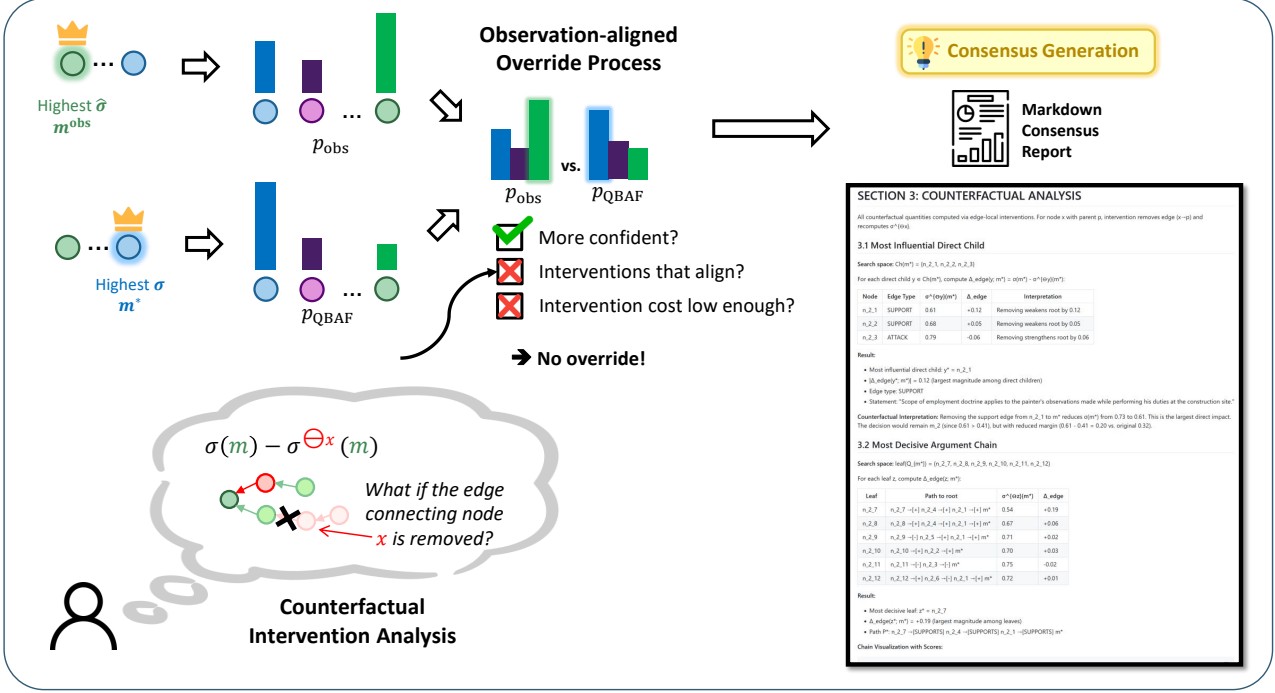

*Figure 3.* **ARGORA overview (Consensus and diagnostics).** SCM casting, counterfactual interventions, and observation-aligned override logic (Sec. 4.3, Def. 4.2, Def. 4.6).

**Consensus generation, counterfactual diagnostics, and overrides.** Finally, Fig. 3 summarizes the post-discussion stage. For explainability, ARGORA casts each constructed QBAF into a deterministic SCM induced by the chosen quantitative semantics (Sec. 4.3). This enables edge-local counterfactual interventions that selectively remove individual influence edges while holding node-local mechanisms and base scores fixed (Def. 4.2), allowing ARGORA to quantify the causal impact of arguments and argument chains on the final decision (Defs. 4.3–4.4). In parallel, ARGORA constructs a structurally independent observational consensus by re-using the Orchestrator in an *LLM-as-a-judge* role with no access to QBAF scores or graph structure; given only the main arguments and a compressed discussion transcript, it produces an observational distribution $p_{\text{obs}}$ and winner $m^{\text{obs}}$ (Def. 4.5). When the observational and argumentative winners disagree, ARGORA evaluates a family of candidate counterfactual intervention configurations and re-evaluates the QBAFs to obtain an intervened distribution $p_{\text{QBAF}}^{\mathcal{I}}$, which is used to determine eligibility for an observation-aligned override (Def. 4.6). The final output is a consensus decision together with a structured Markdown report documenting the winning arguments, confidence scores, counterfactual analyses, and causal diagnostics supporting the decision (App. F).

## B. Extended Related Works

This appendix provides an extended comparison of ARGORA with recent work on how LLMs reason, retrieve information, and engage in multi-agent discussions. We identify two key limitations in existing approaches that motivate ARGORA's design:

1. **Unreliable explanations:** The step-by-step reasoning that LLMs produce cannot always be relied upon as an accurate reflection of their decision-making process, making it hard to diagnose errors.

2. **Lack of justification for decisions:** Many methods improve reliability by generating multiple answers and voting, but they offer limited explainability about the reasoning behind the final answer.

ARGORA addresses these gaps by representing multi-agent discussions as structured argument graphs (called QBAFs) and treating the evaluation process as a deterministic mathematical model (specifically, a Structural Causal Model or SCM). This design enables us to perform targeted "what-if" analyses.

**Interpretability of LLM Reasoning** Prompting techniques like Chain-of-Thought (CoT) ask LLMs to show their work step-by-step, which can improve performance on complex tasks (Wei et al., 2022a). However, these generated reasoning traces may be unfaithful, and reasoning models can exhibit collapse when tasks become more complex (Shojaee et al., 2025; Anthropic, 2025; Turpin et al., 2024). For high-stakes applications, we need more than plausible-sounding explanations—we need mechanisms that can identify which reasoning steps were actually decisive and measure how sensitive the final answer is to changes in those steps.

Graph-of-Thoughts (GoT) organizes intermediate reasoning states into a directed graph, enabling branching, merging, and iterative refinement beyond linear CoT (Besta et al., 2024; Yao et al., 2023a). This is well-suited for inference-time search and aggregation, where node expansion and merging are guided by prompts and heuristic scoring. In contrast, ARGORA focuses on argumentative structure: relations are explicitly typed (support/attack) and evaluated under a deterministic quantitative semantics, which yields a persistent object for diagnosis and counterfactual intervention.

**Limitations of Self-Consistency** Self-Consistency (SC) improves answer quality by generating multiple reasoning paths and selecting the most common answer through majority voting Wang et al. (2022). While effective, this approach has fundamental structural limitations: SC achieves better accuracy primarily by generating many solutions and hoping the correct one appears most frequently. This means computational costs scale roughly linearly with the number of samples, and many of these samples are essentially saying the same thing in different words—they provide repetition rather than genuinely diverse perspectives.

When SC produces a wrong answer, the method doesn't tell us why the incorrect reasoning path won the vote. We don't learn whether the error stemmed from a flawed premise, a logical mistake, or a systematic misunderstanding. The voting process treats all paths equally without identifying which specific steps led to failure.

Since SC treats each reasoning path as a disposable sample, we can't easily test "what-if" scenarios. We could generate new samples, but we can't directly intervene on the existing reasoning structure. Subsequent improvements that add selective

resampling or internal filtering (Chen et al., 2024c) still largely treat reasoning paths as temporary artifacts rather than analyzable objects for causal investigation.

**Multi-Agent Systems** Multi-agent frameworks attempt to improve reasoning through collaboration among multiple AI agents. For example, Mixture-of-Agents uses layered refinement where each layer of agents builds upon previous layers (Wang et al., 2024a), while debate protocols have agents argue iteratively to reach better conclusions (Liang et al., 2024).

However, these methods typically produce only conversation transcripts. The crucial moments where the discussion turned toward the right or wrong answer are buried in unstructured text, making it difficult to trace failures back to their root causes. Without a structured representation of the debate, it's challenging to perform targeted interventions or to measure how a particular critique has an effect the outcome. ARGORA addresses this by converting discussions into an argumentation framework that can be evaluated deterministically and queried at specific points for structured explanations than simply reviewing transcripts.

*Table 3.* Comparative Analysis of Reasoning and Aggregation Approaches for Large Language Models

| Feature | Self-Consistency (Wang et al., 2022) | Graph-of-Thoughts (Besta et al., 2024) | Mixture-of-Agents (Wang et al., 2024a) | MAD (Liang et al., 2024) | ARGORA |
|---|---|---|---|---|---|
| **Core Mechanism** | Answer marginalization (often majority vote) | Arbitrary graph topology operations (Aggregate, Refine) | Layered agent collaboration and synthesis | Multi-agent debate with judge | Quantitative Bipolar Argumentation (QBAF) + SCM |
| **Aggregation** | Statistical (Frequency-based) | Operator-based graph transforms (often LLM-mediated) + scoring/pruning | Generative synthesis (LLM prompt) | Judge-mediated debate synthesis | Formal semantics |
| **Interpretability** | Low (Opaque path selection) | Medium (Visible workflow) | Medium (Visible workflow) | Medium (Visible debate transcript) | **High** (Explicit Support/Attack structure & Base Scores) |
| **Causal Insight** | No explicit causal model | No explicit causal model | No explicit causal model | No explicit causal model | **Supported** (Edge-local interventions and Counterfactuals) |
| **Redundancy** | High (Major limitation, linear cost scaling) | Medium (Mitigated by pruning/selection) | Medium (Layered redundancy) | Medium (Mitigated by debate dynamics) | **Low** (Controlled via Contextual Orthogonality Pruning) |
| **Corrective Action** | No explicit correction mechanism | Feedback loops (Generative) | Iterative refinement (layered synthesis) | Judge-mediated resolution | Observation-Aligned Counterfactual Override |

**Summary** Existing approaches typically involve trade-offs between diversity, structure, and the ability to diagnose problems. Sampling-based methods like Self-Consistency add diversity through multiple attempts but remain opaque about why particular answers won. Multi-agent debate adds rich interaction but leaves only conversation transcripts that are difficult to analyze systematically.

ARGORA unifies these desirable properties by: (1) compiling diverse expert perspectives into a typed argument graph with explicit support and attack relationships, (2) evaluating this graph using deterministic mathematical rules rather than another LLM call, and (3) enabling counterfactual analysis through the SCM framework, along with cost-regularized overrides when needed. The result is an artifact that supports both competitive decision quality and principled, localized diagnosis—capabilities that are difficult to obtain from purely sampling-based methods or purely prompt-mediated aggregation.

# C. Extended Preliminaries and Proofs.

This section collects the formal preliminaries and conventions used throughout the technical development and the complete proofs to serve as an extension to Sections 3 and 4 in the main paper.

## C.1. Abstract arguments, QBAFs, rooted trees, and neighborhood notation

**Definition C.1** (Argument). Generally, in abstract argumentation, no internal structure is enforced in an argument. As such, in ARGORA, a single *argument* is defined to be a free-form textual model output.

**Definition C.2** (QBAF (*recall*)). A *quantitative bipolar argumentation framework (QBAF)* is a tuple

$$Q = \langle \mathcal{A}, R^-, R^+, w \rangle,$$

where $\mathcal{A}$ is a finite set of arguments, $R^- \subseteq \mathcal{A} \times \mathcal{A}$ is the (directed) *attack* relation, $R^+ \subseteq \mathcal{A} \times \mathcal{A}$ is the (directed) *support* relation, and $w : \mathcal{A} \to [0, 1]$ assigns each $a \in \mathcal{A}$ a base score. We write $R := R^- \cup R^+$.

**Definition C.3** (Rooted directed tree QBAF (*recall*)). A QBAF $Q = \langle \mathcal{A}, R^-, R^+, w \rangle$ is a *rooted tree* (with root $m \in \mathcal{A}$) if $(\mathcal{A}, R)$ is a finite directed tree with unique root $m$ and each non-root node has a unique parent. All edges are oriented *child→parent* as in Def. 3.1. For any non-root $a \in \mathcal{A}$, its unique parent is $\mathbf{Pa}(a)$, and for any $a \in \mathcal{A}$, its children are

$$\mathbf{Ch}(a) := \{ c \in \mathcal{A} : (c \to a) \in R \}.$$

**Polarity signs and vectors.** For any edge $(c \to a) \in R$, define

$$\epsilon(c, a) := \begin{cases} +1, & (c \to a) \in R^+, \\ -1, & (c \to a) \in R^-. \end{cases}$$

If $\mathbf{Ch}(a) = \{c_1, \dots, c_n\}$ is any fixed ordering, define the polarity vector

$$\pi(a) := (\epsilon(c_1, a), \dots, \epsilon(c_n, a)) \in \{-1, +1\}^n.$$

(For leaves, $n = 0$.)

## C.2. Gradual semantics and modular semantics

**Definition C.4** (Gradual semantics). A (well-defined) gradual semantics assigns to every QBAF $Q = \langle \mathcal{A}, R^-, R^+, w \rangle$ a *strength function*

$$\sigma : \mathcal{A} \to [0, 1], \qquad a \mapsto \sigma(a),$$

which gives the final acceptability degree of each argument.

We use the modular view of quantitative semantics (Rago et al., 2016; Potyka, 2018; Baroni et al., 2018; Mossakowski & Neuhaus, 2018).

**Definition C.5** (Modular update equation (*recall*)). Fix a modular pair $(\alpha, \iota)$ as in Def. 3.2. For any rooted-tree QBAF $Q = \langle \mathcal{A}, R^-, R^+, w \rangle$ and any $a \in \mathcal{A}$ with $\mathbf{Ch}(a) = \{c_1, \dots, c_n\}$ (any fixed ordering) and polarity vector $\pi(a)$, a strength function $\sigma : \mathcal{A} \to [0, 1]$ is *admissible* if it satisfies

$$\sigma(a) = \iota_{w(a)} \big( \alpha_{\pi(a)} \big( \sigma(c_1), \dots, \sigma(c_n) \big) \big). \tag{2}$$

In the leaf case ($n = 0$), we assume the semantics are normalized so that $\iota_{w(a)}(\alpha_{\pi(a)}()) = w(a)$, hence $\sigma(a) = w(a)$.

**Common modular building blocks.** For completeness, Table 4 lists common choices of aggregation and influence functions used in the literature and in the implementation of ARGORA.

**Assumption C.6** (Permutation invariance of the modular aggregator). For any $a \in \mathcal{A}$ with $\mathbf{Ch}(a) = \{c_1, \dots, c_n\}$, let $\pi(a) = (\epsilon(c_1, a), \dots, \epsilon(c_n, a)) \in \{-1, +1\}^n$ be the polarity vector under some ordering. We assume that for every permutation $\tau$ of $\{1, \dots, n\}$ and any $x_1, \dots, x_n \in [0, 1]$,

$$\alpha_{(\pi_1, \dots, \pi_n)}(x_1, \dots, x_n) = \alpha_{(\pi_{\tau(1)}, \dots, \pi_{\tau(n)})}(x_{\tau(1)}, \dots, x_{\tau(n)}).$$

Equivalently, $\alpha$ depends only on the multiset $\{(\pi_i, x_i)\}_{i=1}^n$ and not on the chosen ordering of children.

*Remark* C.7 (Order-independence of the modular equation). Assumption C.6 implies that the right-hand side of (2) is independent of how $\mathbf{Ch}(a)$ is ordered. Thus the phrase "any fixed ordering" in Def. C.5 is without loss of generality.

*Table 4.* Building blocks of modular semantics (Def. 3.2). Aggregation functions summarize the effects of supporters and attackers; influence functions transform the aggregated value into a final strength.

| **Aggregation functions** $\alpha_\pi(s)$ | |
| --- | --- |
| **Sum** | $\alpha_\pi^\Sigma(s) = \sum_{i=1}^{n} \pi_i \, s_i$ |
| **Product** | $\alpha_\pi^\Pi(s) = \prod_{i:\pi_i=-1} (1-s_i) - \prod_{i:\pi_i=1}(1-s_i)$ |
| **Top** | $\alpha_\pi^{\max}(s) = M_\pi(s) - M_{-\pi}(s), \; M_\pi(s) = \max\{0, \pi_i s_i\}$ |

| **Influence functions** $\iota_w(s)$ | |
| --- | --- |
| **Linear**$(\kappa)$ | $\iota_w^\ell(s) = w - \dfrac{w}{\kappa} \max\{0, -s\} + \dfrac{1-w}{\kappa} \max\{0, s\}$ |
| **Euler-based** | $\iota_w^e(s) = 1 - \dfrac{1-w^2}{1 + w \exp(s)}$ |
| $p$-**Max**$(\kappa)$ | $\iota_w^p(s) = w - w \, h_p(-s/\kappa) + (1-w)\, h_p(s/\kappa)$ |

*Parameters / auxiliary functions.* $\kappa > 0$ is a *conservativeness* (damping) parameter that scales the aggregated input: larger $\kappa$ makes updates more conservative (changes stay closer to $w$). For Linear$(\kappa)$ we assume the aggregation output lies in $[-\kappa, \kappa]$ (e.g., by design of $\alpha_\pi$ or by clipping), ensuring $\iota_w^\ell(s) \in [0,1]$. For $p$-Max, $p \in \{1, 2, \dots\}$ and $h_p(x) = \dfrac{\max\{0, x\}^p}{1 + \max\{0, x\}^p}$.

| **Common modular pairings** | |
| --- | --- |
| DF-QuAD$(\kappa)$ (Rago et al., 2016) | Product + Linear$(\kappa)$ |
| Euler/REB (Amgoud & Ben-Naim, 2018) | Sum + Euler-based |
| QE$(\kappa)$ (Potyka, 2018) | Sum + 2-Max$(\kappa)$ |
| SD-DF-QuAD$(\kappa)$ (Kampik et al., 2024b) | Product + 1-Max$(\kappa)$ |
| EBT (Kampik et al., 2024b) | Top + Euler-based |

## C.3. Structural causal models and interventions

**Definition C.8** (Deterministic SCM). A deterministic structural causal model (SCM) is a tuple $\mathcal{M} = (\mathbf{V}, \mathbf{U}, \mathcal{F})$, where:

- $\mathbf{V}$ is a finite set of endogenous variables;

- $\mathbf{U}$ is a set of exogenous variables; and

- $\mathcal{F} = \{f_v : v \in \mathbf{V}\}$ is a family of structural assignments

$$v = f_v(u_v, \mathbf{Pa}_\mathcal{M}(v)),$$

where $u_v \in \mathbf{U}$ denotes the exogenous input associated with $v$ and $\mathbf{Pa}_\mathcal{M}(v) \subseteq \mathbf{V}$ is the set of endogenous parents of $v$ in the induced causal graph.

In a deterministic SCM, the structural assignments contain no stochastic noise terms, and (when the induced graph is acyclic) the values of all variables in $\mathbf{V}$ are uniquely determined by the equations in $\mathcal{F}$ under fixed exogenous values.

We adapt the conventions for SCM interventions laid out by Massidda et al. (2023) for the following definitions.

**Definition C.9** (Intervention). Let $\mathcal{M} = (\mathbf{V}, \mathbf{U}, \mathcal{F})$ be a deterministic SCM with $\mathcal{F} = \{f_v : v \in \mathbf{V}\}$ and equations

$$v = f_v(u_v, \mathbf{Pa}_\mathcal{M}(v)).$$

An *intervention* on a subset $W \subseteq \mathbf{V}$ is specified by replacement mechanisms $\{g_w : w \in W\}$ (where each $g_w$ is a function of $(u_w, \mathbf{Pa}_\mathcal{M}(w))$) and yields the intervened model

$$\mathcal{M}_{(W \leftarrow g)} := (\mathbf{V}, \mathbf{U}, \mathcal{F}_{(W \leftarrow g)}), \qquad \mathcal{F}_{(W \leftarrow g)} := (\mathcal{F} \setminus \{f_w : w \in W\}) \cup \{g_w : w \in W\}.$$

**Definition C.10** (Soft intervention). An intervention $(W \leftarrow g)$ is *soft* if each replacement mechanism $g_w$ for $w \in W$ is allowed to depend only on $(u_w, \mathbf{Pa}_{\mathcal{M}}(w))$ and may ignore any subset of its original inputs (but cannot depend on any variable outside $\{u_w\} \cup \mathbf{Pa}_{\mathcal{M}}(w)$).

In ARGORA, we do not aim to *force* an argument node to a fixed value; rather, we ask a mechanistic counterfactual question: what changes if a *single* influence edge is removed? This naturally affects only one structural equation while keeping all other mechanisms intact, which is consistent with soft interventions.

**Edge deletion as restricted-mechanism replacement.** Deleting an influence edge $(x \rightarrow p)$ in the rooted QBAF corresponds, in the induced SCM $\mathcal{M} = (\mathbf{V}, \mathbf{U}, \mathcal{F})$, to replacing only the mechanism $f_p$ by a restricted mechanism $f'_p$ that omits the input $v_x$:

$$v_p = f_p\big(u_p, \mathbf{Pa}_{\mathcal{M}}(v_p)\big) \quad \rightsquigarrow \quad v_p = f'_p\big(u_p, \mathbf{Pa}_{\mathcal{M}}(v_p) \setminus \{v_x\}\big),$$

leaving all other mechanisms unchanged.

### C.4. Structural conditions for well-posed evaluation and SCM casting

The following conditions are referenced in Prop. 4.1. This set of assumptions is used to formalize the restricted graph class and invariances assumed when we interpret QBAF evaluation as a deterministic SCM and when we perform edge-local edits.

**Assumption C.11** (Conditions for SCM casting under edge edits). Fix a per-main QBAF $Q_m = \langle \mathcal{A}_m, R_m^-, R_m^+, w_m \rangle$ and write $R_m := R_m^- \cup R_m^+$.

1. (**Rooted directed tree**) $Q_m$ is a rooted directed tree with root $m$ in the sense of Def. C.3; in particular, every non-root argument has a unique parent and edges are oriented child→parent.

2. (**Node-local modular update rule**) Evaluation uses a fixed modular pair $(\alpha, \iota)$ (Def. 3.2). Any admissible strength function $\sigma : \mathcal{A}_m \rightarrow [0, 1]$ must satisfy, for every $a \in \mathcal{A}_m$, the modular update equation (2).

3. (**Mechanism invariance under edge edits**) Under edge addition/deletion in $R_m$, we keep fixed: (i) all base scores $w_m(a)$ and (ii) the modular mechanisms $(\alpha, \iota)$ (equivalently, the induced node-local structural functions). Only the adjacency structure (and thus which child inputs are provided to $\alpha_{\pi(a)}$) is allowed to change.

### C.5. Proof of Proposition 4.1 (A1 and A2)

*Proof of Proposition 4.1.* We prove the two claims stated in the proposition.

**(A1) Well-posed evaluation (existence and uniqueness of $\sigma$).** Fix $Q_m = \langle \mathcal{A}_m, R_m^-, R_m^+, w_m \rangle$ and a modular pair $(\alpha, \iota)$ as in Assumption C.11. Because $Q_m$ is a finite rooted tree oriented child→parent, we can evaluate strengths bottom-up.

Define the *height* of a node $a \in \mathcal{A}_m$ recursively by

$$\mathrm{ht}(a) := \begin{cases} 0, & \mathbf{Ch}(a) = \varnothing, \\ 1 + \max_{c \in \mathbf{Ch}(a)} \mathrm{ht}(c), & \text{otherwise.} \end{cases}$$

Since $Q_m$ is finite and acyclic, $\mathrm{ht}(a) < \infty$ for all $a$.

*Existence.* We construct $\sigma : \mathcal{A}_m \rightarrow [0, 1]$ by induction on height. For each leaf $a$ (i.e., $\mathrm{ht}(a) = 0$), set

$$\sigma(a) := \iota_{w_m(a)}\Big(\alpha_{\pi(a)}(\,)\Big),$$

where $\alpha_{\pi(a)}(\,)$ denotes the $n = 0$ (empty-input) case. Now assume $\sigma(c)$ has been defined for all nodes $c$ with $\mathrm{ht}(c) \leq k$. Let $a$ be any node with $\mathrm{ht}(a) = k + 1$ and write $\mathbf{Ch}(a) = \{c_1, \ldots, c_n\}$. Then each child satisfies $\mathrm{ht}(c_i) \leq k$, so $\sigma(c_i)$ is already defined, and we may set

$$\sigma(a) := \iota_{w_m(a)}\Big(\alpha_{\pi(a)}\big(\sigma(c_1), \ldots, \sigma(c_n)\big)\Big).$$

Proceeding until the maximum height yields a total map $\sigma$ satisfying the modular equations (2) at every node.

*Uniqueness.* Suppose $\tilde{\sigma} : \mathcal{A}_m \to [0, 1]$ is any other function satisfying (2) for all nodes. We show $\tilde{\sigma} = \sigma$ by induction on height. For a leaf $a$, both $\sigma(a)$ and $\tilde{\sigma}(a)$ are forced by the same leaf equation (the $n = 0$ case), hence $\tilde{\sigma}(a) = \sigma(a)$. Assume $\tilde{\sigma}(c) = \sigma(c)$ for all $\mathrm{ht}(c) \le k$. Let $a$ have $\mathrm{ht}(a) = k + 1$ with children $\{c_1, \dots, c_n\}$. Applying (2) to $\tilde{\sigma}$ and using the induction hypothesis on each child yields

$$\tilde{\sigma}(a) = \iota_{w_m(a)}\Big(\alpha_{\pi(a)}\big(\tilde{\sigma}(c_1), \dots, \tilde{\sigma}(c_n)\big)\Big) = \iota_{w_m(a)}\Big(\alpha_{\pi(a)}\big(\sigma(c_1), \dots, \sigma(c_n)\big)\Big) = \sigma(a).$$

Thus $\tilde{\sigma}(a) = \sigma(a)$ for all nodes, proving uniqueness. In other words, if we fix a modular semantics $(\alpha, \iota)$ as specified in Def. 3.2, then the induced strength function $\sigma$ is well-defined and unique.

**(A2) Casting the evaluation system as a deterministic SCM.** We now construct the SCM satisfying the proposition. Define endogenous variables

$$\mathbf{V} := \{ v_a : a \in \mathcal{A}_m \},$$

and introduce exogenous inputs encoding base scores via

$$\mathbf{U} := \{ u_a : a \in \mathcal{A}_m \}, \qquad \text{with fixed instantiation } u_a := w_m(a).$$

For each node $a \in \mathcal{A}_m$, let $\mathbf{Ch}(a) = \{c_1, \dots, c_n\}$ be any fixed ordering (possibly $n = 0$), and let $\pi(a) \in \{-1, +1\}^n$ be the corresponding polarity vector. Define the node-local structural assignment

$$v_a = f_{v_a}\big(u_a, \mathbf{Pa}_{\mathcal{M}_m}(v_a)\big) := \iota_{u_a}\Big(\alpha_{\pi(a)}(v_{c_1}, \dots, v_{c_n})\Big), \tag{3}$$

where the endogenous parent set is

$$\mathbf{Pa}_{\mathcal{M}_m}(v_a) := \{ v_c \in \mathbf{V} : c \in \mathbf{Ch}(a) \}.$$

Equation (3) is exactly the modular update equation (2) under the identification $u_a = w_m(a)$ and $v_a = \sigma(a)$.

Because $Q_m$ is a rooted tree with edges oriented child→parent, the induced causal graph on $\mathbf{V}$ has edges $v_c \to v_a$ whenever $(c \to a) \in R_m$ and is acyclic. Hence the deterministic SCM $\mathcal{M}_m = (\mathbf{V}, \mathbf{U}, \mathcal{F})$ with $\mathcal{F} = \{f_{v_a} : a \in \mathcal{A}_m\}$ is well-defined, and its unique solution under fixed exogenous values coincides with the unique strength function $\sigma$ from (A1). In particular, the root variable $v_m$ equals the evaluated root strength $\sigma(m)$.

Finally, Assumption C.11(3) ensures that when edges are added/removed (e.g., for edge-local interventions), the mechanisms $f_{v_a}$ and the exogenous base scores are held fixed: only the parent/child incidence (and thus the input list to $\alpha_{\pi(a)}$) changes. This is precisely the invariance required to interpret such graph edits as causal interventions within the SCM. $\qquad\square$

**Lemma C.12** (Well-posedness under edge deletions). *Let $Q = \langle \mathcal{A}, R^-, R^+, w \rangle$ be a rooted QBAF tree as in Def. C.3, and fix a modular pair $(\alpha, \iota)$. Let $R' \subseteq R$ be any subset of edges (obtained by deleting edges), and set $R'^- := R' \cap R^-$, $R'^+ := R' \cap R^+$, and*

$$Q' := \langle \mathcal{A}, R'^-, R'^+, w \rangle.$$

*Then the modular system (2) (with child sets induced by $R'$) admits a unique strength function $\sigma' : \mathcal{A} \to [0, 1]$. If Assumption C.6 holds, this $\sigma'$ is independent of the chosen orderings of the children in $Q'$.*

*In particular, the edge-local deletions used in the main paper (deleting a single edge $(x \to p)$) yield a well-defined and unique counterfactual root strength.*

*Proof.* Since $(\mathcal{A}, R)$ is a finite directed tree, any subgraph $(\mathcal{A}, R')$ obtained by deleting edges is still finite and acyclic, and every node has at most one parent. Hence $(\mathcal{A}, R')$ is a directed forest (possibly disconnected), with edges still oriented child→parent.

Write the children in $Q'$ as

$$\mathbf{Ch}'(a) := \{ c \in \mathcal{A} : (c \to a) \in R' \}.$$

Define the height $\mathrm{ht}'(a)$ recursively by

$$\mathrm{ht}'(a) := \begin{cases} 0, & \mathbf{Ch}'(a) = \varnothing, \\ 1 + \max_{c \in \mathbf{Ch}'(a)} \mathrm{ht}'(c), & \text{otherwise.} \end{cases}$$

Acyclicity and finiteness imply $\mathrm{ht}'(a) < \infty$ for all $a$.

*Existence.* Construct $\sigma'$ by induction on $\mathrm{ht}'$ exactly as in the proof of Proposition 4.1(A1): set leaves by the (empty-input) modular equation, and then evaluate each node once all its children have been assigned.

*Uniqueness.* Let $\sigma'_1, \sigma'_2$ satisfy the modular equations on $Q'$. By induction on height: they coincide on all leaves (height 0). If they coincide on all nodes of height $\leq k$, then for any node $a$ with height $k+1$, all children $c \in \mathbf{Ch}'(a)$ have height $\leq k$, hence $\sigma'_1(c) = \sigma'_2(c)$, and the modular update equation forces $\sigma'_1(a) = \sigma'_2(a)$. Thus $\sigma'_1 = \sigma'_2$ on all of $\mathcal{A}$.

Finally, if Assumption C.6 holds, the value of each update is independent of the chosen ordering of $\mathbf{Ch}'(a)$, so $\sigma'$ is order-independent. $\qquad\square$

*Remark* C.13 (Irrelevance of disconnected components for the root). Let $Q' = \langle A, R'^{-}, R'^{+}, w \rangle$ be obtained from a rooted tree $Q$ by deleting edges, so $(A, R')$ is a directed forest (Lemma C.12). Fix the root $m$ of the original tree and define the root-relevant set

$$A_m^{\uparrow} := \{a \in A : \text{there exists a directed path } a \to \cdots \to m \text{ in } (A, R')\}.$$

Let $Q'|_{A_m^{\uparrow}}$ denote the induced sub-QBAF on $A_m^{\uparrow}$ (with inherited edges and weights). Then the evaluated root strength computed on $Q'$ equals that computed on $Q'|_{A_m^{\uparrow}}$; in particular, any component disconnected from $m$ is irrelevant to $\sigma'(m)$. For an edge-local deletion $(x \to p)$, the detached subtree rooted at $x$ lies outside $A_m^{\uparrow}$ and therefore has no effect on the counterfactual root strength $\sigma^{\ominus x}(m)$.

# D. Extended Methodology for ARGORA

This section provides a detailed specification of the argument generation pipeline and QBAF construction procedure in ARGORA. We describe the main pipeline stages algorithmically and discuss the design rationale.

## D.1. Pre-discussion Initialization

ARGORA takes as input a user-defined topic $t$. Depending on the subject, $t$ may bundle multiple sub-questions, implicit assumptions, open-ended goals, or present multiple choices. We therefore establish two anchors before any discussion takes place:

- **Main task** $s_M$: The main task converts an open topic into a single, well-formed *imperative*—the concrete decision or query that ARGORA must resolve.

- **Key elements** $K$: Key elements are the salient factors, entities, and conditions relevant to the user-provided topic and the main task. They serve as a *shared artifact* for evidence generation, retrieval, and critique, and as coverage targets for the discussion.

These two anchors guide the participating experts on *what* needs to be decided and *which* dimensions matter.

**Main task extraction.** The main task $s_M$ is extracted from the user topic $t$ via a prompted LLM call to the Orchestrator. The extraction prompt (see Prompt I.1.1) instructs the Orchestrator to produce a single-sentence, well-formed imperative that specifies the core objective of $t$.

**Example (topic and main task):**

- *Topic* $t$: "Should our team adopt quantization for a 70B LLM serving pipeline to cut latency while maintaining accuracy?"

- *Main task* $s_M$: "Decide whether to adopt quantization for the 70B LLM serving pipeline and justify the decision with expected latency–accuracy trade-offs."

The main task clarifies that the output should be a *decision* with *justification*, not merely a comparison or analysis, which anchors the discussion around a concrete resolution.

**Key element extraction.** The key-element set $K$ is extracted via another LLM call to the Orchestrator, conditioned on both $t$ and $s_M$ (see Prompt I.1.2). The extraction prompt instructs the Orchestrator to identify a finite set of intended semantics—key entities, factors, conditions, or constraints—that are directly relevant to addressing $(t, s_M)$. These elements serve as explicit coverage targets: experts are encouraged to ground their arguments in these dimensions, and the Orchestrator uses them when assessing argument relevance during base score assignment (see Prompt I.1.5 and Appendix D.8).

**Example (key elements for the above topic and main task):**

$$K = \{\text{``70B model family''}, \text{``target latency budget''}, \text{``throughput''}, \text{``accuracy metric''}\}$$

**Orchestrator and expert roles.** ARGORA uses one **Orchestrator** model and a user-defined number of **expert** models. The Orchestrator is responsible for:

1. Extracting $(s_M, K)$ from the user topic $t$ (see Prompts I.1.1 and I.1.2)

2. Selecting and assigning domain-specific roles to each expert in the expert set $E$ (see Prompt I.1.3)

3. Generating tailored, role-conditioned prompts for each expert at the start of each discussion round (see Prompt I.1.4)

4. Mediating the discussion by collecting expert responses after each discussion round.

5. Assigning prior strengths to argument nodes (see Prompt I.1.5)

6. Providing an external observational judgment for the override mechanism (Section 4.4)

Each **expert** $e \in E$ is an LLM instance conditioned on a domain-specific system prompt (see Prompt I.2.1) that specifies its area of expertise. In each round of ARGORA, the Orchestrator issues a *specialized prompt* to every expert $e \in E$, conditioned on the topic $t$, the main task $s_M$, and key elements $K$ in order to generate the main arguments which serve as the root argument nodes for QBAF generation. Unlike existing discussion frameworks where discussions begin with inter-expert interactions, this process is intentionally *orchestrator↔expert*: experts independently produce responses that are later organized into main arguments. Only after this independent output generation do we introduce cross-expert contestation through supplementary argument generation, which allows us to capitalize on viewpoint diversity before introducing inter-expert influence.

We implement inter-expert independent output generation as a core design principle to:

1. **Maximize diversity.** Collective performance improves when judgments are diverse and formed *independently* before aggregation, a principle established in the wisdom-of-crowds literature (Surowiecki, 2004; Lorenz et al., 2011) and nominal group elicitation methods (Hsu & Sandford, 2007).

2. **Avoid premature convergence.** Turn-taking multi-agent dialogues can induce conformity pressures or early anchoring effects, reducing the effective diversity of the final argument pool. With independent main argument responses, we preserve the full range of perspectives that each expert would naturally generate given only the topic and their role.

3. **Mitigate production blocking.** Sequential turn-taking can suppress idea generation relative to independent parallel production, a phenomenon known as production blocking in group decision-making research (Diehl & Stroebe, 1987; Mullen et al., 1991; Fishkin, 2024). Independent first-pass responses (through main argument generation) mitigate this effect.

4. **Enable system-level concurrency.** This design permits straightforward parallelism: each expert receives an independent, role-conditioned prompt and produces a reply with no shared chat history across experts. LLM prompts can be dispatched concurrently and processed in parallel, which enables near-linear speedups for argument generation along with reduced token usage from independent history tracking.

In the remainder of this appendix, we describe the detailed construction procedures for each level of the discussion process, along with argumentation framework generation.

---

**Algorithm 1** Main Argument Extraction

---

**Require:** Topic $t \in \mathcal{T}$, extracted main task $s_M \in \mathcal{S}$, key elements $K \subseteq \mathcal{K}$, expert set $E$
**Ensure:** Main-argument node set $\mathcal{A}_{\mathrm{main}} \subseteq \mathcal{A}$, source expert map $\mathcal{C} : \mathcal{A}_{\mathrm{main}} \to 2^E$
  1: $\tilde{\mathcal{A}}_{\mathrm{main}} \leftarrow \emptyset$ {Candidate (statement text, expert) pairs in $\Sigma^* \times E$}
  2: {**Phase 1: Query experts in parallel**}
  3: **for all** $e \in E$ **in parallel do**
  4:     $u_e \leftarrow$ QUERYEXPERT$(e; t, s_M, K)$ {Returns a single proposed statement text $u_e \in \Sigma^*$, see Prompt I.2.2}
  5:     $\tilde{\mathcal{A}}_{\mathrm{main}} \leftarrow \tilde{\mathcal{A}}_{\mathrm{main}} \cup \{(u_e, e)\}$
  6: **end for**
  7: {**Phase 2: Construct main argument nodes**}
  8: $\mathcal{A}_{\mathrm{main}} \leftarrow \emptyset$
  9: $\mathcal{C} \leftarrow \emptyset$ {$\mathcal{C}[m]$ stores experts supporting canonical node $m$}
10: **for** each $(u, e) \in \tilde{\mathcal{A}}_{\mathrm{main}}$ **do**
11:     $\bar{m} \leftarrow$ FINDEXACTMATCH$\big(u, \{\lambda(m) : m \in \mathcal{A}_{\mathrm{main}}\}\big)$ {Returns a node $\bar{m} \in \mathcal{A}_{\mathrm{main}}$ s.t. $\lambda(\bar{m})$ matches $u$, or null}
12:     **if** $\bar{m} =$ null **then**
13:         $\bar{m} \leftarrow$ MAKEARG$(u)$ {Creates a new argument node $\bar{m} \in \mathcal{A}$ with label $\lambda(\bar{m}) = u$}
14:         $\mathcal{A}_{\mathrm{main}} \leftarrow \mathcal{A}_{\mathrm{main}} \cup \{\bar{m}\}$
15:         $\mathcal{C}[\bar{m}] \leftarrow \emptyset$
16:     **end if**
17:     $\mathcal{C}[\bar{m}] \leftarrow \mathcal{C}[\bar{m}] \cup \{e\}$
18: **end for**
19: **return** $\mathcal{A}_{\mathrm{main}}, \mathcal{C}$

---

## D.2. Discussion Overview

ARGORA constructs a hierarchical argumentation framework (graph in a form of a tree structure) where arguments are organized into levels based on their relationship to the main claims:

- **Level 0 (Main Arguments):** Direct responses to the discussion topic, representing primary claims or answers.

- **Level 1 (First-Level Arguments):** Supporting or attacking arguments that directly address the main arguments. Each first-level argument is attributed to a specific expert, referred to as its *author*.

- **Level 2 (Second-Level Arguments):** Review arguments from *non-author* experts—those who did not write the first-level argument under review—who evaluate and respond to first-level arguments with supporting or attacking justifications.

- **Level 3 (Third-Level Arguments):** Targeted rebuttals where the original *author* of a first-level argument responds to attacks raised against their argument at level 2, defending their position. By design, third-level arguments can only have an attack relation with respect to its parent argument (second-level argument) due to its inherent role as a targeted rebuttal.

This author/non-author distinction is required to simulate a typical discussion pipeline: experts cannot review their own arguments at level 2, which promotes critical external evaluation, while authors retain the opportunity to defend their claims at level 3.

## D.3. Main Argument Generation and Extraction

The first phase identifies a set of *main argument nodes* that serve as the roots of subsequent argumentation graphs. Given a discussion topic $t \in \mathcal{T}$, extracted main task $s_M \in \mathcal{S}$, and a set of key elements $K \subseteq \mathcal{K}$, we query each expert $e \in E$ in parallel to elicit a single candidate response formulated as a textual statement $u_e \in \Sigma^*$. At this stage, expert outputs are treated purely as text, without assuming any pre-existing graph structure. We provide the algorithm for main argument extraction as implemented in ARGORA in Algorithm 1.

To ensure a well-typed argumentation framework, we explicitly distinguish between argument *nodes* and their textual statements. Argument nodes are elements of the global argument set $\mathcal{A}$, while their textual statements are given by a

---

**Algorithm 2** Contextual Orthogonality Pruning

---

**Require:** Candidate set $\mathcal{A}_{\text{cand}}^{(\ell)}$, threshold $\rho_{\text{sim}} \in [0, 1]$, sentence encoder $\phi : \Sigma^* \to \mathbb{R}^d$

**Ensure:** Pruned candidate set $\mathcal{A}_{\text{sel}}^{(\ell)} \subseteq \mathcal{A}_{\text{cand}}^{(\ell)}$

1: **Define** $\text{sim}(u, v) := \cos\big(\phi(u), \phi(v)\big)$ for $u, v \in \Sigma^*$ {Cosine similarity}
2: $\mathcal{A}_{\text{filt}}^{(\ell)} \leftarrow \emptyset$
3: $a_{\text{fb}} \leftarrow \texttt{null}; \quad \rho_{\min} \leftarrow +\infty$ {Fallback candidate tracking}
4: {**Stage 1: Parent-level orthogonality check**}
5: **for** each $a \in \mathcal{A}_{\text{cand}}^{(\ell)}$ **do**
6:    $\rho_{\text{par}}(a) \leftarrow \max_{p \in \mathcal{A}^{(\ell-1)}} \text{sim}\big(\lambda(a), \lambda(p)\big)$
7:    **if** $\rho_{\text{par}}(a) < \rho_{\min}$ **then**
8:       $a_{\text{fb}} \leftarrow a; \quad \rho_{\min} \leftarrow \rho_{\text{par}}(a)$
9:    **end if**
10:   **if** $\rho_{\text{par}}(a) \leq \rho_{\text{sim}}$ **then**
11:      $\mathcal{A}_{\text{filt}}^{(\ell)} \leftarrow \mathcal{A}_{\text{filt}}^{(\ell)} \cup \{a\}$
12:   **end if**
13: **end for**
14: **if** $\mathcal{A}_{\text{filt}}^{(\ell)} = \emptyset$ **then**
15:   **return** $\{a_{\text{fb}}\}$ if $a_{\text{fb}} \neq \texttt{null}$ else $\emptyset$ {Fallback: least parent-similar}
16: **end if**
17: {**Stage 2: Sibling orthogonality check**}
18: Sort $\mathcal{A}_{\text{filt}}^{(\ell)}$ by $\rho_{\text{par}}(\cdot)$ ascending $\to (a_1, \ldots, a_k)$
19: $\mathcal{A}_{\text{sel}}^{(\ell)} \leftarrow \emptyset$
20: **for** $i = 1, \ldots, k$ **do**
21:   $S_{\text{sib}} \leftarrow \{a' \in \mathcal{A}_{\text{sel}}^{(\ell)} : \mathbf{Pa}(a') = \mathbf{Pa}(a_i)\}$ {Already-selected siblings}
22:   $\rho_{\text{sib}}(a_i) \leftarrow \max_{a' \in S_{\text{sib}}} \text{sim}\big(\lambda(a_i), \lambda(a')\big)$ {0 if $S_{\text{sib}} = \emptyset$}
23:   **if** $\rho_{\text{sib}}(a_i) \leq \rho_{\text{sim}}$ **then**
24:      $\mathcal{A}_{\text{sel}}^{(\ell)} \leftarrow \mathcal{A}_{\text{sel}}^{(\ell)} \cup \{a_i\}$
25:   **end if**
26: **end for**
27: **return** $\mathcal{A}_{\text{sel}}^{(\ell)}$

---

statement-label function $\lambda : \mathcal{A} \to \Sigma^*$. Once we have all expert outputs for main arguments, we perform case-insensitive exact matching on the proposed text $u_e$ against the existing label set $\{\lambda(m) : m \in \mathcal{A}_{\text{main}}\}$. If there exists $m \in \mathcal{A}_{\text{main}}$ whose label matches $u_e$ exactly, we treat it as another instance of the same main argument, and record $e$ as an additional source expert for $m$. Otherwise, we instantiate a new main argument node $m \in \mathcal{A}$ with label $\lambda(m) = u_e$ and add it to $\mathcal{A}_{\text{main}}$. The outcome of this step is (i) a set of main argument nodes $\mathcal{A}_{\text{main}} \subseteq \mathcal{A}$ and (ii) a source-expert mapping $\mathcal{C} : \mathcal{A}_{\text{main}} \to 2^E$, where $\mathcal{C}(m)$ contains the expert (or experts) that proposed the statement $\lambda(m)$. This source-expert mapping is retained for use in subsequent phases of the pipeline. In particular, it supports the enforcement of author/non-author constraints when generating higher-level arguments: experts are prevented from reviewing their own first-level arguments at level 2, while original authors can be selectively re-invited at level 3 to produce targeted rebuttals against attacks.

We intentionally use exact string matching at this stage rather than semantic or embedding-based merging. Since main arguments function as root claims that anchor the downstream support/attack structure, we have empirically observed that aggressively collapsing contextually similar main argument statements tends to reduce the diversity of subsequent generations: once root-level variation is removed, later stages lead to a narrower discussion space and, in turn, a degradation in overall discussion quality. We therefore defer semantic similarity-based filtering to later phases (Sec. 4.2), where it is applied explicitly through contextual orthogonality pruning to encourage diversity among supporting and attacking arguments.

**D.4. Contextual Orthogonality Pruning**

A key design component of ARGORA is *contextual orthogonality pruning*, an argument filtering / selection rule that

prevents the argumentation graph from being dominated by near-duplicate argument statements. In early implementations and empirical evaluations of ARGORA, we found that a purely greedy expansion strategy—adding all newly generated candidate arguments at each depth—often yields an explosive growth of highly overlapping statements, and creates two potential practical issues.

First, it rapidly inflates the effective context length for each expert, which must condition on its accumulated interaction history. As the history grows, relevant details are more likely to be truncated or diluted, degrading the quality of subsequent generation.

Second, oversaturation by contextually similar argument nodes reduces the *effective* diversity of the argumentation framework. Empirically, this can lead to (i) strength saturation under modular semantics updates (many near-identical supports/attacks push strengths toward 0 or 1), and (ii) weak counterfactual signals (removing the edges of one of many redundant nodes produces marginal changes), which in turn degrades the usefulness of our counterfactual explanations.

To address these issues, we prune candidates using a two-stage contextual orthogonality test driven by embedding similarity (as presented in Alg. 2). Intuitively, Stage 1 filters out candidates that are too similar to the already-established context in the *parent node* hierarchy, and Stage 2 enforces diversity *within each parent* by greedily rejecting candidates that are too similar to already chosen siblings. We measure contextual overlap between two statement texts $u, v \in \Sigma^*$ using the cosine similarity of their sentence embeddings:

$$\mathrm{sim}(u, v) := \cos\big(\phi(u), \phi(v)\big) \in [-1, 1],$$

where $\phi : \Sigma^* \to \mathbb{R}^d$ is a sentence encoder (we use `all-MiniLM-L6-v2`). As above (Sec. D.3), $\lambda(a)$ denotes the statement text associated with an argument node $a$.

**Inputs.** Fix a depth level $\ell \geq 1$ and let $\mathcal{A}_{\mathrm{cand}}^{(\ell)}$ be the set of candidate argument nodes proposed for insertion at level $\ell$. Each $a \in \mathcal{A}_{\mathrm{cand}}^{(\ell)}$ has statement text $\lambda(a)$ and a parent identifier $\mathbf{Pa}(a)$ (candidates with the same $\mathbf{Pa}(\cdot)$ are siblings).

For Stage 1, we compare candidates against the *parent-level context*, i.e., the set of already-instantiated argument nodes at level $\ell - 1$. Let $\mathcal{A}^{(\ell)}$ denote the set of argument nodes currently present at level $\ell$; then the parent-level context is $\mathcal{A}^{(\ell-1)}$. We also define $\rho_{\mathrm{sim}}$ as the similarity threshold that controls pruning.

**Stage 1: parent-level orthogonality.** For each candidate $a \in \mathcal{A}_{\mathrm{cand}}^{(\ell)}$, we compute its maximum similarity against the parent-level context $\mathcal{A}^{(\ell-1)}$:

$$\rho_{\mathrm{par}}(a) := \max_{p \in \mathcal{A}^{(\ell-1)}} \mathrm{sim}\big(\lambda(a), \lambda(p)\big).$$

A candidate passes Stage 1 if $\rho_{\mathrm{par}}(a) \leq \rho_{\mathrm{sim}}$. This test prevents adding candidates that are essentially restatements of the existing context at the parent level. A purely hard threshold can occasionally reject *all* candidates. To avoid producing an empty expansion in such cases, we additionally track a fallback candidate

$$a_{\mathrm{fb}} := \arg\min_{a \in \mathcal{A}_{\mathrm{cand}}^{(\ell)}} \rho_{\mathrm{par}}(a),$$

i.e., the least parent-similar candidate. If Stage 1 yields no remaining candidates from pruning, we return $\{a_{\mathrm{fb}}\}$.

**Stage 2: sibling orthogonality.** Among the candidates that pass Stage 1, we greedily construct a diverse subset $\mathcal{A}_{\mathrm{sel}}^{(\ell)}$. We first sort the Stage 1 survivors in ascending order of $\rho_{\mathrm{par}}$, so we consider the most parent-orthogonal candidates first. We then iterate through this ordering and accept a candidate only if it is sufficiently orthogonal to already selected siblings under the same parent:

$$\rho_{\mathrm{sib}}(a) := \max_{a' \in \mathcal{A}_{\mathrm{sel}}^{(\ell)} : \, \mathbf{Pa}(a') = \mathbf{Pa}(a)} \mathrm{sim}\big(\lambda(a), \lambda(a')\big),$$

with the convention $\rho_{\mathrm{sib}}(a) = 0$ if no sibling has been selected yet. We add $a$ to $\mathcal{A}_{\mathrm{sel}}^{(\ell)}$ if $\rho_{\mathrm{sib}}(a) \leq \rho_{\mathrm{sim}}$. This second test prevents the selected set from being dominated by multiple similar argument nodes attached to the same parent.

---

**Algorithm 3** First-Level Argument Generation

---

**Require:** Main-argument set $\mathcal{A}_{\text{main}}$, expert set $E$, source map $\mathcal{C} : \mathcal{A}_{\text{main}} \to 2^E$, topic $t$, task statement $s_M$, key elements $K$, similarity threshold $\rho_{\text{sim}}$, sentence encoder $\phi$

**Ensure:** For each $m \in \mathcal{A}_{\text{main}}$, sets of first-level supporting and attacking arguments

1:  **for** each $m \in \mathcal{A}_{\text{main}}$ **do**
2:      $\tilde{\mathcal{A}}_m^{(1)} \leftarrow \emptyset$ {Level-1 candidate arguments for $m$}
3:      $e_{\text{auth}} \leftarrow$ GETPRIMARYSOURCE$(m, \mathcal{C})$
4:      {**Phase 1: Query experts in parallel**}
5:      **for all** $e \in E$ **in parallel do**
6:          $role \leftarrow$ author if $e = e_{\text{auth}}$ else peer
7:          $(\sigma_e, U_e) \leftarrow$ QUERYFIRSTLEVEL$(e; t, s_M, K, \lambda(m), role)$ {$\sigma_e \in \{$agree, disagree$\}$, $U_e$ is list of reasoning texts, see Prompt I.2.3}
8:          **for** each $u \in U_e$ **do**
9:              $a \leftarrow$ MAKEARG$(u)$
10:             $a.polarity \leftarrow$ support if $\sigma_e =$ agree else attack
11:             $a.expert \leftarrow e$
12:             $\tilde{\mathcal{A}}_m^{(1)} \leftarrow \tilde{\mathcal{A}}_m^{(1)} \cup \{a\}$
13:         **end for**
14:     **end for**
15:     {**Phase 2: Contextual orthogonality pruning**}
16:     $\mathcal{P} \leftarrow \{t\} \cup \{s_M\} \cup K$ {Parent context includes topic, task, and key elements}
17:     $\mathcal{A}_{\text{sel}}^{(1)} \leftarrow$ CONTEXTUALORTHOGONALITYPRUNING$(\tilde{\mathcal{A}}_m^{(1)}, \mathcal{P}, \rho_{\text{sim}}, \phi)$ {See Alg. 2}
18:     {**Phase 3: Build relation sets from selected arguments**}
19:     $R_m^+ \leftarrow \{(a, m) : a \in \mathcal{A}_{\text{sel}}^{(1)} \wedge a.polarity =$ support$\}$
20:     $R_m^- \leftarrow \{(a, m) : a \in \mathcal{A}_{\text{sel}}^{(1)} \wedge a.polarity =$ attack$\}$
21:     $\mathcal{A}_m \leftarrow \{m\} \cup \mathcal{A}_{\text{sel}}^{(1)}$
22: **end for**
23: **return** $\{(\mathcal{A}_m, R_m^+, R_m^-) : m \in \mathcal{A}_{\text{main}}\}$

---

**Threshold behavior.** The parameter $\rho_{\text{sim}}$ controls the diversity–coverage trade-off. Smaller values enforce stricter orthogonality (fewer, more diverse arguments), while larger values admit more semantically overlapping candidates (higher coverage but greater redundancy). In our experiments we set $\rho_{\text{sim}}$ to an empirically chosen default that maintains argument diversity while preserving enough coverage to support downstream modular updates and counterfactual explanation queries. We show the effect of different values for the similarity threshold in our ablation tests in Appendix H.

### D.5. First-Level Argument Generation

After the set of main arguments $\mathcal{A}_{\text{main}}$ is fixed, ARGORA expands each main argument into a localized argumentation structure by generating first-level supporting and attacking arguments. This stage constructs what we call the *direct children* of the main arguments: argument nodes that provide the primary justifications most directly connected to each main claim. In particular, these are the nodes used by the counterfactual explanation query in Def. 4.4 that identifies the most influential direct child of a main argument.

For each main argument $m \in \mathcal{A}_{\text{main}}$, we first designate a *primary source expert* $e_{\text{auth}}$ using the source mapping $\mathcal{C}$ (Alg. 3, line 3). All experts $e \in E$ are then queried *in parallel* to assess the validity of $m$ under the shared topic $t$, task statement $s_M$, and key elements $K$. The primary source expert is queried in an *author* role, while all other experts act as *peers*. Each expert returns (i) a binary stance $\sigma_e \in \{$agree, disagree$\}$ toward $m$, and (ii) a set of free-form reasoning statements justifying that stance (Alg. 3, lines 6–11). Each reasoning statement is instantiated as a candidate argument node and assigned a polarity (support or attack) based on the expert's stance. To prevent redundancy, the resulting candidate set $\tilde{\mathcal{A}}_m^{(1)}$ is filtered using *contextual orthogonality pruning* (Sec. D.4). Finally, the selected arguments are attached to the main argument $m$ via signed relations: supporting arguments induce edges in $R_m^+$, and attacking arguments induce edges in $R_m^-$. The resulting structure $\langle \mathcal{A}_m, R_m^+, R_m^- \rangle$ forms the depth-1 (first-level) expansion of the argumentation graph rooted at $m$.

## D.6. Second-Level Argument Generation

---

**Algorithm 4** Second-Level Argument Generation

---

**Require:** For each $m \in \mathcal{A}_{\text{main}}$: the depth-1 structure $\langle \mathcal{A}_m, R_m^+, R_m^- \rangle$ with first-level nodes $\mathcal{A}_m^{(1)} := \mathcal{A}_m \setminus \{m\}$, expert set
$\quad$ $E$, topic $t$, task $s_M$, key elements $K$, threshold $\rho_{\text{sim}}$, encoder $\phi$
**Ensure:** Updated depth-2 structure for each $m \in \mathcal{A}_{\text{main}}$
$\quad$ 1: **for** each $m \in \mathcal{A}_{\text{main}}$ **do**
$\quad$ 2: $\quad$ $\tilde{\mathcal{A}}_m^{(2)} \leftarrow \emptyset$
$\quad$ 3: $\quad$ {**Phase 1: Assign review sets per expert**}
$\quad$ 4: $\quad$ **for** each $e \in E$ **do**
$\quad$ 5: $\quad\quad$ $\mathcal{R}_e \leftarrow \{p \in \mathcal{A}_m^{(1)} : p.\textit{expert} \neq e\}$ {Nodes not authored by $e$}
$\quad$ 6: $\quad$ **end for**
$\quad$ 7: $\quad$ {**Phase 2: Batch query experts in parallel**}
$\quad$ 8: $\quad$ **for all** $e \in E$ with $\mathcal{R}_e \neq \emptyset$ **in parallel do**
$\quad$ 9: $\quad\quad$ $\mathcal{V}_e \leftarrow$ QUERYSECONDLEVEL$(e; t, s_M, K, \lambda(m), \mathcal{R}_e)$ {Returns list of (node index, stance, justification) tuples,
$\quad\quad\quad$ see Prompt I.2.4}
$\quad$ 10: $\quad\quad$ **for** each $(i, \sigma, u) \in \mathcal{V}_e$ **do**
$\quad$ 11: $\quad\quad\quad$ **if** $\sigma \in \{\text{AGREE}, \text{DISAGREE}\}$ and $u \neq \emptyset$ **then**
$\quad$ 12: $\quad\quad\quad\quad$ Let $p$ be the $i$-th node in $\mathcal{R}_e$
$\quad$ 13: $\quad\quad\quad\quad$ $a \leftarrow$ MAKEARG$(u)$
$\quad$ 14: $\quad\quad\quad\quad$ $a.\textit{expert} \leftarrow e$
$\quad$ 15: $\quad\quad\quad\quad$ $a.\textit{parent} \leftarrow p$
$\quad$ 16: $\quad\quad\quad\quad$ $a.\textit{polarity} \leftarrow$ support if $\sigma = $ AGREE else attack
$\quad$ 17: $\quad\quad\quad\quad$ $\tilde{\mathcal{A}}_m^{(2)} \leftarrow \tilde{\mathcal{A}}_m^{(2)} \cup \{a\}$
$\quad$ 18: $\quad\quad\quad$ **end if**
$\quad$ 19: $\quad\quad$ **end for**
$\quad$ 20: $\quad$ **end for**
$\quad$ 21: $\quad$ {**Phase 3: Contextual orthogonality pruning**}
$\quad$ 22: $\quad$ $\mathcal{P} \leftarrow \{\lambda(p) : p \in \mathcal{A}_m^{(1)}\}$ {Parent pool: first-level statements only}
$\quad$ 23: $\quad$ $\mathcal{A}_{\text{sel}}^{(2)} \leftarrow$ CONTEXTUALORTHOGONALITYPRUNING$(\tilde{\mathcal{A}}_m^{(2)}, \mathcal{P}, \rho_{\text{sim}}, \phi)$ {See Alg. 2}
$\quad$ 24: $\quad$ {**Phase 4: Update graph with selected nodes**}
$\quad$ 25: $\quad$ $\mathcal{A}_m \leftarrow \mathcal{A}_m \cup \mathcal{A}_{\text{sel}}^{(2)}$
$\quad$ 26: $\quad$ $R_m^+ \leftarrow R_m^+ \cup \{(a, a.\textit{parent}) : a \in \mathcal{A}_{\text{sel}}^{(2)} \wedge a.\textit{polarity} = \text{support}\}$
$\quad$ 27: $\quad$ $R_m^- \leftarrow R_m^- \cup \{(a, a.\textit{parent}) : a \in \mathcal{A}_{\text{sel}}^{(2)} \wedge a.\textit{polarity} = \text{attack}\}$
$\quad$ 28: **end for**
$\quad$ 29: **return** $\{(\mathcal{A}_m, R_m^+, R_m^-) : m \in \mathcal{A}_{\text{main}}\}$

---

Given the depth-1 expansion for each main argument $m \in \mathcal{A}_{\text{main}}$, ARGORA further refines the argumentation graph by generating second-level arguments that *support* or *attack* the first-level arguments of $m$. While first-level nodes provide direct justifications for (or against) the main claim, second-level nodes capture *cross-expert review*: experts who did not author a first-level argument assess its validity and provide stance-justified feedback. This stage essentially begins a debate-like procedure by inducing disagreement- or agreement-conditioned responses that attach directly to first-level nodes.

Concretely, for each $m$, we form the first-level set $\mathcal{A}_m^{(1)} = \mathcal{A}_m \setminus \{m\}$ and construct, for each expert $e \in E$, a *review set* $\mathcal{R}_e \subseteq \mathcal{A}_m^{(1)}$ consisting of those first-level nodes not authored by $e$ (Alg. 4, Phase 1). Each expert then reviews its assigned set in a *batched* manner: we issue a single parallel query per expert that conditions on the shared topic $t$, task statement $s_M$, key elements $K$, and the main claim $\lambda(m)$, together with the statements of the nodes in $\mathcal{R}_e$ (Phase 2). The expert returns, for each reviewed node, a stance $\sigma \in \{\text{AGREE}, \text{DISAGREE}\}$ and a free-form justification. Each non-empty justification is instantiated as a second-level argument node $a$, attributed to expert $e$, and attached to the corresponding first-level parent $p \in \mathcal{R}_e$. The edge polarity is determined solely by the stance: AGREE induces a support relation and DISAGREE induces an attack relation.

As in the first-level stage, we apply contextual orthogonality pruning to control redundancy among candidate second-level

arguments. Selected nodes are incorporated into the depth-2 structure by adding them to $\mathcal{A}_m$ and connecting each $a$ to its parent $p$ via signed relations in $R_m^+$ or $R_m^-$ (Phase 4). The resulting $\langle \mathcal{A}_m, R_m^+, R_m^- \rangle$ therefore constitutes the depth-2 (second-level) expansion rooted at $m$, in which first-level claims are explicitly subjected to peer review and structured rebuttal.

## D.7. Third-Level Argument Generation

---

**Algorithm 5** Third-Level Argument Generation

---

**Require:** For each $m \in \mathcal{A}_{\mathrm{main}}$: the current depth-2 structure $\langle \mathcal{A}_m, R_m^+, R_m^- \rangle$ with first-level nodes $\mathcal{A}_m^{(1)}$ and second-level nodes $\mathcal{A}_m^{(2)}$; expert set $E$, source map $\mathcal{C} : \mathcal{A}_{\mathrm{main}} \to 2^E$, topic $t$, task $s_M$, key elements $K$, threshold $\rho_{\mathrm{sim}}$, encoder $\phi$

**Ensure:** Updated depth-3 structure for each $m \in \mathcal{A}_{\mathrm{main}}$ (targeted rebuttals)

1: **for** each $m \in \mathcal{A}_{\mathrm{main}}$ **do**
2:    $\tilde{\mathcal{A}}_m^{(3)} \leftarrow \emptyset$ {Level-3 candidate rebuttals for $m$}
3:    {**Phase 1: Identify second-level attacks per first-level parent**}
4:    $\mathcal{A}_{m,\mathrm{atk}}^{(2)} \leftarrow \{a \in \mathcal{A}_m^{(2)} : a.polarity = \mathtt{attack}\}$
5:    **for** each $p \in \mathcal{A}_m^{(1)}$ **do**
6:       $\mathcal{C}_p \leftarrow \{v \in \mathcal{A}_{m,\mathrm{atk}}^{(2)} : v.parent = p\}$ {Attacks on parent $p$}
7:    **end for**
8:    {**Phase 2: Query original authors for targeted rebuttals (in parallel)**}
9:    **for all** $p \in \mathcal{A}_m^{(1)}$ with $\mathcal{C}_p \neq \emptyset$ **in parallel do**
10:      $e_p \leftarrow p.expert$ {Author of the first-level parent}
11:      $\mathcal{R}_p \leftarrow$ QUERYTHIRDLEVEL$(e_p; t, s_M, K, \lambda(m), \lambda(p), \{(\lambda(v), v.expert) : v \in \mathcal{C}_p\})$ {Returns a list of rebuttal texts aligned to attacks in $\mathcal{C}_p$, see Prompt I.2.5}
12:      **for** each attack $v \in \mathcal{C}_p$ with rebuttal text $u \in \mathcal{R}_p$ **do**
13:        **if** $u \neq \emptyset$ **then**
14:          $a \leftarrow$ MAKEARG$(u)$
15:          $a.expert \leftarrow e_p$
16:          $a.parent \leftarrow v$
17:          $a.polarity \leftarrow \mathtt{attack}$ {Rebuttal attacks the critique}
18:          $\tilde{\mathcal{A}}_m^{(3)} \leftarrow \tilde{\mathcal{A}}_m^{(3)} \cup \{a\}$
19:        **end if**
20:      **end for**
21:    **end for**
22:    {**Phase 3: Contextual orthogonality pruning**}
23:    $\mathcal{P} \leftarrow \{\lambda(a.parent) : a \in \tilde{\mathcal{A}}_m^{(3)}\}$ {Parent pool: second-level attacks that received rebuttals}
24:    $\mathcal{A}_{\mathrm{sel}}^{(3)} \leftarrow$ CONTEXTUALORTHOGONALITYPRUNING$(\tilde{\mathcal{A}}_m^{(3)}, \mathcal{P}, \rho_{\mathrm{sim}}, \phi)$ {See Alg. 2}
25:    {**Phase 4: Update graph with selected rebuttals**}
26:    $\mathcal{A}_m \leftarrow \mathcal{A}_m \cup \mathcal{A}_{\mathrm{sel}}^{(3)}$
27:    $R_m^- \leftarrow R_m^- \cup \{(a, a.parent) : a \in \mathcal{A}_{\mathrm{sel}}^{(3)}\}$
28: **end for**
29: **return** $\{(\mathcal{A}_m, R_m^+, R_m^-) : m \in \mathcal{A}_{\mathrm{main}}\}$

---

The third level introduces a targeted *rebuttal phase* in which the original authors of first-level arguments respond to critiques raised at the second level. This mechanism is designed to preserve authorship accountability while maintaining a structured back-and-forth exchange: first-level claims are peer-reviewed at level 2, and any resulting attacks can be answered directly by the originating author at level 3. Rebuttals are modeled as *attacks* on the critiquing second-level arguments.

Concretely, for each main argument $m \in \mathcal{A}_{\mathrm{main}}$, we consider the current depth-2 structure $\langle \mathcal{A}_m, R_m^+, R_m^- \rangle$ and extract the set of second-level attacks $\mathcal{A}_{m,\mathrm{atk}}^{(2)} := \{a \in \mathcal{A}_m^{(2)} : a.polarity = \mathtt{attack}\}$ (Alg. 5, Phase 1). These attacks are grouped by their first-level parent $p \in \mathcal{A}_m^{(1)}$, yielding for each parent a set $\mathcal{C}_p$ of critiques that directly target $p$. For every parent with at least one critique, we query the original author $e_p := p.expert$ to generate targeted rebuttals (Phase 2). The rebuttal query

conditions on the shared context $(t, s_M, K)$, the main claim $\lambda(m)$, the parent claim $\lambda(p)$, and the set of critiques in $\mathcal{C}_p$ (including critique provenance via the attacking expert identifier). The author returns rebuttal texts aligned to the individual critiques; each non-empty rebuttal is instantiated as a new argument node $a \in \tilde{\mathcal{A}}_m^{(3)}$, attributed to $e_p$, and attached to the corresponding second-level critique $v \in \mathcal{C}_p$ as its parent. All such rebuttal nodes are assigned polarity `attack`, reflecting that they are counter-arguments directed at the critiques. As in earlier stages, we apply contextual orthogonality pruning to avoid redundant rebuttals (Phase 3). The retained rebuttals $\mathcal{A}_{\text{sel}}^{(3)}$ are incorporated into the depth-3 structure by adding them to $\mathcal{A}_m$ and inserting edges into the attack relation set $R_m^-$ from each rebuttal to its parent critique (Phase 4).

### D.8. Base Score Assignment

---
**Algorithm 6** Base Score Assignment

---
**Require:** For each $m \in \mathcal{A}_{\text{main}}$: argument set $\mathcal{A}_m$, topic $t$, main task $s_M$, key elements $K$
**Ensure:** Base score function $w_m : \mathcal{A}_m \to (0, 1)$ for each $m \in \mathcal{A}_{\text{main}}$
1: {**Parallel evaluation across all main arguments and nodes**}
2: **for all** $m \in \mathcal{A}_{\text{main}}$ **in parallel do**
3:    **for all** $a \in \mathcal{A}_m$ **in parallel do**
4:       $(w_m^{\text{task}}(a), w_m^{\text{supp}}(a), w_m^{\text{logi}}(a)) \leftarrow \text{EVALUATECRITERIA}(\lambda(a); t, s_M, K)$ {See Prompt I.1.5}
5:       $w_m(a) \leftarrow \dfrac{w_m^{\text{task}}(a) + w_m^{\text{supp}}(a) + w_m^{\text{logi}}(a)}{3}$
6:    **end for**
7: **end for**
8: **return** $\{w_m : m \in \mathcal{A}_{\text{main}}\}$

---

Before evaluating the constructed argumentation graphs under a quantitative semantics, ARGORA assigns an initial *base score (strength)* $w_m(a) \in (0, 1)$ to each argument node $a \in \mathcal{A}_m$ for every main argument $m \in \mathcal{A}_{\text{main}}$. These base score strengths, collectively denoted $w_m : \mathcal{A}_m \to (0, 1)$, serve as the initial argument valuations in the QBAF $Q_m = \langle \mathcal{A}_m, R_m^-, R_m^+, w_m \rangle$ before propagation through support and attack relations.

**Deviation from standard QBAF base score range.** While the standard QBAF definition (Section 3.1) specifies base scores $w : \mathcal{A} \to [0, 1]$ using the inclusive interval $[0, 1]$, ARGORA deliberately restricts base scores to the open interval $(0, 1)$, excluding the boundary values 0 and 1. This design choice addresses two tendencies observed when using LLMs for argument evaluation: (i) LLMs exhibit a strong bias toward assigning extreme scores, particularly the boundary values 0 and 1, even when such extreme judgments are not warranted by the argument quality, and (ii) LLMs tend to be overly generous in their assessments, frequently assigning scores in the range $[0.8, 1.0)$ regardless of quality differences. Both tendencies weaken the integrity of the resulting argument strength distributions: extreme base scores can dominate the graph evaluation, making the final strengths insensitive to the support and attack structure, while inflated scores compress the dynamic range and reduce the discriminative power of the semantics. To mitigate these issues, ARGORA enforces the strict bounds $(0, 1)$ and explicitly instructs the evaluating LLM to treat 0.5 as the baseline score for arguments that adequately meet evaluation criteria, with most scores expected to fall in the range $[0.30, 0.70]$.

Algorithm 6 describes the assignment of base scores across all argument nodes in all main argument graphs. ARGORA computes base scores through a multi-criterion evaluation process that assesses each argument statement along three dimensions: task relevance, evidence support, and logical soundness. For each argument node $a \in \mathcal{A}_m$, the orchestrator LLM evaluates the statement text $\lambda(a)$ in the context of the discussion topic $t$, main task $s_M$, and key elements $K$, producing three scores:

- **Task Relevance** $w_m^{\text{task}}(a) \in (0, 1)$: The degree to which the statement directly and concretely addresses the discussion topic, main task, and key elements.

- **Evidence Support** $w_m^{\text{supp}}(a) \in (0, 1)$: The extent to which the statement provides reasoning, mechanisms, or evidence for its claims.

- **Logical Soundness** $w_m^{\text{logi}}(a) \in (0, 1)$: Whether the statement is internally coherent, free of contradictions, and follows a reasonable inferential structure.

The final base score for argument $a$ is computed as the arithmetic mean of these three criterion scores:

$$w_m(a) = \frac{w_m^{\text{task}}(a) + w_m^{\text{supp}}(a) + w_m^{\text{logi}}(a)}{3}.$$

We adopt this multi-criterion scoring scheme based on early empirical analyses during the development of ARGORA. During manual inspections of expert-generated arguments, we frequently observed partial outputs that were inadequate in different ways: some responses proposed a concrete decision (e.g., an MCQA option) while providing little to no supporting rationale; others offered plausible discussion but failed to resolve the task explicitly (including occasional invalid option formats for multiple-choice benchmarks). These observations motivate separating task relevance (does the statement directly address $(t, s_M, K)$ and yield an actionable stance) from evidence support (does it provide justification, mechanisms, or evidence rather than an unsupported assertion). We additionally include logical soundness to penalize internally inconsistent or weakly connected reasoning, a tendency that was more pronounced when questions required arithmetic reasoning or tight constraint satisfaction. While human annotation would be preferable for assigning initial argument valuations, such supervision is impractical at the scale and automation level required by an end-to-end multi-LLM discussion pipeline like ARGORA. We therefore delegate criterion evaluation to the Orchestrator, using explicit calibration instructions to reduce score saturation and preserve dynamic range (see Prompt. I.1.5). The subsequent application of a modular quantitative semantics (Def. 3.2) then propagates these calibrated priors through the support/attack structure, yielding final strengths that reflect both intrinsic quality and relational context with other arguments.

### D.9. Complete Summary of ARGORA Discussion Round

---

**Algorithm 7** Complete Argumentation Round in ARGORA

---

**Require:** Expert set $E$, topic $t$, task statement $s_M$, key elements $K$, similarity threshold $\rho_{\text{sim}}$, modular semantics $(\alpha, \iota)$
**Ensure:** A family of evaluated graphs $\{\langle Q_m, \sigma_m \rangle : m \in \mathcal{A}_{\text{main}}\}$
  1: {**Phase 1: Main argument extraction (Alg. 1)**}
  2: $(\mathcal{A}_{\text{main}}, \mathcal{C}) \leftarrow \text{MAINARGEXTRACTION}(r; E, t, s_M, K)$ {See prompt I.2.2}
  3: {**Phase 2: First-level argument generation (Alg. 3)**}
  4: **for all** $m \in \mathcal{A}_{\text{main}}$ **in parallel do**
  5:   $(\mathcal{A}_m, R_m^+, R_m^-) \leftarrow \text{GENFIRSTLEVEL}(m; E, \mathcal{C}, t, s_M, K, \rho_{\text{sim}})$ {See prompt I.2.3}
  6: **end for**
  7: {**Phase 3: Second-level peer reviews (Alg. 4)**}
  8: **for all** $m \in \mathcal{A}_{\text{main}}$ **in parallel do**
  9:   $(\mathcal{A}_m, R_m^+, R_m^-) \leftarrow \text{GENSECONDLEVEL}(m; \mathcal{A}_m, R_m^+, R_m^-, E, t, s_M, K, \rho_{\text{sim}})$ {See prompt I.2.4}
 10: **end for**
 11: {**Phase 4: Third-level targeted rebuttals (Alg. 5)**}
 12: **for all** $m \in \mathcal{A}_{\text{main}}$ **in parallel do**
 13:   $(\mathcal{A}_m, R_m^+, R_m^-) \leftarrow \text{GENTHIRDLEVEL}(m; \mathcal{A}_m, R_m^+, R_m^-, E, t, s_M, K, \rho_{\text{sim}})$ {See prompt I.2.5}
 14: **end for**
 15: {**Phase 5: Base score assignment (Alg. 6)**}
 16: **for all** $m \in \mathcal{A}_{\text{main}}$ **in parallel do**
 17:   $w_m \leftarrow \text{ASSIGNBASESCORES}(\mathcal{A}_m; t, s_M, K)$ {See prompt I.1.5}
 18: **end for**
 19: {**Phase 6: Argument strength computation**}
 20: **for all** $m \in \mathcal{A}_{\text{main}}$ **in parallel do**
 21:   $Q_m \leftarrow \langle \mathcal{A}_m, R_m^-, R_m^+, w_m \rangle$
 22:   $\sigma_m \leftarrow \text{EVALUATEQBAF}(Q_m; \alpha, \iota)$ {Compute strengths under the chosen modular semantics}
 23: **end for**
 24: **return** $\{\langle Q_m, \sigma_m \rangle : m \in \mathcal{A}_{\text{main}}\}$

---

Alg. 7 summarizes the complete orchestration of a single discussion round in ARGORA. A round begins by generating candidate responses from all experts and extracting a set of main arguments $\mathcal{A}_{\text{main}}$ along with their source mapping $\mathcal{C}$. For each main argument $m \in \mathcal{A}_{\text{main}}$, ARGORA then constructs a rooted argumentation graph by (i) generating first-level direct children (supporting and attacking justifications), (ii) expanding the discussion via second-level peer reviews of first-level

nodes, and (iii) adding third-level targeted rebuttals by the original first-level authors against second-level critiques. Once the graph structure is complete, ARGORA assigns base scores to all argument nodes through multi-criterion evaluation, producing the base score function $w_m : \mathcal{A}_m \to (0, 1)$ for each main argument graph. Finally, the resulting QBAFs are evaluated under a chosen quantitative semantics to compute final argument strengths, which are subsequently used for downstream decision-making and explanation.

**D.10. Multi-Round Discussion Support and Capability**

To investigate whether iterative refinement through multiple discussion rounds could improve decision quality, we also implemented multi-round support in ARGORA. The hypothesis was that allowing experts to revise their arguments after observing the initial consensus might lead to better reasoning outcomes. However, empirical evaluation reveals that multi-round discussions provide marginal performance improvements that do not justify the increased inference costs and introduce practical challenges in context management and scalable history utilization (therefore, all of the experiments and evaluations reported in this work are fixed as single-round). This section documents the multi-round architecture for completeness, and multi-round ablation results are presented in Appendix H.3.

**Multi-round architecture.** If more than one round of discussion is enabled, the Orchestrator collects all discussion artifacts from the current round $r$ and orchestrates the transition to the next round through a structured prompt generation process. At the conclusion of round $r$, ARGORA aggregates the following discussion artifacts:

- **Expert responses**: Raw textual responses from each expert $e \in E$ and their parsed structural representations.

- **Main arguments**: The set of main arguments $\mathcal{A}_{\text{main}}^{(r)}$ along with their canonical mappings $\mathcal{C}^{(r)}$ from Alg. 1.

- **Argumentation graphs**: For each main argument $m \in \mathcal{A}_{\text{main}}^{(r)}$, the complete evaluated QBAF $\langle Q_m^{(r)}, \sigma_m^{(r)} \rangle$ including all argument nodes $\mathcal{A}_m^{(r)}$, support/attack relations $(R_m^+)^{(r)}$, $(R_m^-)^{(r)}$, base scores $w_m^{(r)}$, and final strengths $\sigma_m^{(r)}$.

These artifacts collectively form the *discussion history* $\mathcal{H}^{(r)}$ for round $r$. However, the mechanism by which this history influences round $r + 1$ operates *indirectly* through expert conversation continuity rather than explicit orchestrator-mediated summarization.

For subsequent rounds, the Orchestrator generates expert prompts that condition only on the task specification $(s_M, K)$, without explicit access to the discussion history $\mathcal{H}^{(r)}$ (see Prompt I.1.4). Instead, *each expert $e \in E$ maintains a persistent conversation history* with the Orchestrator through the message bus. When expert $e$ is prompted for round $r + 1$, the complete context available to $e$ includes:

1. The expert's domain expertise system prompt (unchanging across rounds).

2. All Orchestrator prompts from rounds $1, \ldots, r$.

3. All of expert $e$'s own responses from rounds $1, \ldots, r$.

4. The Orchestrator's new prompt for round $r + 1$.

**Limitations and future directions.** While the multi-round architecture is fully implemented, effective iterative argumentation remains an open challenge. Key issues include: (i) *context window saturation*, where accumulated history dilutes critical information and degrades generation quality; and (ii) *computational cost scaling*, where marginal accuracy gains do not justify linear increases in inference cost. Addressing these challenges represents a promising direction for future work.

# E. Addendum to Counterfactual Explanations in ARGORA.

In this section, we describe additional methods that build on the definitions provided in the main paper (Sections 4.3 and 4.4) to formalize overarching counterfactual explanations.

## E.1. Winner-change counterfactuals via winner-critical interventions

The three counterfactual explanation queries in Section 4.3 (Def. 4.4) characterize *within-graph* causal influence for a fixed main argument $m$ by measuring how edge-local deletions affect the root strength $\sigma(m)$. However, the baseline decision in ARGORA is obtained by comparing root strengths *across* all main arguments (Def. 4.5). Accordingly, the natural question that we would like to ask, *"why did $m^\star$ become the winning main argument?"* can be obtained by verifying which structural perturbations would have changed the identity of the winning argument among $\mathcal{A}^{\mathrm{main}}$.

We formalize this by considering *singleton interventions*—edge-local deletions that remove exactly one edge in exactly one QBAF—and identifying those that would have changed the winner.

**Tie-breaking convention.** Throughout this section, we assume that ties in all $\arg\max$ operations (Defs. 4.5 and 4.6) are broken by a fixed deterministic rule (e.g., argument identifier order). This ensures that the baseline winner $m^\star$ and any intervened winner are uniquely defined.

**Definition E.1** (Singleton intervention). A *singleton intervention* is a pair $(m, x)$ where $m \in \mathcal{A}^{\mathrm{main}}$ is a main argument and $x \in \mathcal{A}_m \setminus \{m\}$ is a non-root node in its QBAF. Applying $(m, x)$ deletes the unique outgoing edge $(x \to \mathbf{Pa}(x))$ in $Q_m$ and leaves all other QBAFs $Q_{m'}$ for $m' \neq m$ unchanged.

We write $\sigma^{(m,x)}$ for the resulting strength function on $Q_m$ after the intervention, and $\sigma^{(m,x)}(m)$ for the intervened root strength.

*Remark* E.2 (Effect localization). Since each $Q_m$ is an independent rooted tree, the structural equations in $\mathcal{M}_m$ depend only on nodes within $\mathcal{A}_m$. Thus, for any singleton intervention $(m, x)$:

$$\sigma^{(m,x)}(m) \neq \sigma(m) \quad \text{in general, but} \quad \sigma(m') \text{ is unchanged for all } m' \in \mathcal{A}^{\mathrm{main}} \setminus \{m\}.$$

Consequently, deciding whether the *winner* changes under $(m, x)$ requires re-evaluating only the single modified tree and then re-taking the $\arg\max$ over $\mathcal{A}^{\mathrm{main}}$ using the updated value $\sigma^{(m,x)}(m)$.

**Definition E.3** (Winner-critical intervention). Let $m^\star$ be the baseline consensus winner (Def. 4.5). A singleton intervention $(m, x)$ is *winner-critical* if applying it changes the identity of the winner.

Formally, let $m^{(m,x)}$ denote the winner after applying $(m, x)$:

$$m^{(m,x)} := \underset{m' \in \mathcal{A}^{\mathrm{main}}}{\arg\max}\, \tilde{\sigma}(m'), \quad \text{where} \quad \tilde{\sigma}(m') := \begin{cases} \sigma^{(m,x)}(m) & \text{if } m' = m, \\ \sigma(m') & \text{otherwise.} \end{cases}$$

Then $(m, x)$ is winner-critical if and only if $m^{(m,x)} \neq m^\star$.

*Remark* E.4 (Causal status of winner-critical interventions). The edge-local deletion underlying a singleton intervention $(m, x)$ is a valid soft intervention within the SCM $\mathcal{M}_m$ induced by Proposition 4.1. The winner-change $m^{(m,x)} \neq m^\star$ is a deterministic downstream consequence of this intervention, propagated through the $\arg\max$ selection rule. If one wishes to treat winner-change as a first-class causal quantity, one can define a *decision-level SCM* $\mathcal{M}_{\mathrm{dec}}$ with endogenous variables $V_{\mathrm{dec}} := \bigcup_{m \in \mathcal{A}^{\mathrm{main}}} V_m \cup \{v_{\mathrm{winner}}\}$ and structural assignment $v_{\mathrm{winner}} = \arg\max_{m \in \mathcal{A}^{\mathrm{main}}} v_m^{\mathrm{root}}$. Under this construction, winner-critical interventions become standard soft interventions in $\mathcal{M}_{\mathrm{dec}}$.

*Remark* E.5 (Existence of winner-critical interventions). A winner-critical singleton intervention need not always exist. If the baseline winner $m^\star$ dominates all competitors by a margin that exceeds the maximum possible single-edge impact, no single deletion will flip the outcome. In such cases, the *absence* of winner-critical interventions can itself be interpreted as evidence of decision robustness: the chosen main argument $m^\star$ is structurally stable under all single-edge perturbations.

**Interpretation.** Winner-critical interventions directly answer the decision-level causal query *"which single argumentative channel, if removed, would have flipped the chosen main argument?"* If the intervention $(m^\star, x)$ targets the baseline winner's own QBAF, then winner-criticality indicates that the edge $(x \to \mathbf{Pa}(x))$ is a *necessary* supportive or defensive channel for $m^\star$ to prevail. If instead the intervention $(m, x)$ targets a competing QBAF with $m \neq m^\star$, then winner-criticality indicates that removing this edge changes the competitor's evaluation enough for it to overtake $m^\star$.

**Minimal winner-change explanations.** Among all singleton interventions, we can summarize "why $m^\star$ won" by reporting the *smallest-perturbation winner flip*. Let

$$\mathcal{S} := \{(m, x) : m \in \mathcal{A}^{\mathrm{main}}, x \in \mathcal{A}_m \setminus \{m\}\}$$

denote the set of all singleton interventions. Using the intervention cost $C(\cdot)$ from Def. 4.6, we select any minimizer

$$(\hat{m}, \hat{x}) \in \arg\min_{(m,x)\in\mathcal{S}} C(m, x) \quad \text{s.t.} \quad m^{(m,x)} \neq m^\star,$$

when such a winner-critical intervention exists (cf. Remark E.5). The edge $(\hat{x} \to \mathbf{Pa}(\hat{x}))$ in $Q_{\hat{m}}$ can be interpreted as the most decision-relevant structural dependency: it is the single argumentative channel whose removal flips the winner with minimal perturbation to the internal state.

### E.2. Comparative causal explanations via margin decomposition

Winner-critical interventions identify *which* edges are necessary for the current winner to prevail, but they do not directly explain *why* a main argument with an initially lower base score might nonetheless win. To complete the explanatory picture, we introduce a margin decomposition that separates the contributions of base scores from the cumulative effects of argumentation.

**Definition E.6** (Net argumentative lift). For each main argument $m \in \mathcal{A}^{\mathrm{main}}$, let $w(m)$ denote its base score (assigned by the Orchestrator) and $\sigma(m)$ its final evaluated strength under the chosen modular semantics. The *net argumentative lift* of $m$ is

$$\Delta_{\mathrm{lift}}(m) := \sigma(m) - w(m).$$

This quantity measures the cumulative effect of all supporting and attacking arguments on $m$'s final strength relative to its base score (which we call *prior*):

- $\Delta_{\mathrm{lift}}(m) > 0$ indicates that the argumentation structure *strengthened* $m$ beyond its prior (net support dominates);

- $\Delta_{\mathrm{lift}}(m) < 0$ indicates that the argumentation structure *weakened* $m$ below its prior (net attack dominates);

- $\Delta_{\mathrm{lift}}(m) = 0$ indicates that supporting and attacking influences exactly cancel.

**Definition E.7** (Pairwise margin decomposition). Let $m^\star$ be the baseline winner. For each competitor $m_j \in \mathcal{A}^{\mathrm{main}} \setminus \{m^\star\}$, the *pairwise winning margin* $\sigma(m^\star) - \sigma(m_j)$ decomposes as:

$$\underbrace{\sigma(m^\star) - \sigma(m_j)}_{\text{final margin}_j} = \underbrace{w(m^\star) - w(m_j)}_{\text{prior margin}_j} + \underbrace{\Delta_{\mathrm{lift}}(m^\star) - \Delta_{\mathrm{lift}}(m_j)}_{\text{argumentative margin}_j}.$$

This yields a margin decomposition for each competitor, which allows us to analyze how $m^\star$ prevailed against competitive main arguments.

**Definition E.8** (Pairwise victory type). For each competitor $m_j \in \mathcal{A}^{\mathrm{main}} \setminus \{m^\star\}$, we classify the pairwise victory of $m^\star$ over $m_j$ into one of three types based on the signs of the prior margin and argumentative margin:

1. **Prior-dominated**: prior margin$_j \geq 0$ and argumentative margin$_j \geq 0$. The winner led in prior and argumentation maintained or extended this lead.

2. **Argumentation-reversed**: prior margin$_j < 0$ and final margin$_j > 0$. The winner overcame a prior deficit through stronger argumentation.

3. **Argumentation-eroded**: prior margin$_j > 0$ and argumentative margin$_j < 0$. The winner's prior lead was narrowed by argumentation.

**Definition E.9** (Aggregate robustness). Let $m^\star$ be the baseline winner. The *minimum final margin* is

$$\Delta_{\min} := \min_{m_j \in \mathcal{A}^{\mathrm{main}} \setminus \{m^\star\}} \big( \sigma(m^\star) - \sigma(m_j) \big),$$

and the *closest competitor* is any $m_j$ achieving this minimum. The larger the value of $\Delta_{\min}$, the more robust our winner is.

**Combining margin decomposition with winner-critical interventions.** The pairwise margin decomposition provides a *global* accounting of why the winner prevailed against each competitor, while winner-critical interventions provide *local* identification of necessary edges. Together, they form a complete mechanistic explanation:

1. **Compute pairwise decompositions** for all competitors to determine the victory type (Def. E.8) for each.

2. **Identify aggregate robustness** via $\Delta_{\min}$ and the closest competitor.

3. **Enumerate winner-critical interventions** to identify which specific edges in which QBAFs are necessary for the current outcome.

4. **Cross-reference**: For each aggregate-reversed victory over $m_j$, the winner-critical edges in $Q_{m^\star}$ represent argumentative channels that enabled $m^\star$ to overcome its prior deficit against $m_j$. Conversely, winner-critical edges in $Q_{m_j}$ represent attacks or weak supports that prevented $m_j$ from capitalizing on its prior advantage.

**Example.** Consider four main arguments $m_1, m_2, m_3, m_4 \in \mathcal{A}^{\mathrm{main}}$ with:

| | $w(m)$ | $\sigma(m)$ | $\Delta_{\mathrm{lift}}(m)$ | Structure |
|---|---|---|---|---|
| $m_1$ | 0.70 | 0.68 | $-0.02$ | slightly attacked |
| $m_2^\star$ | 0.55 | 0.71 | $+0.16$ | strongly supported |
| $m_3$ | 0.72 | 0.69 | $-0.03$ | slightly attacked |
| $m_4$ | 0.45 | 0.52 | $+0.07$ | moderately supported |

The winner is $m_2^\star$ with $\sigma(m_2^\star) = 0.71$. The pairwise decompositions are:

| vs. | Prior | Arg. | Final | Type |
|---|---|---|---|---|
| $m_1$ | $-0.15$ | $+0.18$ | $+0.03$ | ARGREV |
| $m_3$ | $-0.17$ | $+0.19$ | $+0.02$ | ARGREV |
| $m_4$ | $+0.10$ | $+0.09$ | $+0.19$ | PRIORDOM |

**Interpretation.** $m_2^\star$ won against 3 competitors despite having the second-lowest prior. It overcame prior deficits against both $m_1$ and $m_3$ through substantially stronger argumentation ($+0.18$ and $+0.19$ argumentative margins, respectively). However, the victory is FRAGILE: the margin over $m_3$ is only 0.02, meaning small perturbations to either QBAF could flip the outcome.

Winner-critical interventions would identify:

- In $Q_{m_2^\star}$: key support edges whose removal drops $\sigma(m_2^\star)$ below 0.69 (the closest competitor's strength);

- In $Q_{m_3}$: attack edges whose removal raises $\sigma(m_3)$ above 0.71.

Together, these explain *why $m_2^\star$ won* (argumentation overcame prior deficits against multiple competitors) and *which specific edges* are necessary for this outcome, while the FRAGILE robustness classification warns that the decision is sensitive to structural perturbations.

**Relation to within-graph counterfactuals.** For any main argument $m$, the net argumentative lift is

$$\Delta_{\mathrm{lift}}(m) := \sigma(m) - w(m).$$

To attribute which internal nodes are most responsible for this lift, we use the within-graph edge-local deletion counterfactual from Def. 4.3:

$$\Delta_{\mathrm{edge}}(x; m) := \sigma(m) - \sigma^{\ominus x}(m),$$

which answers "how much would $\sigma(m)$ change if node $x$ were removed?"

Since the root strength is obtained by recursively composing the aggregation and influence maps (Eq. (1)),

$$\sigma(a) = \iota_{w(a)}\big(\alpha_{\pi(a)}\big(\sigma(c_1), \ldots, \sigma(c_n)\big)\big),$$

the mapping from the set of internal nodes to $\sigma(m)$ is generally not additive. In particular, for the modular instantiations used in this work (Table 4), non-additivity can arise because the update rule is typically *nonlinear or piecewise-linear*. Consequently, the effect of removing a node depends on what other nodes remain in the graph. Concretely, for distinct nodes $x, x'$ it may hold that

$$\Delta_{\mathrm{edge}}(x;\, m) + \Delta_{\mathrm{edge}}(x';\, m) \;\neq\; \sigma(m) - \sigma^{\ominus\{x,x'\}}(m),$$

so the leave-one-out deletion impacts are generally *not* an additive decomposition of any global quantity. In particular, $\sum_{x \in \mathcal{A}_m \setminus \{m\}} \Delta_{\mathrm{edge}}(x;\, m)$ need not equal $\Delta_{\mathrm{lift}}(m)$, and rankings induced by $\Delta_{\mathrm{edge}}$ need not coincide with any additive "marginal contribution" allocation. Accordingly, Def. 4.4 uses $|\Delta_{\mathrm{edge}}(x;\, m)|$ as a counterfactual importance score: nodes with larger magnitude are those whose removal causes a larger change in the realized root strength.

*Remark* E.10 (Scope of causal explanations). The counterfactual queries defined in this work—within-graph edge-local impacts (Def. 4.3), winner-critical interventions (Def. E.3), and pairwise margin decomposition (Def. E.7)—collectively address the primary explanatory questions for ARGORA's decision process:

1. Which arguments causally influence each main argument's strength?

2. Which argumentative channels are necessary for the winner to prevail?

3. Did the outcome depend more on prior assessments or on argumentation, and does this vary across competitors?

4. How robust is the winner's victory across the full field?

A formal characterization of completeness—showing that these queries exhaust all meaningful causal questions about the decision—would require specifying the precise class of admissible queries, which we leave to future work. We note, however, that single-edge interventions suffice for any query of the form "was argument $a$ (and its edge $e$) necessary for outcome $o$?" by the modularity of the underlying SCM.

# F. Consensus Report in ARGORA

ARGORA automatically generates a technical consensus report in Markdown format upon completion of the discussion and evaluation pipeline. The report serves as a reference for checking all quantities available in the ARGORA framework: all quantities are computed from internal data structures (without additional LLM calls) and each entry is traceable to a formal definition, equation, or procedure introduced in the paper. The report template and generation code are included in the released implementation (Appendix 4.5). Table 5 provides a compact index of the report entries and pointers to the corresponding locations in the paper.

# G. Extended Evaluation

This appendix supplements the main evaluation (Sec. 5) with additional experimental details, hyperparameter settings, and statistical analyses that support our claims. We summarize the fixed evaluation settings used throughout the main experiments, provide the full paired significance-testing procedure for Net Reversal Efficiency (NRE) using the one-sided exact McNemar test, and report complete per-benchmark significance results and diagnostics. We additionally present targeted ablations of the observation-aligned override mechanism—including JS-divergence diagnostics, intervention-cost proxy comparisons, and the effect of the winner-confidence gate—to characterize when and how external grounding improves reliability.

We further include an extended evaluation with a stronger backbone model (gpt-5-mini) to assess whether the reversal-pair patterns and task-dependent utility profile persist under higher base capability, alongside additional analyses relevant to hallucination risk reduction through contestation, quantitative semantics, and causal intervention.

### G.1. Parameter Values Used for Evaluation

Table 6 summarizes the fixed hyperparameter settings used in the main evaluation. These parameters govern the amount of discussion ($R$), pruning behavior via similarity-based contextual orthogonality ($\lambda_{\mathrm{sim}}$), and the sensitivity of the observation-aligned override mechanism via its tradeoff parameter ($\lambda$) and winner-confidence gate threshold ($\tau$). Unless explicitly stated otherwise, the same values are used across all benchmarks and base model variants reported in the main paper.

*Table 5.* Compact index of the Markdown technical consensus report emitted by ARGORA. The **Reference** column points to where each report entry is defined or discussed in the paper.

| Number | Entry | Short description | Reference |
|---|---|---|---|
| **Section 0: Configuration and Metadata** | | | |
| 0.1 | Task specification | User topic $t$, extracted main task $s_M$, and key elements $K$. | Sec. 4 |
| 0.2 | Framework configuration | Experts, rounds, base model, semantics choice, pruning/override settings. | Table 4, Sec. 4.2, Def. 4.6 |
| 0.3 | Execution summary | Counts of main arguments, nodes, and support/attack edges. | Sec. 4.1 |
| 0.4 | Evaluation summary | $m^\star$, $m^{\text{obs}}$, parsed answers, override flag, and ID-to-answer mapping. | Def. 4.5, Def. 4.6 |
| **Section 1: Main Argument Enumeration** | | | |
| 1.1 | All main arguments (with metadata) | Verbatim listing of each $m \in \mathcal{A}^{\text{main}}$ with identifier; includes source expert, round index, and parsed answer (if applicable). | Sec. 4.1 |
| **Section 2: QBAF Evaluation** | | | |
| 2.1 | Base scores and criteria ($w(m)$) | Criterion scores and base score $w(m)$ per main argument; may include criterion-level rationale text (if recorded). | Def. 3.2, Sec. D.8 |
| 2.2 | Per-main QBAF structure | Rooted-tree QBAF visualization with edge polarity and node metadata. | Def. C.3, Sec. 4.1 |
| 2.3 | Final strengths | Root strengths $\sigma(m)$ under the chosen modular semantics. | Def. 3.2 |
| 2.4 | QBAF consensus distribution | Normalized consensus $p_{\text{QBAF}}$ over $\mathcal{A}^{\text{main}}$. | Def. 4.5 |
| 2.5 | Winner and margin | Winner $m^\star$ and its margin to the closest competitor. | Def. 4.5 |
| 2.6 | Winner margin decomposition | Net lift, pairwise margin decomposition, victory type, and robustness summary. | App. E.2, Def. E.6–E.9 |
| **Section 3: Counterfactual Analysis** | | | |
| 3.1 | Most influential direct child | Edge-local impacts for direct children of the winning root. | Def. 4.2, Def. 4.4 |
| 3.2 | Most decisive argument chain | Leaf-to-root chain with the largest edge-local impact and interpretation. | Def. 4.2, Def. 4.4 |
| 3.3 | Most influential overall node | Global ranking of within-$Q_{m^\star}$ nodes by impact magnitude. | Def. 4.2, Def. 4.4 |
| 3.4 | Winner-critical interventions | Summary metrics, list of winner-changing singleton edge deletions, and brief mechanism notes (if any exist). | Def. E.3 |
| **Section 4: Observational Override Analysis** | | | |
| 4.1 | Observational distribution | Judge scores and normalized observational consensus $p_{\text{obs}}$. | Def. 4.5 |
| 4.2 | Observational winner mapping | Parsed answer corresponding to $m^{\text{obs}}$ (when applicable). | Def. 4.5 |
| 4.3 | Distribution comparison | Per-argument differences and distributional divergence summary. | Def. 4.6 |
| 4.4 | Override decision | Whether override triggers and the reason (agreement vs. disagreement). | Def. 4.6 |
| 4.5 | Override search details | Candidate interventions and the selected $\hat{\mathcal{I}}$ (only if triggered). | Def. 4.6 |
| **Section 5: Aggregation Method Comparison** | | | |
| 5.1 | Method comparison table | Side-by-side winners from QBAF evaluation and the observational judge. | Def. 4.5 |
| 5.2 | Winning answer analysis | Agreement indicator and, when available, comparison to ground truth. | Def. 4.5 |
| **Section 6: Final Decision Summary** | | | |
| 6 | Final decision block | Consolidated decision with key evidence and robustness indicators. | Def. 4.5, Def. E.3 |

*Table 6.* Evaluation settings used throughout the main experiments.

| Symbol | Meaning | Value | Reference |
|---|---|---|---|
| $\rho_{\mathrm{sim}}$ | contextual orthogonality similarity threshold | 0.7 | Sec. 4.2 |
| $\lambda$ | observation-aligned tradeoff parameter | 0.05 | Def. 4.6 |
| $\tau$ | winner-confidence gate threshold | 0 | Def. 4.6 |

### G.2. Significance Testing via McNemar's Test

This section provides the full specification of the paired significance tests referenced in Sec. 5.1. Our evaluation compares paired outcomes on the same instances: the prior winner versus the final winner (for NRE). For each instance in the disagreement set $\mathcal{N}_{\mathrm{disagree}}$, we observe one of four correctness transitions: correct→correct, wrong→correct, correct→wrong, or wrong→wrong. Our key question is: *"When the mechanism changes the winner's correctness status, does it fix errors more often than it introduces them?"*

To answer this, we focus only on *reversal pairs*: instances where correctness differs between the two paired conditions. Instances that remain correct or remain wrong under both conditions are informative about base difficulty, but they do not identify the mechanism's directional effect. This motivates McNemar's test, which conditions on reversal pairs and tests whether positive reversals systematically exceed negative reversals.

**Exact McNemar test (one-sided).** Our hypothesis for NRE is *directional*: we seek evidence that ARGORA's corrective argumentation fixes errors more often than it introduces them. Accordingly, we use the one-sided exact McNemar test with alternative

$$H_1 : \ n_{-\rightarrow+} > n_{+\rightarrow-},$$

where $n_{-\rightarrow+}$ counts *positive reversals* (incorrect → correct) and $n_{+\rightarrow-}$ counts *negative reversals* (correct → incorrect) under a paired comparison on the same instances. The null hypothesis is symmetry of reversal directions,

$$H_0 : \ n_{-\rightarrow+} = n_{+\rightarrow-},$$

which implies that, conditional on a reversal occurring, each direction is equally likely. Let $T := n_{-\rightarrow+} + n_{+\rightarrow-}$ denote the total number of reversal pairs. Under $H_0$, the exact McNemar test treats

$$n_{-\rightarrow+} \sim \mathrm{Binomial}\left(T, \tfrac{1}{2}\right).$$

We report the one-sided exact $p$-value (upper tail) corresponding to $H_1$:

$$p = \sum_{k=n_{-\rightarrow+}}^{T} \binom{T}{k} 2^{-T}. \tag{4}$$

If $T = 0$ (no reversal pairs), we set $p := 1$.

**McNemar test for Net Reversal Efficiency (NRE).** We recall the positive reversal and negative reversal counts from Sec. 5.1, as follows:

$$n_{-\rightarrow+} = \sum_{i\in\mathcal{N}_{\mathrm{disagree}}} \mathbb{I}\big[L(m_i^{(0)}) \neq y_i^* \ \wedge \ L(m_i^*) = y_i^*\big], \qquad n_{+\rightarrow-} = \sum_{i\in\mathcal{N}_{\mathrm{disagree}}} \mathbb{I}\big[L(m_i^{(0)}) = y_i^* \ \wedge \ L(m_i^*) \neq y_i^*\big].$$

We then compute the one-sided exact McNemar $p$-value for NRE by applying (4) with $T = n_{-\rightarrow+} + n_{+\rightarrow-}$, and denote the resulting value by $p_{\mathrm{NRE}}$. We interpret statistical support for improvement only when the corresponding net metric is nonnegative (NRE $\geq 0$).

### G.3. NRE Significance Test Results and Discussion

Table 7 reports the complete significance test results for **Argumentative Utility** (as referenced in Sec. 5.1). We discuss three findings: (1) the relationship between effect size and statistical power, (2) task-dependent patterns of argumentative efficacy, and (3) the role of semantic choice in shaping both accuracy and paired statistical evidence.

*Table 7.* Paired significance tests for Argumentative Utility under ARGORA (*pre-override*). For each dataset, we report the pre-override accuracy (Acc), Net Reversal Efficiency (NRE), disagreement-conditioned positive and negative reversal counts ($n_{-\to+}$, $n_{+\to-}$), and the exact McNemar $p$-value for NRE ($p_{\mathrm{NRE}}$) as defined in Appendix G. Results are shown for DF-QuAD and the best-performing semantics ("Best"). We indicate $p_{\mathrm{NRE}}$ with N/A if the NRE value is negative, as our one-sided $p$-value test is only interpretable when the metric is nonnegative.

| Dataset | $N$ | $|\mathcal{N}_{\mathrm{disagree}}|$ | ARGORA (*pre-override*): DF-QuAD | | | | | ARGORA (*pre-override*): Best | | | | |
|---|---|---|---|---|---|---|---|---|---|---|---|---|
| | | | Acc | $n_{-\to+}$ | $n_{+\to-}$ | NRE | $p_{\mathrm{NRE}}$ | Acc | $n_{-\to+}$ | $n_{+\to-}$ | NRE | $p_{\mathrm{NRE}}$ |
| MMLU-Pro | 500 | 180 | 0.638 | 20 | 11 | 0.050 | **0.074** | 0.640 | 19 | 8 | 0.061 | **0.026** |
| MedQA | 500 | 86 | 0.812 | 14 | 5 | 0.105 | **0.032** | | ≡ | | | **0.032** |
| TruthfulQA | 500 | 61 | 0.882 | 8 | 4 | 0.066 | **0.194** | 0.886 | 9 | 3 | 0.098 | **0.073** |
| GPQA Diamond | *198* | 106 | 0.450 | 12 | 10 | 0.019 | **0.416** | 0.480 | 15 | 6 | 0.085 | **0.039** |
| MuSR (Murder Mystery) | *250* | 57 | 0.664 | 14 | 7 | 0.123 | **0.095** | | ≡ | | | **0.095** |
| MuSR (Object Placement) | *256* | 75 | 0.523 | 7 | 11 | −0.053 | N/A | 0.532 | 8 | 8 | 0.000 | **0.598** |
| MuSR (Team Allocation) | *250* | 70 | 0.564 | 9 | 8 | 0.014 | **0.500** | | ≡ | | | **0.500** |

**Effect Size vs. Statistical Power: The Reversal-Pair Bottleneck.** A central observation from Table 7 is that large Net Reversal Efficiency (NRE) values do not always correspond to conventional thresholds such as $p_{\mathrm{NRE}} < 0.05$. This reflects a basic constraint of McNemar testing: statistical power depends on the number of *reversal pairs* $T$, not on the overall evaluation size.

Consider TruthfulQA under the best-performing semantics: ARGORA achieves NRE $= +0.098$ with $n_{-\to+} = 9$ and $n_{+\to-} = 3$, yielding $p_{\mathrm{NRE}} = 0.073$. Although this falls short of 0.05, the directionality is clear: among the $T = 12$ reversal pairs, ARGORA fixes errors three times as often as it introduces them. It is also worth noting that $T$ is small relative to the disagreement set (12 out of $|\mathcal{N}_{\mathrm{disagree}}| = 61$), which limits power and yields coarse $p$-value resolution; for example, with $T = 12$, a 10:2 split would be required to reach $p_{\mathrm{NRE}} < 0.05$ under the one-sided exact test.

This pattern appears across benchmarks. On GPQA Diamond (best semantics), NRE $= +0.085$ with $n_{-\to+} = 15$ and $n_{+\to-} = 6$ ($T = 21$) yields $p_{\mathrm{NRE}} = 0.039$. On the MuSR dataset's Murder Mystery subtask, NRE $= +0.123$ with $n_{-\to+} = 14$ and $n_{+\to-} = 7$ ($T = 21$) yields $p_{\mathrm{NRE}} = 0.095$. In both cases, positive reversals outnumber negative reversals by at least a 2:1 ratio, indicating systematic corrective behavior even when formal significance tests might be interpreted as marginal.

Finally, the scarcity of reversal pairs is itself informative, but should be interpreted cautiously: it suggests that the mechanism flips correctness status relatively rarely, and that its aggregate benefit is driven by a limited set of decisive instances rather than frequent overturning.

**Task-Dependency: Where Argumentation Adds Value.** The paired tests suggest that argumentative utility depends on task structure, consistent with the accuracy patterns in Table 1 and the reversal statistics in Table 7. For example, three benchmarks show statistically significant positive NRE ($p_{\mathrm{NRE}} < 0.05$) under at least one semantics:

- **MMLU-Pro (Best - Euler-based):** $p_{\mathrm{NRE}} = 0.026$, NRE $= +0.061$. This aligns with a $+2.2\%$ absolute accuracy improvement in the main results ($61.8\% \to 64.0\%$), suggesting that the argumentation process helps resolve cross-domain disputes toward correct answers.

- **MedQA (DF-QuAD):** $p_{\mathrm{NRE}} = 0.032$, NRE $= +0.105$. Despite a seemingly equivalent result between the baseline majority vote method and ARGORA ($81.2\%$ vs. $81.2\%$), the positive NRE still indicates that *conditional on a correctness flip*, the mechanism tends to correct more than it harms. A plausible explanation is a near-ceiling regime: when base accuracy is already high, a small number of harmful flips can outweigh beneficial flips in the aggregate even if the reversal direction is favorable on average.

- **GPQA Diamond (Best - Quadratic Energy):** $p_{\mathrm{NRE}} = 0.039$, NRE $= +0.085$. Graduate-level science reasoning appears to benefit from structured debate, aligning with accuracy gain in the main results ($45.9\% \to 48.0\%$).

Two benchmarks show positive NRE with $0.05 \leq p_{\mathrm{NRE}} < 0.1$:

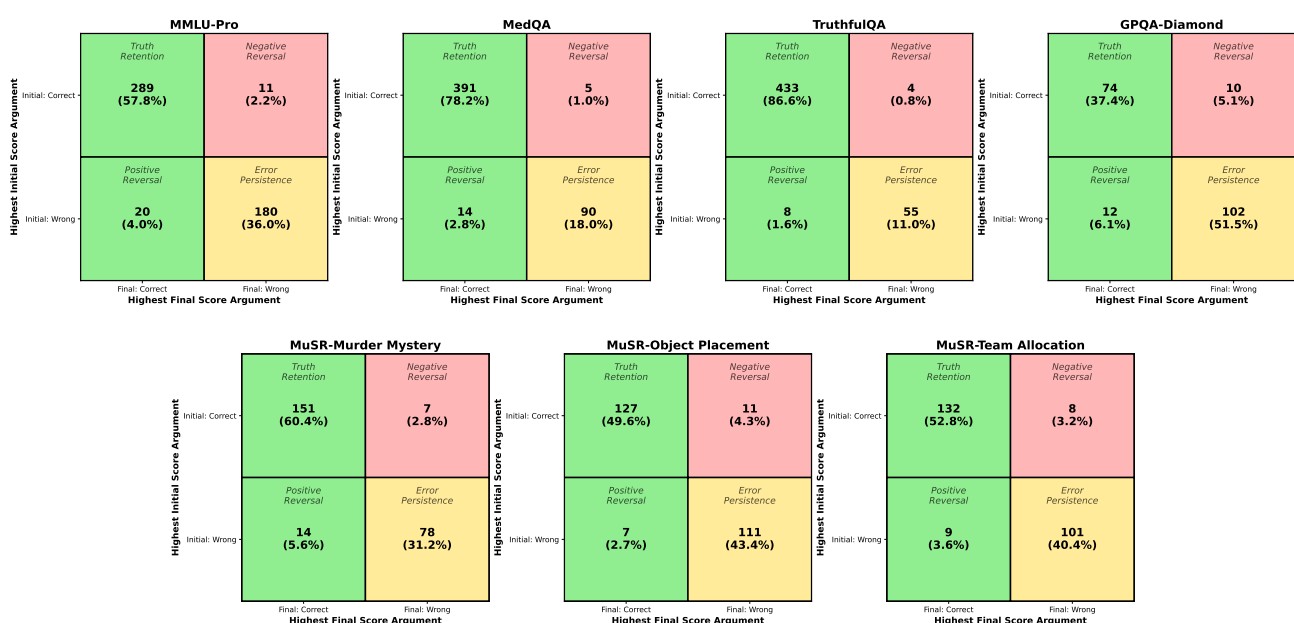

*Figure 4.* The four transition counts: $n_{+\to+}$ (truth retention), $n_{-\to+}$ (positive reversal), $n_{+\to-}$ (negative reversal), $n_{-\to-}$ (error persistence), as defined in Section 5.1 visualized as a confusion matrix for each of the evaluation benchmarks tested on the `gpt-4o-mini` model, with DF-QuAD used as our choice of quantitative semantics. We highlight the truth retention and positive reversal counts as green (good outcome), the error persistence as yellow, and the negative reversal as red (unwanted outcome).

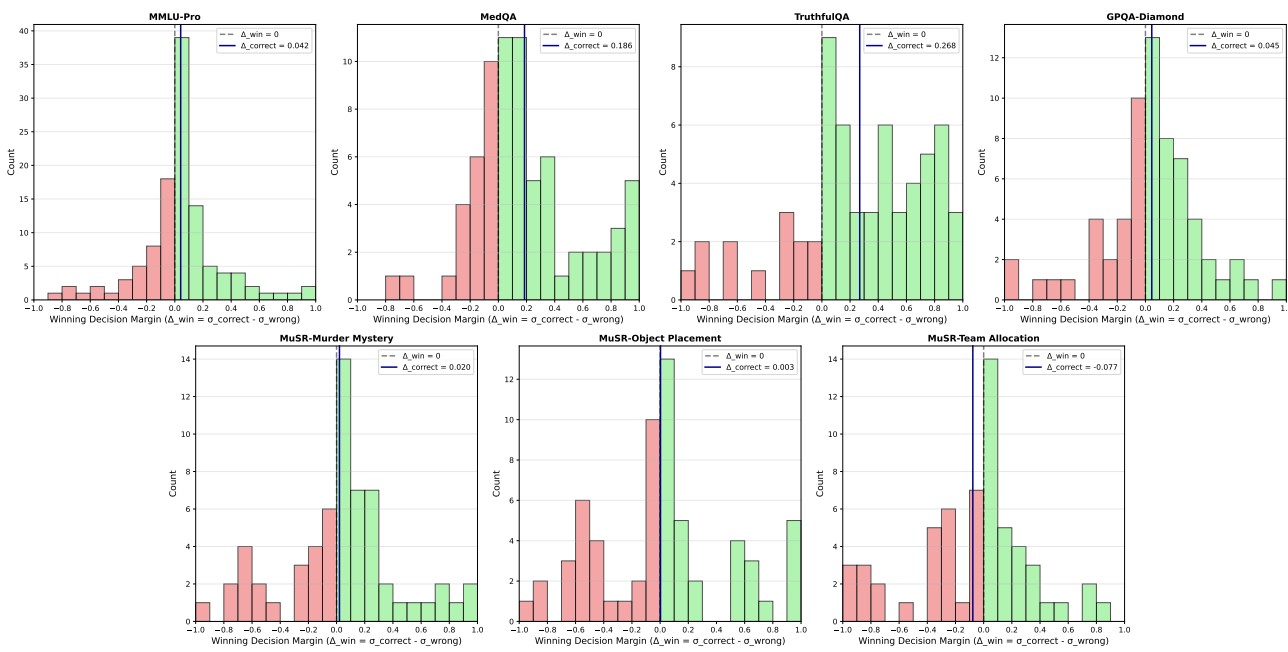

*Figure 5.* Score distribution histogram plots for each of the evaluation benchmarks tested on the `gpt-4o-mini` model, with DF-QuAD used as our choice of quantitative semantics. The strongest winning argument difference, $\Delta_{\text{win}}$, is defined as the difference between $\sigma_{\text{correct}}$ and $\sigma_{\text{wrong}}$ (refer to Section 5.1). That is, $\Delta_{\text{win}} = \sigma_{\text{correct}} - \sigma_{\text{wrong}}$. The blue line specifies the average $\Delta_{\text{win}}$ value for the entire benchmark.

- **TruthfulQA (Best - SD-DF-QuAD):** $p_{\mathrm{NRE}} = 0.073$, NRE $= +0.098$, with a substantial accuracy gain in the main results ($82.4\% \rightarrow 88.6\%$). The improvement is consistent with the favorable reversal imbalance and a strong retention of correct decisions on non-reversal instances.

- **MuSR Murder Mystery (DF-QuAD):** $p_{\mathrm{NRE}} = 0.095$, NRE $= +0.123$, with an accuracy gain ($63.2\% \rightarrow 66.4\%$). Mystery-solving benefits from integrating partially conflicting evidence, a structure well-matched to multi-expert argumentation.

These outcomes support the hypothesis that argumentation is most beneficial when tasks demand misconception detection (TruthfulQA), reconciling cross-domain constraints (MMLU-Pro, GPQA), or integrating conflicting clues (MuSR Murder Mystery).

**Null or near-null results.** Two of the MuSR subtasks show little evidence of directional benefit:

- **Object Placement (best):** NRE $= 0.000$, $p_{\mathrm{NRE}} = 0.598$. A symmetric split (8 positive, 8 negative) suggests no net corrective bias under reversal-pair conditioning.

- **Team Allocation:** NRE $= +0.014$, $p_{\mathrm{NRE}} = 0.500$, indicating a near-zero effect with balanced reversal directions.

These tasks emphasize consistent long-context state tracking across multi-step updates; distributing reasoning across multiple experts may dilute critical state information, which suggests a potential mismatch with ARGORA's argumentation format.

**Semantic Choice Shapes Both Accuracy and Paired Evidence.** Table 7 shows that the choice of quantitative modular semantics (as shown in Table 4) affects not only accuracy (Table 1) but also the reversal-pair distribution $(n_{-\rightarrow+}, n_{+\rightarrow-})$, and thus the strength of paired statistical evidence. Notably:

- **MMLU-Pro:** DF-QuAD yields $p_{\mathrm{NRE}} = 0.074$ with NRE $= +0.050$, while the best semantics (REB, Euler-based) yields $p_{\mathrm{NRE}} = 0.026$ with NRE $= +0.061$. Even with similar NRE magnitudes, semantics can change which instances become reversal pairs and how those reversals split by direction.

- **TruthfulQA:** DF-QuAD yields $p_{\mathrm{NRE}} = 0.194$ with $(n_{-\rightarrow+}, n_{+\rightarrow-}) = (8, 4)$, while the best semantics (SDQ, SD-DF-QuAD) yields $p_{\mathrm{NRE}} = 0.073$ with $(9, 3)$. Here, shifting a small number of reversals materially changes the paired evidence when $T$ is small.

- **GPQA Diamond:** DF-QuAD yields $p_{\mathrm{NRE}} = 0.416$ with a near-balanced $(12, 10)$, while the best semantics (QE, Quadratic Energy) yields $p_{\mathrm{NRE}} = 0.039$ with $(15, 6)$. This shift suggests that QE better aligns the aggregation/influence behavior with expert-level scientific reasoning disagreements.

These comparisons are *within-ARGORA* (using the same resulting arguments and the argumentation graph) and indicate that semantics is a meaningful design axis that governs how support and attack signals translate into corrective versus harmful reversals. Because "best" is selected per benchmark, we interpret these differences as descriptive evidence that semantics matters, rather than as a claim of universally optimal tuning. This motivates future work on task-adaptive or learned semantics that can specialize to domain-specific disagreement structures.

### G.4. Main Evaluation with a Different Base Model

We additionally repeat the main evaluation using a stronger base model to assess whether the reversal-pair patterns and task-dependent utility profile persist under higher base capability. In particular, we use the more capable `GPT-5-mini` model over the `GPT-4o-mini` used for our main evaluations. Due to the increased API cost of using the `GPT-5-mini` model over `GPT-4o-mini`, we limit our test to the five most difficult benchmark datasets from our original evaluation. We repeat the main evaluations (Table 8) and report the relevant significance testing (Table 9) for each of the evaluation benchmarks.

**Discussion.** The results shown for the `gpt-5-mini` model (Table 8) suggest when the base model is strong enough to maintain coherent long-context state and represent multiple constraints faithfully, structured argumentation can become *more* effective on tasks where *collective reasoning* matters. This appears most clearly on MuSR Team Allocation, where

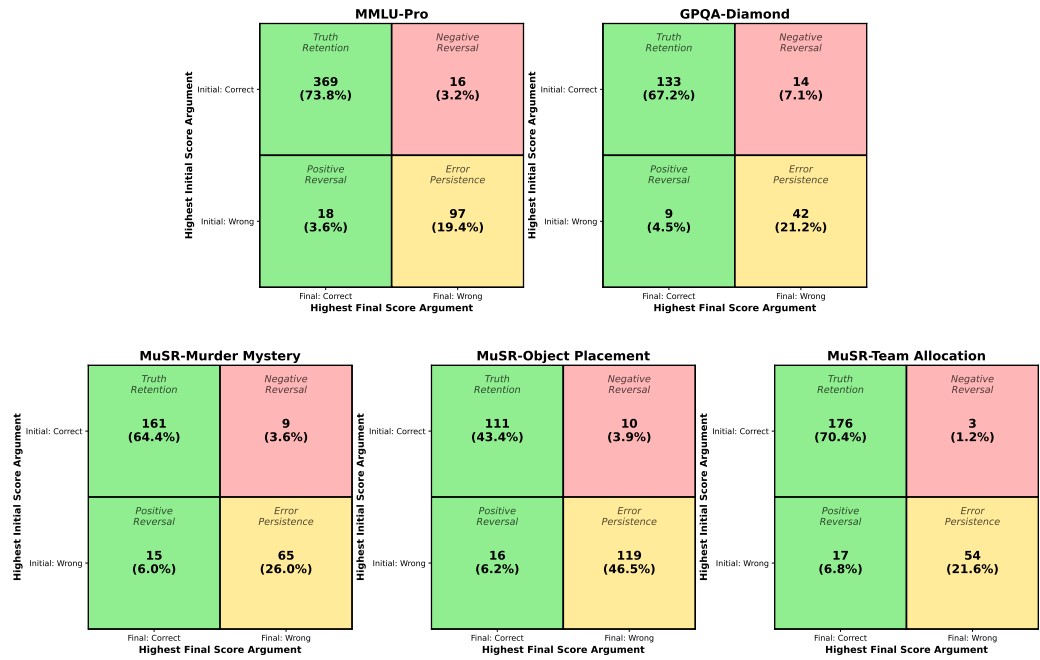

*Figure 6.* The four transition counts: $n_{+\to+}$ (truth retention), $n_{-\to+}$ (positive reversal), $n_{+\to-}$ (negative reversal), $n_{-\to-}$ (error persistence), as defined in Section 5.1 visualized as a confusion matrix for each of the evaluation benchmarks tested on the `gpt-5-mini` model, with DF-QuAD used as our choice of quantitative semantics. We highlight the truth retention and positive reversal counts as green (good outcome), the error persistence as yellow, and the negative reversal as red (unwanted outcome).

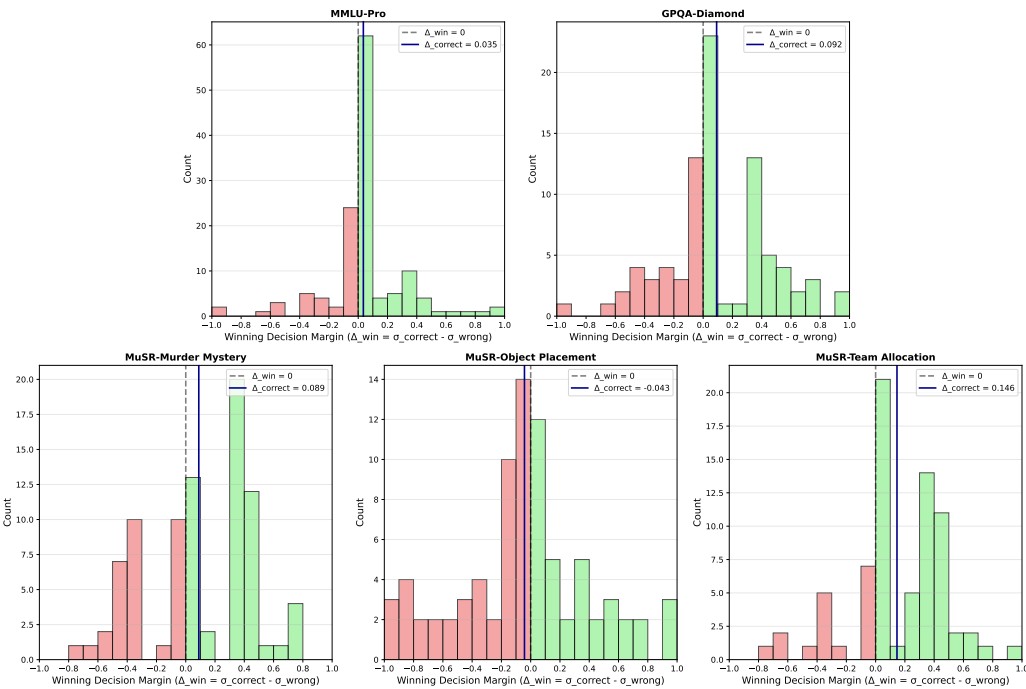

*Figure 7.* Score distribution histogram plots for each of the evaluation benchmarks tested on the `gpt-5-mini` model, with DF-QuAD used as our choice of quantitative semantics. The strongest winning argument difference, $\Delta_{\text{win}}$, is defined as the difference between $\sigma_{\text{correct}}$ and $\sigma_{\text{wrong}}$ (refer to Section 5.1). That is, $\Delta_{\text{win}} = \sigma_{\text{correct}} - \sigma_{\text{wrong}}$. The blue line specifies the average $\Delta_{\text{win}}$ value for the entire benchmark.

*Table 8.* Evaluation of ARGORA across datasets with a different baseline model (unlike Table 1, all variants use the `GPT-5-mini` model). In this experiment, we choose our number of experts to be 5. Therefore, the baseline majority vote ($MV_5$) uses 5 independent samples. We report both DF-QuAD and the best-performing semantics ("Best") under the same conditions. If the best-performing semantics coincides with DF-QuAD, we replace the entire "Best" block with the marker $\equiv$ to indicate identical semantics (and thus identical entries) to the DF-QuAD block. Metrics: Acc = accuracy; NRE (Net Reversal Efficiency), marked as $+$ or $-$; $\Delta_{correct}$ = correctness margin. We mark the best-performing variant with **boldface**, and the second best-performing variant with an underline (we only consider the *post-override* metrics of the DF-QuAD semantics).

| Dataset | N | Baselines | | | | ARGORA (*pre-override*) | | | | | | ARGORA (*post-override*) | |
|---|---|---|---|---|---|---|---|---|---|---|---|---|---|
| | | Direct 1× | Direct MV₃ | ARGORA-like (modified CoT) 1× | ARGORA-like (modified CoT) MV₃ | DF-QuAD | | | Best | | | DF-QuAD | Best |
| | | Acc | Acc | Acc | Acc | Acc | NRE | $\Delta_{correct}$ | Acc | NRE | $\Delta_{correct}$ | Acc | Acc |
| MMLU-Pro | 500 | 0.780 | **0.821** | 0.808 | **0.821** | 0.798 | $+$ | $+0.035$ | 0.806 | $+$ | $+0.027$ (SDQ) | 0.794 ($\downarrow$) | 0.804 ($\downarrow$) |
| GPQA Diamond | 198 | 0.671 | 0.742 | 0.747 | 0.762 | 0.768 | $-$ | $+0.092$ | 0.778 | $-$ | $+0.141$ (QE) | **0.768** ($-$) | 0.778 ($-$) |
| MuSR (Murder Mystery) | 250 | 0.568 | 0.704 | 0.680 | 0.696 | 0.712 | $+$ | $+0.089$ | 0.724 | $+$ | $+0.029$ (EBT) | **0.720** ($\uparrow$) | 0.724 ($-$) |
| MuSR (Object Placement) | 256 | 0.441 | 0.480 | 0.441 | 0.453 | 0.492 | $+$ | $-0.042$ | | $\equiv$ | | **0.492** ($-$) | $\equiv$ |
| MuSR (Team Allocation) | 250 | 0.712 | 0.764 | 0.744 | **0.796** | 0.780 | $+$ | $+0.146$ | 0.780 | $+$ | $+0.096$ (SDQ) | 0.776 ($\downarrow$) | 0.780 ($-$) |

*Table 9.* Paired significance tests for Argumentative Utility under ARGORA (*pre-override*) to complement the evaluation of ARGORA under the `GPT-5-mini` base model (Table 8). For each dataset, we report the pre-override accuracy (Acc), Net Reversal Efficiency (NRE), disagreement-conditioned positive and negative reversal counts ($n_{-\to+}$, $n_{+\to-}$), and the exact McNemar $p$-value for NRE ($p_{NRE}$) as defined in Appendix G. Results are shown for DF-QuAD and the best-performing semantics ("Best"). We indicate $p_{NRE}$ with N/A if the NRE value is negative, as our one-sided $p$-value test is only interpretable when the metric is nonnegative.

| Dataset | N | $|\mathcal{N}_{disagree}|$ | ARGORA (*pre-override*): DF-QuAD | | | | | ARGORA (*pre-override*): Best | | | | |
|---|---|---|---|---|---|---|---|---|---|---|---|---|
| | | | Acc | $n_{-\to+}$ | $n_{+\to-}$ | NRE | $p_{NRE}$ | Acc | $n_{-\to+}$ | $n_{+\to-}$ | NRE | $p_{NRE}$ |
| MMLU-Pro | 500 | 169 | 0.798 | 18 | 16 | 0.012 | **0.432** | 0.806 | 11 | 4 | 0.041 | **0.059** |
| GPQA Diamond | 198 | 97 | 0.768 | 9 | 14 | $-0.051$ | N/A | 0.778 | 12 | 14 | $-0.021$ | N/A |
| MuSR (Murder Mystery) | 250 | 88 | 0.712 | 15 | 9 | 0.068 | **0.153** | 0.724 | 12 | 6 | 0.068 | **0.119** |
| MuSR (Object Placement) | 256 | 95 | 0.492 | 16 | 10 | 0.063 | **0.163** | | | $\equiv$ | | **0.163** |
| MuSR (Team Allocation) | 250 | 83 | 0.780 | 17 | 3 | 0.169 | **0.001** | 0.780 | 17 | 2 | 0.181 | **0.0004** |

the `GPT-4o-mini` setting showed near-null paired evidence (Table 7: NRE $= +0.014$, $p_{NRE} = 0.500$), but the `GPT-5-mini` setting yields the strongest paired signal (NRE $= +0.169$, $p_{NRE} = 0.001$, with $(n_{-\to+}, n_{+\to-}) = (17, 3)$). We interpret this as a **capability-threshold effect**: below a minimum competence level, multi-expert fragmentation can harm state tracking (so argumentation adds noise), but above that threshold, disagreement becomes informative and ARGORA can reliably surface inconsistencies and reconcile constraints.

### G.5. Hallucination Risk Reduction

In this section, we provide an extended analysis of how ARGORA's core mechanisms—contestation, quantitative semantics, and causal intervention—address persistent challenges in language models, particularly those related to factual accuracy and output verification. While our benchmarks do not directly measure free-form factuality, the main evaluations and paired reversal tests provide a concrete empirical signal of *when* the mechanism behaves in a corrective direction (i.e., when decision flips are more often wrong→correct than correct→wrong), which is closely aligned with reducing hallucination-like failure modes in black-box settings (Table 7 and Table 9).

Large language models (LLMs) may produce *hallucinations*: fluent, confident statements that are unsupported or false. In the common black-box deployment setting, the model output is often the only available artifact, making hallucinations hard to detect, contest, or correct without additional structure or external grounding (Huang et al., 2025). ARGORA does not claim to *guarantee* factual correctness in the absence of external evidence; rather, its design choices can *reduce hallucination risk* by (i) increasing contestability, (ii) exposing mechanism-level diagnostics that quantify when corrections are likely versus risky, and (iii) enabling guardrailed corrections when an additional signal (human or model-based) is available. Empirically, our paired tests show that on several benchmarks ARGORA exhibits a directional corrective bias on reversal pairs (Table 7), meaning that *when* the mechanism changes a decision, it more often fixes an error than introduces one—a

property that is directly relevant to mitigating hallucination-like failures under limited observability.

**(1) Contestation reduces *uncontested* hallucinations.** ARGORA elicits multiple main arguments and supplementary support/attack arguments, forcing candidate answers to face explicit counter-arguments. A brittle or internally inconsistent claim is more likely to be challenged during the debate phase, reducing the chance that it survives as a single unopposed narrative. While multi-agent discussion cannot eliminate shared misconceptions, it can reduce the frequency of hallucinations that persist *without* any articulated opposition (Du et al., 2024). In our evaluations, this intuition is consistent with reversal-pair patterns in which positive reversals can outnumber negative reversals on misconception- and reasoning-heavy benchmarks (Table 7), suggesting that contestation can create opportunities for error detection rather than merely amplifying noise.

**(2) Quantitative semantics acts as a consistency filter.** By mapping the resulting arguments from the discussion process to a QBAF and applying a fixed modular semantics, ARGORA converts qualitative debate into a quantitative score over candidate main arguments. This provides a principled way to downweight candidate answers that rely on weak auxiliary claims or that are strongly attacked, and to prefer answers with stronger internal support under the chosen semantics. In this way, the scoring layer serves as a soft filter that penalizes internally weak or poorly supported content—a common failure mode of hallucinations (Dhuliawala et al., 2024). Notably, Table 7 shows that semantics can change the reversal-pair distribution $(n_{-\to+}, n_{+\to-})$ and thus the strength of paired evidence even when the same argument graph is used; this supports the view that quantitative semantics is not merely a post-hoc score, but a control factor that governs whether contestation yields corrective versus harmful flips.

**(3) Mechanism-level counterfactuals identify fragile rationales.** Casting the QBAF evaluation as a deterministic SCM (see Prop. 4.1) makes it possible to ask targeted counterfactual questions of the form: *"Would the selected answer remain strong if this particular influence edge were removed?"* Edge-local interventions (Def. 4.2) and the observation-aligned objective (Def. 4.6) provide a computationally cheap diagnostic of sensitivity: if the strength of the main argument changes substantially under small, localized removals, then the decision is *fragile* and may rely on a small number of influential argumentative channels. Such fragility can be surfaced as a warning signal (e.g., "this decision is highly sensitive to a single supporting claim"), which is especially relevant when hallucinations manifest as a confident conclusion supported by a small number of brittle premises. Viewed alongside the paired statistics (Table 7), these counterfactual probes provide complementary diagnostics: reversal-pair bias characterizes *net corrective tendency*, while edge-local sensitivity characterizes *how brittle the internal justification is*.

**(4) Human- or judge-assisted priors enable direct correction without changing the mechanism.** In ARGORA, base scores (priors) enter the SCM as exogenous inputs. This means the framework is compatible with substituting or calibrating priors using external feedback: a capable human reviewer (or a trusted tool-driven verifier) can directly adjust priors for questionable arguments without altering the downstream causal propagation defined by the modular semantics (Yao et al., 2023b). Moreover, the observation-aligned override mechanism (Def. 4.6) operationalizes this idea: if an external (QBAF-agnostic) signal prefers a different main argument, ARGORA can search for a minimal-cost edge-local intervention that aligns the internal consensus with the external preference, which may correct certain hallucination-induced failures in a controlled and interpretable manner. We further analyze this guardrailed correction mechanism via observation-aligned overrides in App. G.6.

**Clarifications and limitations.** The above mechanisms and design choices from ARGORA as described primarily address hallucination risk arising from (i) lack of contestation, (ii) weak internal support structure, or (iii) reliance on a small number of fragile premises. They do not guarantee factuality when all agents share the same false belief or when no reliable external signal is available. Further, the paired reversal tests themselves highlight that corrective behavior is *not universal*: depending on task structure and base capability, reversal-pair conditioning can be neutral or even negative (Table 9), indicating that additional contestation can sometimes over-correct a previously-correct decision. Therefore, ARGORA should be viewed as complementary to grounding methods (e.g., knowledge base retrieval, tools, or human verification): it adds interpretability and guardrailed control over how internal argumentative structure affects the final decision, and it provides actionable diagnostics and intervention handles for reducing hallucination risk in practice.

### G.6. Observation-Aligned Override by Cost Type

As motivated in Section 4.4, the observation-aligned override addresses a fundamental grounding failure mode: internal causal coherence does not necessarily imply external correctness. In this section, we examine when the observational signal meaningfully disagrees with the internal consensus, and perform an ablation on the override objective with respect to the intervention-cost proxy and its tradeoff weight.

**Cost-proxy ablation motivation.** While the main paper uses the SCM-state cost $C_{\text{state}}$ as the default perturbation proxy, this choice is not unique. One can penalize interventions by (i) induced changes in SCM node strengths (state-change costs) or (ii) induced disruption of graph structure (edge-based costs), and each can be computed either globally or only over the intervention-reachable substructure. We therefore examine multiple cost formulations to test whether override behavior is sensitive to how "minimal causal edits" are operationalized, and to verify that our default formulation is not a fragile design choice.

Concretely, we evaluate (i) whether distributional divergence provides a computable diagnostic of override headroom, (ii) how the choice of cost proxy changes the "structural footprint" required to align with $p_{\text{obs}}$, and (iii) whether the winner-confidence gate prevents harmful over-correction across tasks. From the hallucination-risk perspective (App. G.5), this mechanism can be viewed as a controlled correction step: when an external signal flags the internal winner as likely wrong, we seek the smallest causal intervention that achieves label-level alignment.

**Four cost metrics $C_k(\mathcal{I})$.** We compare the efficacy of the override process by varying the definition of the intervention-cost term. The four proxies below form two families (state-change vs. edge-structure), each with a global main-argument-set variant and an intervention-reachable variant.

**(1) SCM-state cost, $C_{\text{state}}$.** Refer to Def. 4.6. Let

$$N_{\text{tot}} := \sum_{m \in \mathcal{A}^{\text{main}}} |\mathcal{A}_m|.$$

Then

$$C_{\text{state}}(\mathcal{I}) := \frac{1}{N_{\text{tot}}} \sum_{m \in \mathcal{A}^{\text{main}}} \sum_{a \in \mathcal{A}_m} \left| \sigma^{Q_m}(a) - \sigma^{Q_m^{\mathcal{I}}}(a) \right|. \tag{5}$$

**(2) Intervention-reachable SCM-state cost, $C_{\text{state}}^{\text{reach}}$.** Instead of averaging over all nodes, we average only over those whose strengths are affected by $\mathcal{I}$. Define the affected index set

$$\mathcal{A}_{\text{reach}}(\mathcal{I}) := \left\{ (m, a) : m \in \mathcal{A}^{\text{main}}, \ a \in \mathcal{A}_m, \ \sigma^{Q_m^{\mathcal{I}}}(a) \neq \sigma^{Q_m}(a) \right\}. \tag{6}$$

We then define

$$C_{\text{state}}^{\text{reach}}(\mathcal{I}) := \begin{cases} \dfrac{1}{|\mathcal{A}_{\text{reach}}(\mathcal{I})|} \displaystyle\sum_{(m,a) \in \mathcal{A}_{\text{reach}}(\mathcal{I})} \left| \sigma^{Q_m}(a) - \sigma^{Q_m^{\mathcal{I}}}(a) \right|, & |\mathcal{A}_{\text{reach}}(\mathcal{I})| > 0, \\[4mm] 0, & |\mathcal{A}_{\text{reach}}(\mathcal{I})| = 0. \end{cases} \tag{7}$$

**(3) Edge-count cost, $C_{\text{edge}}$.** Let $E_m$ be the edge set of the rooted subgraph for main argument $m$, and let

$$M_{\text{tot}} := \sum_{m \in \mathcal{A}^{\text{main}}} |E_m|.$$

Since $\mathcal{I} = (I_m)_{m \in \mathcal{A}^{\text{main}}}$ deletes exactly one edge $(u \to \mathbf{Pa}(u))$ per $u \in I_m$, the total number of deleted edges is $\sum_{m \in \mathcal{A}^{\text{main}}} |I_m|$. We use the normalized edge-count proxy

$$C_{\text{edge}}(\mathcal{I}) := \frac{\sum_{m \in \mathcal{A}^{\text{main}}} |I_m|}{M_{\text{tot}}}. \tag{8}$$

(Under a single-edge intervention regime, $\sum_m |I_m| \in \{0, 1\}$.)

*Table 10.* Jensen–Shannon divergence (JS divergence) statistics between the observation distribution $p_{\mathrm{obs}}$ and the baseline consensus distribution $p_{\mathrm{QBAF}}$ (*pre-override*). We stratify by whether the top-1 labels match (*same*) or differ (*different*). All numeric entries are truncated to at most 3 significant digits.

| Dataset | $N$ | $N_{\mathrm{same}}$ | $N_{\mathrm{diff}}$ | Overall | | | Same label | | | Different label | | |
|---|---|---|---|---|---|---|---|---|---|---|---|---|
| | | | | Mean (Std) | Median | P90 | Mean (Std) | Median | P90 | Mean (Std) | Median | P90 |
| MMLU-Pro | 500 | 459 | 41 | 0.00413 (0.0145) | 0.000066 | 0.00763 | 0.00369 (0.0141) | 0.000046 | 0.00557 | 0.00911 (0.0168) | 0.00117 | 0.0322 |
| MedQA | 500 | 484 | 16 | 0.00411 (0.0179) | 0.000023 | 0.00204 | 0.00372 (0.0169) | 0.000018 | 0.00123 | 0.0159 (0.0343) | 0.000394 | 0.0503 |
| TruthfulQA | 500 | 488 | 12 | 0.00299 (0.0129) | 0.000057 | 0.00187 | 0.00273 (0.0121) | 0.000054 | 0.000894 | 0.0135 (0.0306) | 0.00288 | 0.0221 |
| GPQA Diamond | 198 | 171 | 27 | 0.00872 (0.0235) | 0.000318 | 0.0238 | 0.00789 (0.0235) | 0.000169 | 0.0185 | 0.0140 (0.0226) | 0.00298 | 0.0559 |
| MuSR (Murder Mysteries) | 250 | 237 | 13 | 0.00725 (0.0263) | 0.000029 | 0.00842 | 0.00525 (0.0196) | 0.000024 | 0.00222 | 0.0438 (0.0704) | 0.00382 | 0.105 |
| MuSR (Object Placement) | 256 | 245 | 11 | 0.0075 (0.0208) | 0.000054 | 0.0219 | 0.0063 (0.0174) | 0.000051 | 0.0199 | 0.0343 (0.0511) | 0.0120 | 0.101 |
| MuSR (Team Allocation) | 250 | 234 | 16 | 0.0138 (0.0357) | 0.000086 | 0.0673 | 0.0102 (0.0307) | 0.000081 | 0.0399 | 0.0664 (0.0562) | 0.0631 | 0.152 |

*Table 11.* Cost statistics of the intervention that minimizes JS divergence (i.e., $\lambda = 0$), computed over instance where the top-1 labels differ. All numeric entries are truncated to at most 3 significant digits.

| Dataset | $N_{\mathrm{diff}}$ | $C_{\mathrm{edge}}$ | | $C_{\mathrm{edge}}^{\mathrm{reach}}$ | | $C_{\mathrm{state}}$ | | $C_{\mathrm{state}}^{\mathrm{reach}}$ | |
|---|---|---|---|---|---|---|---|---|---|
| | | Mean (Std) | Median | Mean (Std) | Median | Mean (Std) | Median | Mean (Std) | Median |
| MMLU-Pro | 41 | 0.0622 (0.0379) | 0.0532 | 0.117 (0.0615) | 0.0993 | 0.00899 (0.00997) | 0.00632 | 0.127 (0.113) | 0.123 |
| MedQA | 16 | 0.0538 (0.0255) | 0.0454 | 0.117 (0.0575) | 0.120 | 0.0093 (0.0104) | 0.00464 | 0.104 (0.0803) | 0.102 |
| TruthfulQA | 12 | 0.0664 (0.0326) | 0.0526 | 0.133 (0.0577) | 0.125 | 0.0131 (0.0121) | 0.00597 | 0.160 (0.0668) | 0.149 |
| GPQA Diamond | 27 | 0.0360 (0.00933) | 0.0344 | 0.0732 (0.0266) | 0.0689 | 0.00407 (0.00389) | 0.00282 | 0.0994 (0.072) | 0.0734 |
| MuSR (Murder Mysteries) | 13 | 0.0679 (0.0256) | 0.0575 | 0.111 (0.0711) | 0.0888 | 0.0191 (0.0113) | 0.0203 | 0.255 (0.209) | 0.186 |
| MuSR (Object Placement) | 11 | 0.129 (0.0121) | 0.125 | 0.211 (0.0775) | 0.250 | 0.0541 (0.0423) | 0.0335 | 0.354 (0.168) | 0.263 |
| MuSR (Team Allocation) | 16 | 0.0632 (0.0283) | 0.0513 | 0.145 (0.0919) | 0.102 | 0.0127 (0.00745) | 0.0146 | 0.276 (0.187) | 0.257 |

**(4) Intervention-reachable edge-count cost, $C_{\mathrm{edge}}^{\mathrm{reach}}$.** Edge deletions can disconnect entire subtrees from the main-argument root. Let $\mathcal{A}_m^{\mathcal{I},\uparrow} \subseteq \mathcal{A}_m$ denote the set of nodes that remain connected to the root $m$ in the intervened tree $Q_m^{\mathcal{I}}$. Define the set of edges that lose root-reachability:

$$E_{\mathrm{lost}}(\mathcal{I}) := \big\{ (m, u \to v) : m \in \mathcal{A}^{\mathrm{main}}, (u \to v) \in E_m, u \notin \mathcal{A}_m^{\mathcal{I},\uparrow} \text{ or } v \notin \mathcal{A}_m^{\mathcal{I},\uparrow} \big\}. \tag{9}$$

We then normalize by the total number of edges:

$$C_{\mathrm{edge}}^{\mathrm{reach}}(\mathcal{I}) := \frac{|E_{\mathrm{lost}}(\mathcal{I})|}{M_{\mathrm{tot}}}. \tag{10}$$

Intuitively, this proxy charges an intervention according to the size of the substructure that becomes disconnected from the main-argument root.

**Evaluation methodology.** We sweep the override objective over (i) the cost proxy $C_k \in \{C_{\mathrm{state}}, C_{\mathrm{state}}^{\mathrm{reach}}, C_{\mathrm{edge}}, C_{\mathrm{edge}}^{\mathrm{reach}}\}$ and (ii) the tradeoff weight $\lambda$. For each setting, we apply the resulting override policy across all evaluation instances and report beneficial and harmful overrides (and their net) under both the ungated policy and the winner-confidence gated policy (Def. 4.6). The analysis done in this section uses the experimental settings in Section 5 of the main paper, where we use `gpt-4o-mini` as the base model and DF-QuAD as the choice of semantics.

**JS divergence as a diagnostic for override headroom.** Since the benchmarks we consider are evaluated at the *answer-label* level, we initially compare the baseline and observational winners via the answer-mapping function $L : \mathcal{A}^{\mathrm{main}} \to \mathcal{Y}$ (Sec. 5.1). Concretely, let $m^\star$ and $m^{\mathrm{obs}}$ be the baseline and observational winners from Def. 4.5, and define the induced

predicted labels $y^\star := L(m^\star)$ and $y^{\mathrm{obs}} := L(m^{\mathrm{obs}})$. We then partition instances into same answer label ($y^\star = y^{\mathrm{obs}}$) and different answer label ($y^\star \neq y^{\mathrm{obs}}$), yielding counts $N_{\mathrm{same}}$ and $N_{\mathrm{diff}}$ in Tab. 10.

To quantify distributional misalignment, we report $\mathrm{JS}(p_{\mathrm{QBAF}}\|p_{\mathrm{obs}})$ (Tab. 10), where $p_{\mathrm{QBAF}}$ and $p_{\mathrm{obs}}$ are the normalized consensus distributions over main arguments (Def. 4.5). The statistics exhibit two stable trends. When the induced labels agree, the JS divergence median is extremely small (on the order of $10^{-5}$ across datasets), indicating that $p_{\mathrm{QBAF}}$ and $p_{\mathrm{obs}}$ allocate probability mass similarly even when they are produced by different mechanisms. When the induced labels disagree (different label), the divergence increases by orders of magnitude, with medians ranging from $3.94 \times 10^{-4}$ (MedQA) up to $6.31 \times 10^{-2}$ (MuSR Team Allocation), and high-percentile values reaching $> 10^{-1}$. This observational vs. argumentative discrepancy is where an observation-aligned override can be meaningful: there is nontrivial distributional misalignment *and* a disagreement at the answer-label level.

Finally, $N_{\mathrm{diff}}$ is a small minority of each benchmark split (e.g., 12/500 for TruthfulQA, 16/250 for MuSR Team Allocation, and 27/198 for GPQA Diamond), implying that label-level disagreement between $p_{\mathrm{QBAF}}$ and $p_{\mathrm{obs}}$ is relatively rare. Together with the winner-confidence gate in Def. 4.6, this suggests that observation-aligned overrides will naturally be triggered only on a limited subset of instances.

**Cost statistics under pure alignment** ($\lambda = 0$). To calibrate the perturbation scale, we next measure the intervention cost incurred by the *best-alignment* intervention, i.e., the configuration $\mathcal{I}^\star \in \arg\min_{\mathcal{I} \in \mathcal{I}_{\mathrm{cand}}} \mathrm{JS}(p_{\mathrm{QBAF}}^{\mathcal{I}}\|p_{\mathrm{obs}})$ obtained by setting $\lambda = 0$ in $J(\mathcal{I})$ (Def. 4.6), subject to the same override feasibility constraints (in particular, the alignment target is defined at the label level through $L$). We report costs only on valid disagreement instances ($N_{\mathrm{diff}}$, Tab. 11), since there is no reason to undergo override when the two answer labels match.

Tab. 11 shows that absolute cost scales differ by proxy: $C_{\mathrm{state}}$ tends to be smallest (medians $\approx 10^{-3}$–$10^{-2}$), $C_{\mathrm{edge}}$ is moderate (medians $\approx 3 \times 10^{-2}$–$1.3 \times 10^{-1}$), and the reachable variants are larger still by construction (medians often $\approx 7 \times 10^{-2}$–$2.6 \times 10^{-1}$). Overall, even when enforcing label-level alignment with the observational signal, strong JS improvements are often achievable with localized interventions, but the measured "structural footprint" can depend strongly on the chosen $C_k$.

$\lambda$ **sweep range selection.** The tables provide a principled guide for selecting $\lambda$: we want $\lambda C_k(\mathcal{I})$ to be comparable to the typical JS divergence values; otherwise the optimizer either behaves like pure alignment ($\lambda$ too small) or no overrides at all ($\lambda$ too large). Using medians as an order-of-magnitude calibration, the implied break-even ratios $\lambda \approx \mathrm{median}(\mathrm{JS}_{\mathrm{diff}})/\mathrm{median}(C_k)$ span roughly $3 \times 10^{-3}$ up to $\approx 4$ across datasets and cost types (with the largest values arising when $\mathrm{JS}_{\mathrm{diff}}$ is high while $C_{\mathrm{state}}$ is low).

To cover this range while keeping a single shared grid across all four $C_k$, we use a log-spaced sweep of $\lambda$ as follows:

$$\lambda \in \{0,\ 0.001,\ 0.002,\ 0.005,\ 0.01,\ 0.02,\ 0.05,\ 0.1,\ 0.2,\ 0.5\}.$$

This includes (i) the pure-alignment endpoint ($\lambda = 0$), (ii) the expected trade-off region for all four cost types under label-level disagreement, and (iii) sufficiently large values to empirically confirm override inhibition (where the optimal configuration increasingly prefers non-interventions due to perturbation penalty).

**Selecting a single default** $(C_k, \lambda)$. For the main experiments we require a single, fixed override configuration shared across datasets. We select this configuration by prioritizing *safety* (nonnegative net gain across most datasets) and *stability* (behavior not overly sensitive to rare disagreement instances). Empirically, this criterion favors the original SCM-state cost $C_{\mathrm{state}}$ with a moderate tradeoff weight $\lambda = 0.05$, which tends to suppress harmful corrections while still permitting beneficial overrides during observational disagreement. Accordingly, we adopt $C_{\mathrm{state}}$ with $\lambda = 0.05$ as the default configuration in all experiments in our work.

**Winner-confidence gate prevents harmful over-correction.** Table 12 evaluates override behavior across the four cost variants and $\lambda \in \{0, 0.001, \dots, 0.5\}$, with an additional ablation test for comparing the effect of override policies with and without the confidence gate $p_{\mathrm{obs}}(m^{\mathrm{obs}}) - p_{\mathrm{QBAF}}(m^\star) \geq \tau$ as defined in Def. 4.6. As mentioned in Tab. 6, we set the gating threshold value to $\tau = 0$. Without the winner-confidence gate, forcing alignment with $p_{\mathrm{obs}}$ whenever labels differ yields substantial negative net gains:

- **MuSR Team Allocation**: Ungated override with $C_{\mathrm{state}}$ at $\lambda = 0$ produces 0/5 (net $-5$).

*Table 12.* Override grid results (full $\lambda$ sweep). Each cell reports *positive / negative* (**net gain**), where positive counts beneficial overrides (*wrong* QBAF winner → *correct* observational winner), negative counts harmful overrides (*correct* QBAF winner → *wrong* observational winner), and net gain=positive−negative. We compare the ungated policy (ungated) against the winner-confidence gated policy (gated).

| Dataset | Cost | Variant | $\lambda$ | | | | | | | | | |
|---|---|---|---|---|---|---|---|---|---|---|---|---|
| | | | 0 | 0.001 | 0.002 | 0.005 | 0.01 | 0.02 | 0.05 | 0.1 | 0.2 | 0.5 |
| MMLU-Pro | $C_{\text{edge}}$ | ungated | 3/3 (**0**) | 2/3 (**−1**) | 2/3 (**−1**) | 2/3 (**−1**) | 2/3 (**−1**) | 2/3 (**−1**) | 2/1 (**+1**) | 1/0 (**+1**) | 0/0 (**0**) | 0/0 (**0**) |
| | | gated | 0/1 (**−1**) | 0/1 (**−1**) | 0/1 (**−1**) | 0/1 (**−1**) | 0/1 (**−1**) | 0/1 (**−1**) | 0/0 (**0**) | 0/0 (**0**) | 0/0 (**0**) | 0/0 (**0**) |
| | $C_{\text{edge}}^{\text{reach}}$ | ungated | 3/3 (**0**) | 2/3 (**−1**) | 2/3 (**−1**) | 2/3 (**−1**) | 2/3 (**−1**) | 2/2 (**0**) | 1/0 (**+1**) | 0/0 (**0**) | 0/0 (**0**) | 0/0 (**0**) |
| | | gated | 0/1 (**−1**) | 0/1 (**−1**) | 0/1 (**−1**) | 0/1 (**−1**) | 0/1 (**−1**) | 0/0 (**0**) | 0/0 (**0**) | 0/0 (**0**) | 0/0 (**0**) | 0/0 (**0**) |
| | $C_{\text{state}}$ | ungated | 3/3 (**0**) | 3/3 (**0**) | 3/3 (**0**) | 2/3 (**−1**) | 2/3 (**−1**) | 2/3 (**−1**) | 2/3 (**−1**) | 2/3 (**−1**) | 1/3 (**−2**) | 0/2 (**−2**) |
| | | gated | 0/1 (**−1**) | 0/1 (**−1**) | 0/1 (**−1**) | 0/1 (**−1**) | 0/1 (**−1**) | 0/1 (**−1**) | 0/1 (**−1**) | 0/1 (**−1**) | 0/1 (**−1**) | 0/1 (**−1**) |
| | $C_{\text{state}}^{\text{reach}}$ | ungated | 3/3 (**0**) | 2/3 (**−1**) | 2/3 (**−1**) | 2/3 (**−1**) | 2/3 (**−1**) | 0/3 (**−3**) | 0/0 (**0**) | 0/0 (**0**) | 0/0 (**0**) | 0/0 (**0**) |
| | | gated | 0/1 (**−1**) | 0/1 (**−1**) | 0/1 (**−1**) | 0/1 (**−1**) | 0/1 (**−1**) | 0/1 (**−1**) | 0/0 (**0**) | 0/0 (**0**) | 0/0 (**0**) | 0/0 (**0**) |
| MedQA | $C_{\text{edge}}$ | ungated | 4/3 (**+1**) | 2/2 (**0**) | 2/2 (**0**) | 2/1 (**+1**) | 1/1 (**0**) | 1/0 (**+1**) | 0/0 (**0**) | 0/0 (**0**) | 0/0 (**0**) | 0/0 (**0**) |
| | | gated | 2/1 (**+1**) | 1/1 (**0**) | 1/1 (**0**) | 1/0 (**+1**) | 0/0 (**0**) | 0/0 (**0**) | 0/0 (**0**) | 0/0 (**0**) | 0/0 (**0**) | 0/0 (**0**) |
| | $C_{\text{edge}}^{\text{reach}}$ | ungated | 4/3 (**+1**) | 2/2 (**0**) | 2/2 (**0**) | 1/1 (**0**) | 1/0 (**+1**) | 0/0 (**0**) | 0/0 (**0**) | 0/0 (**0**) | 0/0 (**0**) | 0/0 (**0**) |
| | | gated | 2/1 (**+1**) | 1/1 (**0**) | 1/1 (**0**) | 0/0 (**0**) | 0/0 (**0**) | 0/0 (**0**) | 0/0 (**0**) | 0/0 (**0**) | 0/0 (**0**) | 0/0 (**0**) |
| | $C_{\text{state}}$ | ungated | 4/3 (**+1**) | 4/3 (**+1**) | 4/3 (**+1**) | 4/3 (**+1**) | 3/3 (**0**) | 3/3 (**0**) | 3/2 (**+1**) | 1/1 (**0**) | 0/1 (**−1**) | 0/0 (**0**) |
| | | gated | 2/1 (**+1**) | 2/1 (**+1**) | 2/1 (**+1**) | 2/1 (**+1**) | 2/1 (**+1**) | 2/1 (**+1**) | 2/0 (**+2**) | 1/0 (**+1**) | 0/0 (**0**) | 0/0 (**0**) |
| | $C_{\text{state}}^{\text{reach}}$ | ungated | 4/3 (**+1**) | 3/3 (**0**) | 3/2 (**+1**) | 1/1 (**0**) | 1/1 (**0**) | 0/0 (**0**) | 0/0 (**0**) | 0/0 (**0**) | 0/0 (**0**) | 0/0 (**0**) |
| | | gated | 2/1 (**+1**) | 2/1 (**+1**) | 2/0 (**+2**) | 0/0 (**0**) | 0/0 (**0**) | 0/0 (**0**) | 0/0 (**0**) | 0/0 (**0**) | 0/0 (**0**) | 0/0 (**0**) |
| TruthfulQA | $C_{\text{edge}}$ | ungated | 2/3 (**−1**) | 2/3 (**−1**) | 2/3 (**−1**) | 2/3 (**−1**) | 1/2 (**−1**) | 1/2 (**−1**) | 1/0 (**+1**) | 0/0 (**0**) | 0/0 (**0**) | 0/0 (**0**) |
| | | gated | 0/0 (**0**) | 0/0 (**0**) | 0/0 (**0**) | 0/0 (**0**) | 0/0 (**0**) | 0/0 (**0**) | 0/0 (**0**) | 0/0 (**0**) | 0/0 (**0**) | 0/0 (**0**) |
| | $C_{\text{edge}}^{\text{reach}}$ | ungated | 2/3 (**−1**) | 2/3 (**−1**) | 2/3 (**−1**) | 2/2 (**0**) | 1/2 (**−1**) | 0/1 (**−1**) | 0/0 (**0**) | 0/0 (**0**) | 0/0 (**0**) | 0/0 (**0**) |
| | | gated | 0/0 (**0**) | 0/0 (**0**) | 0/0 (**0**) | 0/0 (**0**) | 0/0 (**0**) | 0/0 (**0**) | 0/0 (**0**) | 0/0 (**0**) | 0/0 (**0**) | 0/0 (**0**) |
| | $C_{\text{state}}$ | ungated | 2/3 (**−1**) | 2/3 (**−1**) | 2/3 (**−1**) | 2/3 (**−1**) | 2/3 (**−1**) | 2/3 (**−1**) | 2/3 (**−1**) | 1/2 (**−1**) | 1/1 (**0**) | 0/0 (**0**) |
| | | gated | 0/0 (**0**) | 0/0 (**0**) | 0/0 (**0**) | 0/0 (**0**) | 0/0 (**0**) | 0/0 (**0**) | 0/0 (**0**) | 0/0 (**0**) | 0/0 (**0**) | 0/0 (**0**) |
| | $C_{\text{state}}^{\text{reach}}$ | ungated | 2/3 (**−1**) | 2/3 (**−1**) | 2/3 (**−1**) | 1/2 (**−1**) | 1/2 (**−1**) | 0/0 (**0**) | 0/0 (**0**) | 0/0 (**0**) | 0/0 (**0**) | 0/0 (**0**) |
| | | gated | 0/0 (**0**) | 0/0 (**0**) | 0/0 (**0**) | 0/0 (**0**) | 0/0 (**0**) | 0/0 (**0**) | 0/0 (**0**) | 0/0 (**0**) | 0/0 (**0**) | 0/0 (**0**) |
| GPQA Diamond | $C_{\text{edge}}$ | ungated | 5/5 (**0**) | 4/4 (**0**) | 4/4 (**0**) | 3/3 (**0**) | 3/3 (**0**) | 2/3 (**−1**) | 1/1 (**0**) | 0/1 (**−1**) | 0/1 (**−1**) | 0/0 (**0**) |
| | | gated | 2/0 (**+2**) | 1/0 (**+1**) | 1/0 (**+1**) | 1/0 (**+1**) | 1/0 (**+1**) | 1/0 (**+1**) | 0/0 (**0**) | 0/0 (**0**) | 0/0 (**0**) | 0/0 (**0**) |
| | $C_{\text{edge}}^{\text{reach}}$ | ungated | 5/5 (**0**) | 4/4 (**0**) | 3/3 (**0**) | 3/3 (**0**) | 2/3 (**−1**) | 1/1 (**0**) | 0/1 (**−1**) | 0/1 (**−1**) | 0/0 (**0**) | 0/0 (**0**) |
| | | gated | 2/0 (**+2**) | 1/0 (**+1**) | 1/0 (**+1**) | 1/0 (**+1**) | 1/0 (**+1**) | 0/0 (**0**) | 0/0 (**0**) | 0/0 (**0**) | 0/0 (**0**) | 0/0 (**0**) |
| | $C_{\text{state}}$ | ungated | 5/5 (**0**) | 5/4 (**+1**) | 5/4 (**+1**) | 4/4 (**0**) | 4/4 (**0**) | 3/4 (**−1**) | 3/4 (**−1**) | 3/4 (**−1**) | 3/4 (**−1**) | 0/2 (**−2**) |
| | | gated | 2/0 (**+2**) | 2/0 (**+2**) | 2/0 (**+2**) | 1/0 (**+1**) | 1/0 (**+1**) | 1/0 (**+1**) | 1/0 (**+1**) | 1/0 (**+1**) | 1/0 (**+1**) | 0/0 (**0**) |
| | $C_{\text{state}}^{\text{reach}}$ | ungated | 5/5 (**0**) | 3/4 (**−1**) | 3/4 (**−1**) | 3/4 (**−1**) | 2/2 (**0**) | 0/2 (**−2**) | 0/0 (**0**) | 0/0 (**0**) | 0/0 (**0**) | 0/0 (**0**) |
| | | gated | 2/0 (**+2**) | 1/0 (**+1**) | 1/0 (**+1**) | 1/0 (**+1**) | 0/0 (**0**) | 0/0 (**0**) | 0/0 (**0**) | 0/0 (**0**) | 0/0 (**0**) | 0/0 (**0**) |
| MuSR (Murder Mysteries) | $C_{\text{edge}}$ | ungated | 2/2 (**0**) | 2/2 (**0**) | 2/2 (**0**) | 2/1 (**+1**) | 2/1 (**+1**) | 1/1 (**0**) | 0/1 (**−1**) | 0/1 (**−1**) | 0/1 (**−1**) | 0/0 (**0**) |
| | | gated | 0/0 (**0**) | 0/0 (**0**) | 0/0 (**0**) | 0/0 (**0**) | 0/0 (**0**) | 0/0 (**0**) | 0/0 (**0**) | 0/0 (**0**) | 0/0 (**0**) | 0/0 (**0**) |
| | $C_{\text{edge}}^{\text{reach}}$ | ungated | 2/2 (**0**) | 2/2 (**0**) | 2/2 (**0**) | 2/1 (**+1**) | 2/1 (**+1**) | 1/1 (**0**) | 0/1 (**−1**) | 0/1 (**−1**) | 0/0 (**0**) | 0/0 (**0**) |
| | | gated | 0/0 (**0**) | 0/0 (**0**) | 0/0 (**0**) | 0/0 (**0**) | 0/0 (**0**) | 0/0 (**0**) | 0/0 (**0**) | 0/0 (**0**) | 0/0 (**0**) | 0/0 (**0**) |
| | $C_{\text{state}}$ | ungated | 2/2 (**0**) | 2/2 (**0**) | 2/2 (**0**) | 2/2 (**0**) | 2/2 (**0**) | 2/2 (**0**) | 1/2 (**−1**) | 1/2 (**−1**) | 0/1 (**−1**) | 0/1 (**−1**) |
| | | gated | 0/0 (**0**) | 0/0 (**0**) | 0/0 (**0**) | 0/0 (**0**) | 0/0 (**0**) | 0/0 (**0**) | 0/0 (**0**) | 0/0 (**0**) | 0/0 (**0**) | 0/0 (**0**) |
| | $C_{\text{state}}^{\text{reach}}$ | ungated | 2/2 (**0**) | 2/2 (**0**) | 2/2 (**0**) | 1/2 (**−1**) | 1/2 (**−1**) | 0/1 (**−1**) | 0/0 (**0**) | 0/0 (**0**) | 0/0 (**0**) | 0/0 (**0**) |
| | | gated | 0/0 (**0**) | 0/0 (**0**) | 0/0 (**0**) | 0/0 (**0**) | 0/0 (**0**) | 0/0 (**0**) | 0/0 (**0**) | 0/0 (**0**) | 0/0 (**0**) | 0/0 (**0**) |
| MuSR (Object Placement) | $C_{\text{edge}}$ | ungated | 1/2 (**−1**) | 1/1 (**0**) | 1/1 (**0**) | 1/1 (**0**) | 1/1 (**0**) | 1/1 (**0**) | 0/1 (**−1**) | 0/0 (**0**) | 0/0 (**0**) | 0/0 (**0**) |
| | | gated | 0/1 (**−1**) | 0/0 (**0**) | 0/0 (**0**) | 0/0 (**0**) | 0/0 (**0**) | 0/0 (**0**) | 0/0 (**0**) | 0/0 (**0**) | 0/0 (**0**) | 0/0 (**0**) |
| | $C_{\text{edge}}^{\text{reach}}$ | ungated | 1/2 (**−1**) | 1/1 (**0**) | 1/1 (**0**) | 1/1 (**0**) | 1/1 (**0**) | 1/1 (**0**) | 0/0 (**0**) | 0/0 (**0**) | 0/0 (**0**) | 0/0 (**0**) |
| | | gated | 0/1 (**−1**) | 0/0 (**0**) | 0/0 (**0**) | 0/0 (**0**) | 0/0 (**0**) | 0/0 (**0**) | 0/0 (**0**) | 0/0 (**0**) | 0/0 (**0**) | 0/0 (**0**) |
| | $C_{\text{state}}$ | ungated | 1/2 (**−1**) | 1/2 (**−1**) | 1/2 (**−1**) | 1/1 (**0**) | 1/1 (**0**) | 1/1 (**0**) | 0/1 (**−1**) | 0/1 (**−1**) | 0/0 (**0**) | 0/0 (**0**) |
| | | gated | 0/1 (**−1**) | 0/1 (**−1**) | 0/1 (**−1**) | 0/0 (**0**) | 0/0 (**0**) | 0/0 (**0**) | 0/0 (**0**) | 0/0 (**0**) | 0/0 (**0**) | 0/0 (**0**) |
| | $C_{\text{state}}^{\text{reach}}$ | ungated | 1/2 (**−1**) | 1/1 (**0**) | 1/1 (**0**) | 1/1 (**0**) | 1/1 (**0**) | 0/0 (**0**) | 0/0 (**0**) | 0/0 (**0**) | 0/0 (**0**) | 0/0 (**0**) |
| | | gated | 0/1 (**−1**) | 0/0 (**0**) | 0/0 (**0**) | 0/0 (**0**) | 0/0 (**0**) | 0/0 (**0**) | 0/0 (**0**) | 0/0 (**0**) | 0/0 (**0**) | 0/0 (**0**) |
| MuSR (Team Allocation) | $C_{\text{edge}}$ | ungated | 0/5 (**−5**) | 0/5 (**−5**) | 0/4 (**−4**) | 0/4 (**−4**) | 0/3 (**−3**) | 0/2 (**−2**) | 0/2 (**−2**) | 0/2 (**−2**) | 0/1 (**−1**) | 0/1 (**−1**) |
| | | gated | 0/0 (**0**) | 0/0 (**0**) | 0/0 (**0**) | 0/0 (**0**) | 0/0 (**0**) | 0/0 (**0**) | 0/0 (**0**) | 0/0 (**0**) | 0/0 (**0**) | 0/0 (**0**) |
| | $C_{\text{edge}}^{\text{reach}}$ | ungated | 0/5 (**−5**) | 0/4 (**−4**) | 0/4 (**−4**) | 0/4 (**−4**) | 0/2 (**−2**) | 0/2 (**−2**) | 0/2 (**−2**) | 0/1 (**−1**) | 0/1 (**−1**) | 0/1 (**−1**) |
| | | gated | 0/0 (**0**) | 0/0 (**0**) | 0/0 (**0**) | 0/0 (**0**) | 0/0 (**0**) | 0/0 (**0**) | 0/0 (**0**) | 0/0 (**0**) | 0/0 (**0**) | 0/0 (**0**) |
| | $C_{\text{state}}$ | ungated | 0/5 (**−5**) | 0/5 (**−5**) | 0/5 (**−5**) | 0/5 (**−5**) | 0/5 (**−5**) | 0/5 (**−5**) | 0/4 (**−4**) | 0/2 (**−2**) | 0/2 (**−2**) | 0/1 (**−1**) |
| | | gated | 0/0 (**0**) | 0/0 (**0**) | 0/0 (**0**) | 0/0 (**0**) | 0/0 (**0**) | 0/0 (**0**) | 0/0 (**0**) | 0/0 (**0**) | 0/0 (**0**) | 0/0 (**0**) |
| | $C_{\text{state}}^{\text{reach}}$ | ungated | 0/5 (**−5**) | 0/5 (**−5**) | 0/4 (**−4**) | 0/3 (**−3**) | 0/2 (**−2**) | 0/1 (**−1**) | 0/1 (**−1**) | 0/0 (**0**) | 0/0 (**0**) | 0/0 (**0**) |
| | | gated | 0/0 (**0**) | 0/0 (**0**) | 0/0 (**0**) | 0/0 (**0**) | 0/0 (**0**) | 0/0 (**0**) | 0/0 (**0**) | 0/0 (**0**) | 0/0 (**0**) | 0/0 (**0**) |

- **TruthfulQA**: Ungated $C_{\text{edge}}$ at low $\lambda$ yields 2/3 (net $-1$).

- **MMLU-Pro / GPQA**: Ungated configurations frequently achieve net gain $\leq 0$.

On the other hand, the confidence gate successfully blocks most harmful corrections. Across the full grid (7 datasets $\times$ 4 cost types $\times$ 10 $\lambda$ values), gated overrides achieve non-negative net gain in 247/280 configurations (88.2%), versus 161/280 (57.5%) for ungated. In particular:

- On **MuSR Team Allocation**, the gate blocks all override attempts, correctly recognizing internal consensus as more reliable.

- On **MedQA** and **GPQA Diamond**, gated overrides with $C_{\text{state}}$ achieve consistent positive net gains (+1 to +2).

The override mechanism exhibits clear task-dependency consistent with the grounding hypothesis:

- **High utility** (MedQA, GPQA): Gated overrides yield net positive corrections where domain-specific reasoning may cause QBAF over-fitting.

- **Neutral utility** (MMLU-Pro, TruthfulQA, MuSR Murder Mystery): Balanced outcomes suggest internal/external agreement.

- **Correctly inhibited** (MuSR Team Allocation): Gate blocks all attempts where long-context coherence is critical.

**Effect of cost-proxy choice.** Across all four cost proxies, the qualitative role of the override mechanism is stable: overrides are only potentially useful on the rare $N_{\text{diff}}$ subset, and the winner-confidence gate is the dominant factor preventing harmful over-correction. The primary effect of changing $C_k$ is to rescale the effective perturbation penalty (shifting where the "override-active" region occurs along the $\lambda$ axis), rather than to change which datasets benefit from overrides. Among the proxies, $C_{\text{state}}$ is the most direct operationalization of "minimal causal edit" in the induced SCM and yields the most consistent trade-off behavior under a shared $\lambda$ grid; accordingly, we adopt $C_{\text{state}}$ with $\lambda = 0.05$ as the default in all experiments.

**Summary** Our evaluation demonstrates some key properties:

1. **Diagnostic value**: JS divergence conditioned on label disagreement signals potential over-confidence.

2. **Safety**: Winner-confidence gate maintains $\geq 88\%$ non-negative net gain versus 57% ungated.

3. **Default configuration**: Based on the sweep, we fix $C_{\text{state}}$ and $\lambda = 0.05$ for all experiments, as it provides a robust safety–utility trade-off under a single shared setting.

We view these results as evidence that observation-aligned overrides can provide targeted external grounding with measurable safeguards when paired with an explicit confidence gate.

# H. Ablation Study

As ARGORA contains various design choices and tunable hyperparameters, we detail out some of the ablation studies to examine how critical components or design choices affect the framework.

### H.1. Contextual Orthogonality Pruning Threshold

We examine the role of contextual orthogonality pruning in ARGORA by varying the pruning threshold $\rho_{\text{sim}}$ while holding all other components of the pipeline fixed. Concretely, we evaluate ARGORA on the GPQA-Diamond benchmark using the same configuration as in the main evaluation (Sec. 5): `gpt-4o-mini` as the base model, three experts, and DF-QuAD as the choice of modular semantics. The pruning threshold $\rho_{\text{sim}}$ controls the aggressiveness with which supplementary arguments are removed based on contextual similarity (Sec. 4.2). We sweep values of $\rho_{\text{sim}}$ from 0.1 to 0.9, with more fine-grained increments from the 0.5 to 0.9 range. We also evaluate a special case of applying no contextual pruning with the value of 1.0 for the threshold.

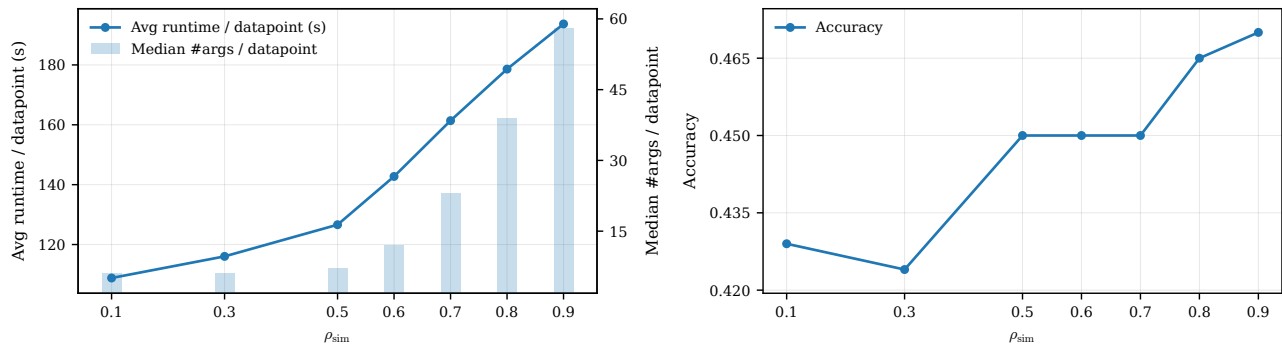

*Figure 8.* **Effect of threshold for contextual orthogonality pruning on the GPQA-Diamond benchmark.** (Left) Average runtime per datapoint as a function of the pruning threshold $\rho_{\text{sim}}$, overlaid with the median number of supplementary arguments retained per datapoint (bars; excludes main arguments). (Right) Pre-override accuracy versus $\rho_{\text{sim}}$ under the same evaluation setting.

*Table 13.* Paired significance tests and performance statistics for Argumentative Utility under ARGORA based on base score initialization method. For each dataset, we report the pre-override accuracy (Acc), Net Reversal Efficiency (NRE), disagreement-conditioned positive and negative reversal counts ($n_{-\to+}$, $n_{+\to-}$), and the exact McNemar $p$-value for NRE ($p_{\text{NRE}}$) as defined in Appendix G. Results are shown for DF-QuAD and the best-performing semantics ("Best"). We indicate $p_{\text{NRE}}$ with N/A if the NRE value is negative, as our one-sided $p$-value test is only interpretable when the metric is nonnegative. We report the higher final accuracy metric in **bold**.

| Dataset | $N$ | $|\mathcal{N}_{\text{disagree}}|$ | ARGORA (Orchestrator Init.) | | | | | ARGORA (Neutral Init. ($w = 0.5$)) | | | | |
|---|---|---|---|---|---|---|---|---|---|---|---|---|
| | | | Acc | $n_{-\to+}$ | $n_{+\to-}$ | NRE | $p_{\text{NRE}}$ | Acc | $n_{-\to+}$ | $n_{+\to-}$ | NRE | $p_{\text{NRE}}$ |
| MMLU-Pro | 500 | 180 | **0.638** | 20 | 11 | 0.050 | 0.074 | 0.618 | 18 | 13 | 0.027 | 0.237 |
| MedQA | 500 | 86 | **0.812** | 14 | 5 | 0.105 | 0.032 | 0.806 | 9 | 8 | 0.116 | 0.500 |
| TruthfulQA | 500 | 61 | **0.882** | 8 | 4 | 0.066 | 0.194 | 0.874 | 18 | 5 | 0.213 | 0.005 |
| GPQA Diamond | *198* | 106 | **0.450** | 12 | 10 | 0.019 | 0.416 | 0.419 | 12 | 13 | $-0.009$ | N/A |
| MuSR (Murder Mystery) | *250* | 57 | **0.664** | 14 | 7 | 0.123 | 0.095 | 0.648 | 8 | 10 | $-0.036$ | N/A |
| MuSR (Object Placement) | *256* | 75 | **0.523** | 7 | 11 | $-0.053$ | N/A | 0.512 | 7 | 5 | 0.026 | 0.387 |
| MuSR (Team Allocation) | *250* | 70 | **0.564** | 9 | 8 | 0.014 | 0.500 | 0.560 | 11 | 12 | $-0.014$ | N/A |

**Effect of the contextual orthogonality pruning threshold.** Figure 8 illustrates the effect of the contextual orthogonality pruning threshold $\rho_{\text{sim}}$ on both computational cost and performance on GPQA-Diamond. When the pruning threshold is set below 0.5, only a small number of supplementary arguments survive pruning, resulting in compact argument graphs with relatively low runtime. As $\rho_{\text{sim}}$ increases beyond 0.5, pruning becomes substantially more permissive, leading to a steep increase in the median number of retained supplementary arguments per datapoint. This transition marks the range in which contextual orthogonality pruning meaningfully shapes the size of the induced QBAF.

While a more lenient pruning threshold yields modest gains in pre-override accuracy, these gains come at a rapidly increasing computational cost. As the number of supplementary arguments grows, runtime per datapoint increases sharply, reflecting not only the larger argumentation graph but also the increased history context that must be tracked and processed by each expert during discussion. In the extreme case where $\rho_{\text{sim}} = 1.0$, effectively disabling contextual orthogonality pruning, smaller models such as `gpt-4o-mini` fail to complete inference reliably due to excessive context length induced by unfiltered supplementary arguments. Our evaluations show that beyond computational overhead, uncontrolled argument growth can render the discussion process infeasible under practical context window constraints on smaller LLMs.

We conclude that contextual orthogonality pruning is a necessary mechanism for controlling token usage and maintaining tractable context windows during multi-argument discussion, while preserving most of the attainable accuracy. Based on the observed trade-off between accuracy and efficiency, we fix $\rho_{\text{sim}} = 0.7$ as the default pruning threshold in all experiments.

## H.2. Base Score Initialization

We examine the role of base score initialization in ARGORA by comparing two variants that differ *only* in their initial argument scores. Since our goal is to isolate the effect of the base score function $w$ on the induced argumentative winner, we

evaluate *pre-override* performance only (i.e., before applying any observation-alignment override). In the **orchestrator-initialized** setting, each argument is assigned a task-aware base score produced by the Orchestrator during initialization (as described in Sec. D.8). In the **neutral initialization** setting, all arguments are instead initialized to the constant $w(a) = 0.5$. All other components of the pipeline—including expert generations, the constructed argument graphs, and the quantitative semantics used for aggregation—are held fixed across the two settings. Consequently, this ablation constitutes a controlled intervention on the base score over argument strengths, and any resulting differences in the pre-override winner and accuracy can be attributed to the choice of initialization.

Table 13 reports the (pre-override) performance and paired significance tests for Argumentative Utility under both settings. For each dataset, we report the accuracy, disagreement-conditioned positive and negative reversal counts ($n_{-\to+}$, $n_{+\to-}$), Net Reversal Efficiency (NRE), and the exact one-sided McNemar $p$-value for NRE as defined in Appendix G. The $p_{\text{NRE}}$ statistic is reported only when NRE is nonnegative, as the one-sided test is interpretable only when overrides are directionally beneficial.

**Effect of base score initialization.** Across datasets, orchestrator initialization consistently yields nonnegative NRE and interpretable significance tests, indicating that overrides tend to correct incorrect argumentative winners rather than introduce new errors. In contrast, neutral initialization frequently degrades directional reliability: for several datasets (e.g., GPQA Diamond and multiple MuSR tasks), NRE becomes negative, rendering $p_{\text{NRE}}$ undefined. This behavior indicates that, under neutral base scores, the override mechanism can become actively harmful, flipping correct argumentative winners to incorrect observational ones.

In addition, we observe that the *accuracy* under orchestrator initialization is consistently higher than the corresponding accuracy under neutral initialization across all evaluated datasets, suggesting that task-aware base scores improve not only the directionality of reversals but also the quality of the underlying argumentative consensus significantly. Notably, these differences arise despite identical argumentation framework structures and fixed quantitative semantics, and therefore directly support the view that orchestrator-based base score initialization is a necessary structural component that stabilizes the induced causal pathways in ARGORA.

**Interpreting apparent gains under neutral initialization.** An apparent exception arises on TruthfulQA, where neutral initialization yields a statistically significant $p_{\text{NRE}}$ (0.005). However, this result warrants careful interpretation. When all arguments are initialized to $w(a) = 0.5$, the base score induces a complete tie among candidates, effectively reducing the initial argumentative winner selection to random choice. In this environment, reversal counts are driven less by the quality of argumentative structure and more by properties of the benchmark dataset itself.

In particular, TruthfulQA is a binary classification task with only two answer choices and a relatively high base accuracy (around 0.88 in our evaluations as shown in Table 13). Under such conditions, random initial arguments with wrong labels are disproportionately likely to revert to winners corresponding to the correct answer, making positive reversals statistically easier to obtain regardless of argumentative merit. Consequently, the observed significance under neutral initialization reflects dataset-level class structure and accuracy saturation rather than meaningful argumentative correction. This interpretation is further supported by the instability of neutral initialization on multi-choice datasets, where no such effect is observed.

**Implication.** Overall, this ablation demonstrates that task-aware base score initialization is essential for stable and interpretable argumentative correction in ARGORA. While neutral initialization may occasionally yield favorable statistics in highly constrained binary settings, it lacks robustness and can induce harmful override behavior on more complex tasks. In contrast, orchestrator-initialized base scores consistently preserve the directional reliability of NRE across datasets, reinforcing their role as a necessary design component rather than an optional optimization.

### H.3. Number of Rounds

ARGORA is implemented with optional support for multi-round discussions (Sec. D.10), where the full discussion pipeline can be repeated for $R > 1$ rounds. Based on this design, we run ARGORA using the exact evaluation settings in the main evaluation 5 and varying the number of total rounds $R$ from 1 to 3 rounds. For each dataset, we report (i) accuracy before the observation-aligned override, (ii) accuracy after the override, and (iii) the average runtime per datapoint in seconds.

Table 14 shows the results of the ablation on the number of rounds. Across the seven benchmarks, increasing $R$ does not

*Table 14.* Ablation on the number of discussion rounds $R$ in ARGORA as implemented in Section D.10. For each dataset, we report *pre-override* and *post-override* accuracy, and the average runtime (in seconds) per datapoint. We use the same evaluation configuration as the main experiment in Section 5: gpt-4o-mini as the base model, 3 experts, and DF-QuAD semantics. The best performing *post-override* accuracy is formatted in **bold**.

| Dataset | N | ARGORA ($R=1$) | | | ARGORA ($R=2$) | | | ARGORA ($R=3$) | | |
|---|---|---|---|---|---|---|---|---|---|---|
| | | Accuracy (*pre-override*) | Accuracy (*post-override*) | Runtime (sec) (avg per data) | Accuracy (*pre-override*) | Accuracy (*post-override*) | Runtime (sec) (avg per data) | Accuracy (*pre-override*) | Accuracy (*post-override*) | Runtime (sec) (avg per data) |
| MMLU-Pro | 500 | 0.638 | **0.636** ($\downarrow$) | 129.33 | 0.624 | 0.620 ($\downarrow$) | 174.04 | 0.633 | 0.633 ($-$) | 232.83 |
| MedQA | 500 | 0.812 | **0.816** ($\uparrow$) | 111.51 | 0.804 | 0.804 ($-$) | 151.99 | 0.812 | 0.812 ($-$) | 207.32 |
| TruthfulQA | 500 | 0.882 | **0.882** ($-$) | 97.14 | 0.852 | 0.852 ($-$) | 147.82 | 0.876 | 0.876 ($-$) | 202.00 |
| GPQA Diamond | 198 | 0.450 | **0.455** ($\uparrow$) | 129.41 | 0.414 | 0.414 ($-$) | 199.56 | 0.455 | 0.455 ($-$) | 285.88 |
| MuSR (Murder Mystery) | 250 | 0.664 | **0.664** ($-$) | 108.63 | 0.620 | 0.620 ($-$) | 170.08 | 0.644 | 0.644 ($-$) | 243.86 |
| MuSR (Object Placement) | 256 | 0.523 | 0.523 ($-$) | 88.85 | 0.543 | **0.551** ($\uparrow$) | 139.57 | 0.547 | 0.547 ($-$) | 201.20 |
| MuSR (Team Allocation) | 250 | 0.564 | 0.564 ($-$) | 103.79 | 0.524 | 0.524 ($-$) | 166.47 | 0.564 | **0.568** ($\uparrow$) | 236.80 |

consistently improve either pre-override or post-override accuracy; the best post-override result is in fact often achieved at $R = 1$. We do observe isolated cases where additional rounds help, but these gains are not systematic and do not indicate a reliable performance–rounds correlation under the current multi-round prompting and history utilization methodology.

In contrast, runtime scales predictably in a linear fashion with respect to $R$. This linear scaling in runtime cost is expected from the framework design, since each additional round repeats the full discussion pipeline (expert argument generation, multi-level argument generation, and QBAF construction/evaluation).

**Implication.** Taken together, the results suggest that, in the current implementation, multi-round discussion incurs a clear cost increase without delivering consistent accuracy gains. For this reason, we fix $R = 1$ in all of our experiments and evaluations for ARGORA in this paper, and treat improved multi-round mechanisms as potential future direction for research.

### H.4. Number of Experts Chosen

ARGORA allows the number of participating expert LLMs to be configured, with each expert contributing one main argument to the discussion graph. Because the size of the induced QBAF grows with the number of experts, this choice impacts both predictive behavior and computational cost.

We ran a small screening study on GPQA Diamond, MedQA, and MMLU-Pro, varying the number of experts from 2 to 7 to identify a practically relevant operating regime. The screening results suggest diminishing returns in raw performance: after roughly 3–5 experts, accuracy improvements largely plateau, and adding more experts beyond 5 yields little additional gain under our current prompting and aggregation setup. In contrast, because the number of experts directly scales the number of main arguments, the runtime of ARGORA grows approximately linearly with expert count. Taken together, these trends indicate that 3–5 experts offers the most favorable accuracy–efficiency trade-off in our implementation.

From a computational perspective, the number of experts determines the number of main arguments and thus the scale of the argumentation graph. Consequently, runtime grows approximately linearly in the number of experts due to expert prompting and supplementary argument generation, although contextual orthogonality pruning (Sec. 4.2) mitigates the overhead from the supplementary argument generation process by suppressing redundant arguments. Since this study is intended only as an upfront calibration and is sensitive to implementation-level choices (e.g., pruning thresholds and prompt templates), we omit the full sweep results for brevity and focus our reported experiments on our choice of expert counts (in the $3 - 5$ expert range) used throughout the main evaluation.

## I. Prompts Used in ARGORA

We provide the list of prompts that act as main components of ARGORA.

### I.1. Orchestrator Prompts

The Orchestrator is responsible for coordinating the multi-expert discussion, extracting the main task, selecting appropriate experts, and generating custom prompts for each expert.

### I.1.1. MAIN TASK EXTRACTION

```
Your job is to formulate the main task for a given discussion topic. The main task is a single sentence,
well-formed imperative that specifies the core objective to be answered from the given topic.

The discussion topic is given as follows:
Discussion Topic: {topic}
Give your output strictly in the following format:
{"main_task": <single sentence, well-formed imperative>}
```

### I.1.2. KEY ELEMENT EXTRACTION

```
You are given a discussion topic along with the main task. From this, you should specify the list of key
elements. The key elements should be a set of intended semantics (key entities, factors, elements, task
type of the discussion) to guide the discussion to properly address the topic and the main task.

The discussion topic and main task are given as follows:
Discussion Topic: {topic}
Main Task: {main_task}
Give your output in the following format:
{"key_element": <set of intended semantics>}
```

### I.1.3. EXPERT SELECTION

```
Choose the expert LLMs of a particular field to participate for a given discussion topic. You are free to
 choose any field of expertise as you see fit, so that the discussion for the topic can be as
comprehensive as possible. If the topic is very specific (for example, a specific domain like mathematics
), your topic selection can be fine-grained to optimize for that domain (e.g., Real Analysis LLM,
Algebraic Structures LLM). You can use the main task and key elements to help you choose the experts most
 relevant to the fruitful discussion of the topic. Here are some examples:

Return your answer strictly as an array of expert names as shown in the examples.
Here are the relevant details:
"{topic}"
Main task: {main_task}
Key elements: {key_elements}
```

### I.1.4. EXPERT-SPECIFIC PROMPT GENERATION

```
You are to generate a unique prompt for each expert based on the discussion topic and the expert's field
of expertise. This prompt is to be added right after the discussion topic when presenting the prompt to
the expert. Your prompt should be designed specifically to elicit each expert to fully utilize their
domain expertise.

The relevant information are given as follows:
Discussion Topic: {topic}
Main task to be addressed: {main_task}
Key elements to consider: {key_elements}
Experts: {experts}
Return your response in the following format:
{"expert_name": "custom prompt for expert", ...}
```

### I.1.5. BASE SCORE GENERATION

```
<USER>: You are to assess the overall strength of a statement given a discussion topic, main task, and
key elements. You will evaluate the statement across three separate criteria and provide individual
scores for each.

Evaluate the statement across the following three criteria, assigning a score strictly between 0 and 1 (
non-inclusive) for each:

1. Task Relevance: How directly and substantively the statement addresses the discussion topic, main task,
 and key elements.
- High-score example: A statement that explicitly engages the core question and key elements, directly
contributing to the task.
- Low-score example: A statement that is generic, tangential, or unrelated to the task, or fails to
answer the core question as instructed.
```

```
2. Evidence Support: The extent to which the statement provides reasoning, mechanisms, or evidence for
its claims.
- High-score example: A statement that justifies its claims with clear reasoning or concrete support.
- Low-score example: A statement that makes unsupported assertions or vague opinions.

3. Logical Soundness: Whether the statement is internally coherent, free of contradictions, and follows a
 reasonable inferential structure.
- High-score example: A statement with clear premises leading logically to a conclusion.
- Low-score example: A statement relying on invalid reasoning, circular logic, or unjustified leaps.

Scoring guidance for each criterion:
- Unless truly exceptional or poor, most scores that adequately meet each criteria should cluster around
0.50.
- Scores close to 1.00: Exceptional quality in this dimension.
- Scores close to 0.00: Poor quality in this dimension.
- Be as precise as you can within the two decimal places range (precision of 0.01).
- Be conservative with extreme scores; most should fall between.

Statement: "{statement}"
Discussion Topic: "{topic}"
Main Task: {main_task}
Key Elements: {key_elements}

Return your response strictly in JSON with the following format:
{
  "task_relevance_assessment": "<your assessment for the task relevance score>",
  "task_relevance": <float between 0 and 1>,
  "evidence_support_assessment": "<your assessment for the evidence support score>",
  "evidence_support": <float between 0 and 1>,
  "logical_soundness_assessment": "<your assessment for the logical soundness score>",
  "logical_soundness": <float between 0 and 1>,
}
```

## I.2. Expert Prompts

Each expert model is assigned a specific domain of expertise and generates arguments from that perspective.

### I.2.1. EXPERT SYSTEM PROMPT

```
You are the {expert_name}. As {expert_name}, you have deep expertise in {domain}. Using your domain
expertise, you are to provide a well-reasoned argument to come to a consensus on a given topic.
```

### I.2.2. MAIN ARGUMENT GENERATION

```
You are to provide the initial arguments for a given discussion task or topic. These arguments should be
standalone answers that directly address the given task, that are intended to be either agreed or
disagreed with by other experts.

Now, provide your main arguments for the following discussion topic:
{topic}
{orchestrator_custom_prompt}

You should STRICTLY PROVIDE at most {max_main_args} main arguments (generally, the fewer the better).
Even if some examples shown have more, you must limit yourself to the above number. Keep your arguments
as varied and distinct as possible. Your main arguments should be able to directly answer or address the
discussion topic on their own (standalone), without being dependent on other main arguments you provide.
Respond strictly in the following schema format:
{ "main_args": [ "statement 1", ... ] }
```

### I.2.3. FIRST-LEVEL ARGUMENT PROMPT

```
You are contributing first-level arguments for a debate graph.
Discussion Topic: {topic}
Primary Task: {main_task}
Key Elements: {key_elements}

Main Argument Under Review: {main_argument}

{role_description}
```

```
Provide a well-reasoned argument that justifies your stance. Your reasoning should provide a clear
insight into your thought process about why you agree or disagree with the main argument, and should not
be a simple rephrasing of the main argument itself (i.e., "I disagree with <main argument>").
{limit_instruction}
Respond strictly with the following schema format:
{
  "stance": "agree" | "disagree",
  "reasoning": [
    "<detailed reasoning behind your stance>"
  ]
}
Do not add any commentary outside of the format.
```

### I.2.4. SECOND-LEVEL ARGUMENT PROMPT

```
Expert: {expert}
Main argument under discussion:
{main_statement}
You should review the following statements contributed by other experts:
(1) {polarity} argument from {author}: {statement}
(2) ...

- Use integers starting from 1 for the index that match the numbering above.
- Stance must be exactly one of AGREE, DISAGREE, or NONE.
- Provide a well-reasoned justification for your stance based on your own expertise. Your justification
should be a novel, insightful reasoning based on your expertise, and not just a rewording or rephrasing
of the target statement.

For each statement, reply exactly in the following format:
[
  {
    "index": <integer item number>,
    "stance": "AGREE" | "DISAGREE" | "NONE",
    "justification": "<justification of chosen stance>"
  },
  ...
]
```

### I.2.5. THIRD-LEVEL ARGUMENT PROMPT

```
Expert: {expert}
Main argument:
{main_statement}
Your prior statement under review:
{parent_statement}
Other experts raised the following critiques:
(1) {critic} challenges your claim with: {statement}
(2) ...

Respond with a JSON object where each key is the critique index and each value has a 'rebuttal' field
containing your detailed rebuttal.Your rebuttal should be a novel, insightful reasoning based on your
expertise, and not just a simple negation of the critiquing statement.

- Use JSON string keys that match the numbering above ("1", "2", ...).

Reply exactly in the following format:
{
  "<critique index as string>": {
    "rebuttal": "<a detailed rebuttal or NONE>"
  },
  ...
}

If you have no rebuttal for a critique, set its 'rebuttal' to the exact token NONE.
If you offer no rebuttals at all, reply with the single token NONE.
```

**Implementation notes.** All prompts are implemented as static methods in the `Orchestrator` and `Expert` classes. Parameters shown in curly braces (e.g., {`topic`}, {`expert_name`}) are replaced with actual values at runtime.

**GPT-5 Prompt Optimization.** When using GPT-5 model variants (namely `gpt-5-mini`), additional constraints are automatically appended to system prompts to minimize reasoning tokens and disable web search:

```
 – Do not perform any web searches or external lookups.
 – Only use your internal knowledge to answer the questions.
```

## J. Use Case Evaluation Details

### J.1. Synthetic Cyber Incident Description

This appendix presents the complete synthetic cyber incident used in the use case study. The incident does not correspond to any real-world breach, and is constructed to resemble a realistic industrial ransomware report with controlled inconsistencies intended for a holistic evaluation on multi-dimensional reasoning. In particular, the specification embeds: (i) operational security (OPSEC) violations in attacker infrastructure, (ii) timeline alignment anomalies, (iii) the absence of expected forensic artifacts, and (iv) unusually precise impact and recovery metrics. All evaluated systems were provided with this exact input without modification.

The following JSON document constitutes the full specification of the synthetic ransomware incident. It is included verbatim in the input prompt for ARGORA and the baseline.

```
{
  "incident_id": "2024-APAC-HS-2331",
  "victim": {
    "organization_name": "HorizonSprings Water Technologies",
    "industry": "Industrial water treatment / IoT control systems",
    "headquarters": "Singapore",
    "employee_count": 540,
    "regional_operations": ["SG", "MY", "ID"]
  },
  "attack_overview": {
    "classification": "Ransomware (double-extortion)",
    "suspected_ransomware_family": "BlackSuit",
    "confidence": "medium",
    "notes": [
      "PowerShell-based lateral movement consistent with BlackSuit.",
      "Exfiltration used custom Python script (less common for this group)."
    ]
  },
  "timeline": {
    "initial_access": "2024-08-17T11:42:19Z",
    "privilege_escalation_detected": "2024-08-17T14:03:55Z",
    "lateral_movement": {
      "start": "2024-08-17T14:12:10Z",
      "end": "2024-08-18T02:51:44Z"
    },
    "data_exfiltration": {
      "start": "2024-08-18T03:03:18Z",
      "end": "2024-08-18T05:47:02Z"
    },
    "encryption_event": "2024-08-18T06:11:29Z",
    "incident_response_engaged": "2024-08-18T09:25:10Z",
    "attacker_public_claim_date": "2024-08-22T12:00:00Z"
  },
  "exfiltration_details": {
    "estimated_volume_gb": 22.7,
    "transfer_mechanism": "HTTPS upload to attacker-controlled VPS",
    "data_categories": [
      "SCADA configuration exports",
      "internal network diagrams",
      "controller firmware prototypes",
      "limited HR payroll extract"
    ],
    "sample_files": [
      "/mnt/ops/plant/config/scada_topology_rev5.png",
      "C:\\Archive\\firmware_prototype_v2\\controller_beta_0814.bin",
      "/shared/hr/payroll_2024_Q2_partial.csv"
    ]
  },
  "ransom_demand": {
    "currency": "BTC",
    "amount_btc": 63,
    "deadline": "2024-08-29T23:59:59Z",
    "wallet_address": "1FK8tmR4kW8YpghLNQx5G2Mhsdgz57DabQ"
  },
  "ransom_note": {
    "file_name": "RESTORE-HORIZON.txt",
```

```
      "note_excerpt": "Your systems are encrypted and your files have been downloaded by BlackSuit. Do not
      attempt to restore or modify encrypted servers. We have obtained essential internal documents,
      including technical designs and employee information. To recover your data and prevent public
      disclosure, follow payment instructions in our portal.",
      "contact_portal": "hxxp://horizonhelpdesk45n1lk.onion",
      "victim_id": "HSW-2024-884"
    },
    "impact": {
      "encrypted_servers": 14,
      "production_disruption_hours": 17,
      "iot_command_delay_seconds": 5,
      "internal_erp_recovery_time_hours": 26
    },
    "regulatory_and_compliance": {
      "pii_exposed": true,
      "regulator_notified": "Singapore National Digital Infrastructure Office",
      "notification_date": "2024-08-23"
    },
    "current_status": {
      "ransom_paid": "unknown",
      "data_leaked": "not_observed",
      "system_restoration_completed": true
    }
}
```

## J.2. Model Output Excerpts

We present the representative excerpts from the outputs of the evaluated models when applied to the synthetic cyber incident described in Appendix J. All excerpts are reproduced verbatim or lightly abridged for clarity.

**ARGORA: Final Consensus Overview**  The proposed system classified the incident as *fabricated* or misattributed. Unlike the single-model baselines, the consensus explanation explicitly grounded its decision in the absence of core forensic evidence and violations of ransomware-family–specific operational norms. In addition to a direct judgment, the system articulated counterfactual reasoning, identifying which factors were decisive and how the conclusion would change if key assumptions were removed.

**Final decision and evidence summary.**  The consensus determined that the incident was not a genuine BlackSuit ransomware attack. While some surface-level tactics (e.g., PowerShell-based lateral movement) aligned with BlackSuit, the analysis emphasized multiple contradictions:

"No encryption artifacts (file hashes or decryption keys) were found to confirm data encryption."

"The ransom portal domain includes the victim's name, violating BlackSuit's operational security norms."

"The combination of tooling deviations, timeline implausibility, and missing encryption evidence suggests a staged or misattributed attack rather than an authentic double-extortion campaign."

**Decisive reasoning chain.**  A key distinguishing feature of the consensus output is its explicit articulation of a decisive reasoning chain, identifying a minimal set of conditions that jointly support the fabricated-incident hypothesis:

1. *Ransom portal domain includes the victim's name* → violation of BlackSuit OpSec norms.
2. *No encryption artifacts observed* → core double-extortion claim remains unverified.
3. *Timeline inconsistency for a large industrial victim* → operational tempo implausible for BlackSuit.

The system concluded that this chain constitutes a set of critical assumptions: breaking any one element would weaken the conclusion, while resolving multiple elements (e.g., proving encryption or normalizing domain naming) could plausibly flip the decision.

**Counterfactual explanations.**  Beyond identifying anomalies, the consensus explanation explicitly reasoned about counterfactual scenarios—what evidence would need to change for the decision to change.

"Removing the tooling deviation (custom Python exfiltration) would weaken the argument but would not flip the decision, as other factors such as domain naming and encryption absence would still dominate."

"Breaking the reasoning chain—by confirming encryption artifacts or aligning the ransom portal with BlackSuit's OpSec norms—would significantly undermine the staged-attack hypothesis."

**Confidence and limitations.**   The system reported a medium confidence level, explicitly acknowledging limitations in attribution and incomplete external validation:

> "Misattribution to BlackSuit cannot be fully ruled out without deeper threat intelligence."

> "The absence of confirmed data leakage leaves some ransom-note claims unverified."

Overall, the output illustrates how the system not only produces a classification, but also exposes the internal dependency structure of its reasoning, identifying which assumptions are decisive and how alternative evidence would affect the outcome.

**Open-Source Model (GPT-OSS 120B)**   GPT-OSS 120B classified the incident as *real*. Its reasoning primarily emphasized narrative completeness, sector-specific plausibility, and the presence of artifacts commonly associated with ransomware incidents.

> "All observable evidence points to a genuine ransomware incident."

> "The presence of a detailed attack timeline, a ransom note with a victim-specific identifier, and quantifiable operational impact is consistent with a real double-extortion ransomware attack."

> "While attribution to BlackSuit remains tentative, the core event itself appears authentic based on the available evidence."

**Proprietary Model (Gemini 2.5 Pro)**   Gemini likewise classified the incident as *real*. Its analysis focused on the plausibility of the incident lifecycle reported and the realism of industry-specific details, treating ambiguity and incomplete information as characteristic of real-world investigations.

> "The incident appears real, with industry-specific impact metrics supporting authenticity."

> "The reported timeline and delayed incident-response engagement reflect realistic organizational discovery and escalation processes."

> "Ambiguity in attribution and incomplete status fields are common in genuine post-incident reports."

### J.3. Consensus Formation in the Multi-Expert Discussion

This appendix documents how ARGORA arrived at its final consensus for the fabricated cyber-incident use case. Rather than presenting a full discussion transcript, we selectively report key analytical signals that were independently identified and subsequently reinforced by multiple domain experts, ultimately converging on the conclusion that the incident was likely fabricated or misattributed.

Notably, this case did not involve strong adversarial disagreement. Instead, consensus emerged through progressive reinforcement, where initial observations proposed by an expert were contextualized, refined, or strengthened by others with complementary expertise. This process reflects realistic expert analysis workflows in cybersecurity incident response, where agreement often forms through corroboration rather than direct rebuttal.

**Operational Security Anomaly.**   *Initial observation (Ransomware Forensics Expert).* The ransomware contact portal domain (`horizonhelpdesk45n1lk.onion`) explicitly includes the victim's name and organizational context. This violates the operational security practices typically observed in BlackSuit campaigns, which historically rely on randomized, non-descriptive onion domains to reduce attribution risk.

*Supporting analysis (Cybersecurity Incident Analyst).* The expert noted that embedding victim-identifiable information directly into attacker infrastructure is inconsistent with the threat model of mature double-extortion groups. Such actors prioritize deniability and resilience over usability, suggesting that the portal was constructed to appear authentic rather than to support a sustained extortion operation.

*Consensus implication.* Across experts, this anomaly was interpreted not as a stylistic deviation but as a structural OPSEC failure, substantially increasing the likelihood of imitation or false-flag behavior.

**Absence of Forensic Corroboration.**   *Initial observation (Cybersecurity Incident Analyst).* Despite explicit claims of encryption and data exfiltration, the report contained no verifiable forensic artifacts, including encryption hashes, leaked file samples, cryptographic proof of possession, or publicly observable leak-site evidence.

*Supporting analysis (Digital Forensics Expert).* From a forensic point of view, the expert emphasized that BlackSuit's double-extortion model depends on credible evidence of data control to exert pressure. The absence of even minimal corroborating signals (e.g., file hashes or teaser leaks) was therefore treated as a significant inconsistency rather than a temporary information gap.

*Consensus implication.* The lack of forensic corroboration was interpreted as a decisive negative signal, as it contradicts both the operational incentives and the historical behavior of real double-extortion operations.

**Timeline Inconsistency Between Restoration and Attacker Claim.** *Initial observation (Cybersecurity Incident Analyst).* The date of the attacker's public claim coincided exactly with the reported completion of the system restoration. This synchronization is atypical, as public claims are usually timed to maximize pressure before or during recovery, not after its completion.

*Supporting analysis (Ransomware Forensics Expert).* The forensics expert noted that such temporal alignment could occur in isolation, but when combined with the absence of leaks and missing forensic artifacts, it suggests a coordinated narrative rather than an adversarial interaction between attacker and victim.

*Consensus implication.* Experts converged on the interpretation that the timeline appeared constructed to look plausible, yet lacked the asymmetric pressure dynamics normally observed in real ransomware incidents.

**Over-Precise Impact Metrics.** *Initial observation (Ransomware Forensics Expert).* The incident report specified unusually precise operational impact metrics, including a 17-hour production disruption and a 26-hour ERP recovery time.

*Supporting analysis (Cybersecurity Incident Analyst).* In real-world incident reporting, particularly during ongoing response, such figures are typically approximated or revised over time. The exactness of these metrics, in the absence of independent third-party validation, was interpreted as consistent with post-hoc narrative construction.

*Consensus implication.* Metric precision was therefore treated not as evidence of transparency, but as a credibility-engineering artifact, reinforcing the broader hypothesis of fabrication.

**Ambiguous Signals.** Certain elements, such as the use of a custom Python script for data exfiltration, generated discussion but did not play a decisive role in the final verdict. While atypical for BlackSuit, experts acknowledged that tooling adaptation is plausible in industrial IoT environments. As a result, this signal was explicitly classified as ambiguous and excluded from the core decision criteria.

**Summary.** Across multiple expert models, the discussion converged on a small set of structurally inconsistent signals, including operational security violations, missing forensic corroboration, temporal coincidence between restoration and attacker claims, and unusually precise impact metrics. The final consensus did not emerge from any single anomaly, but from reinforcement of weak signals across complementary expert perspectives, resulting in a stable classification of the incident as fabricated or misattributed.

