# OpenReview forum: "ARGORA: Orchestrated Argumentation for Causally Grounded LLM Reasoning and Decision Making"
_ICML.cc/2026/Conference — Submitted to ICML 2026_

### Official Review · Reviewer_ipPa · 2026-03-12

**Soundness:** 2
**Presentation:** 2
**Significance:** 2
**Originality:** 2
**Overall Recommendation:** 3
**Confidence:** 3

**Summary:**

This paper proposes ARGORA, a multi-expert LLM reasoning framework that converts expert discussions into explicit quantitative bipolar argumentation frameworks (QBAFs), where arguments can support or attack one another and are assigned strengths under a modular evaluation rule. The key idea is to treat the resulting graph-evaluation procedure as a deterministic structural model, enabling edge-local counterfactual interventions that identify which arguments or reasoning chains most affected the final decision.

**Compliance With Llm Reviewing Policy:**

Affirmed.

**Final Justification:**

The authors did not add experiments that addressed my concerns.

**Key Questions For Authors:**

1. What exactly distinguishes “Direct 1×” from “ARGORA-like (modified CoT) 1×,” and how do these baselines relate to the actual expert prompts used inside ARGORA? The current naming is somewhat vague. In particular, it is unclear whether “ARGORA-like (modified CoT) 1×” corresponds to one of the expert-role prompts used inside the framework or to a separate prompt template. A clearer mapping would make the baseline comparisons more interpretable. What would change my evaluation: If the authors clarify that these are already close proxies for the actual expert prompts, that would reduce my concern about baseline ambiguity; if not, I would view the current evaluation as less diagnostic.

2. Can the authors report the standalone performance of each individual expert configuration used inside ARGORA, as well as the average and best individual expert?
Since ARGORA uses three expert instances with orchestrated prompting, I think this ablation is important for separating gains from structured aggregation versus gains from prompt specialization.
What would change my evaluation: If ARGORA clearly outperforms both the average and the best standalone expert, I would view the empirical contribution more favorably. If one expert already explains most of the gain, that would weaken my assessment of the framework’s added value.

3. How should readers interpret the paper’s causal claims more precisely?
My understanding is that the SCM is induced by the deterministic QBAF update equations, and the interventions operate over that induced graph. If so, the method appears to provide counterfactual sensitivity within the aggregation formalism, rather than evidence about the LLM’s true internal reasoning process.
What would change my evaluation: If the authors substantially narrow and clarify the causal claim, I would view the paper as more sound; if they continue to use strong causal language without stronger justification, this remains a major concern.

4. Can the authors provide main-benchmark results across multiple stronger model families, rather than primarily GPT-4o-mini plus limited appendix/case-study evidence?
The main table uses GPT-4o-mini throughout, while the stronger-model evidence is more limited and the cross-model cybersecurity example is qualitative.
What would change my evaluation: If the framework shows consistent gains across stronger and more diverse backbones on the main benchmark suite, I would upgrade both significance and soundness.

5. How sensitive are the results to the single-edge restriction in the override/intervention mechanism?
The current framework appears to rely on edge-local interventions for tractability, but this may limit how meaningful the override mechanism is in more complex graphs.
What would change my evaluation: Evidence that the conclusions remain stable under richer intervention families, or a convincing argument that single-edge interventions are sufficient in practice, would make the method more compelling.

**Limitations:**

See my comments above.

**Strengths And Weaknesses:**

The paper is well motivated. It targets a real weakness of many multi-agent or debate-style LLM systems, namely that they typically aggregate answers through voting or judging without leaving behind a structured object that explains why a particular answer won. Representing the discussion as an explicit support/attack graph is a reasonable and interesting way to make the reasoning process more inspectable.

A second strength is that the paper presents a fairly coherent end-to-end framework rather than just an isolated modeling idea. The combination of multi-expert discussion, QBAF-based aggregation, edge-local intervention analysis, and an override mechanism gives the method a clear systems-level identity. I also appreciated that the paper goes beyond plain accuracy and introduces additional metrics such as Net Reversal Efficiency and correctness margin to analyze whether the argumentation layer is actually helping resolve disagreements rather than just changing answers arbitrarily.

The evaluation is also reasonably broad in task type. The benchmark suite spans general reasoning, medical QA, truthfulness, graduate-level science QA, and long-context reasoning, which supports the claim that the framework is intended as a general orchestration approach rather than something narrowly tuned to one dataset. The cybersecurity use case is also a helpful illustration of the type of explanation the authors want the system to provide.

My main concern is that the causal framing feels overstated. In the paper, the SCM is induced by recasting the authors’ own deterministic QBAF update equations, and the interventions are edge deletions inside that induced graph. This supports a notion of counterfactual sensitivity within the formal aggregation mechanism, but I am not convinced it justifies stronger causal claims about the model’s actual internal reasoning process. The paper itself effectively acknowledges this boundary when it says these explanations are internal to the QBAF-induced SCM rather than external reality. As a result, I found the term “causally grounded” stronger than what is actually established.

A second concern is that the baseline setup is not sufficiently diagnostic. The main evaluation uses GPT-4o-mini throughout and compares ARGORA against “Direct 1×,” “Direct MV3,” “ARGORA-like (modified CoT) 1×,” and “ARGORA-like (modified CoT) MV3.” However, these labels are not very transparent, and the paper does not clearly report the standalone performance of each actual expert configuration used inside ARGORA. Because ARGORA uses three expert instances with orchestrated expert prompting, I think the paper should report the individual performance of each expert role, or at least the average and best individual expert, to separate gains from orchestration and graph aggregation versus gains from prompt specialization. As written, it is hard to tell how much of the improvement comes from the framework itself.

Relatedly, although I understand the logic of holding the backbone fixed to isolate orchestration effects, the empirical case would be stronger if the main benchmark suite were tested across multiple stronger model families, rather than being centered on GPT-4o-mini in the main results table. The paper does include a cross-model cybersecurity case study, but that is a single qualitative example rather than a systematic comparison on the main quantitative benchmarks.

I also found the empirical evidence somewhat mixed. The paper reports gains on several tasks, but not all results are strong or cleanly convincing. MedQA is essentially competitive rather than clearly better, and some MuSR settings are weak or inconsistent. In addition, the “Best” semantics column is explicitly described as an oracle upper bound because it uses ground-truth labels for selection, so those numbers should not be treated as deployable evidence in favor of the method. This makes the practical advantage of the full framework feel less decisive than the presentation sometimes suggests.

Finally, some important parts of the method seem only partially validated. The override mechanism is restricted to single-edge interventions for computational reasons, which makes it more of a limited correction heuristic than a broadly convincing intervention framework. More generally, the paper feels strongest as an interpretability-oriented orchestration paper, but weaker as a demonstration of robust empirical superiority over simpler baselines.

---

> ### Author Rebuttal · Authors · 2026-03-28
>
> We thank the reviewer for taking the time to read and give a thorough assessment of our work. We answer each of the key questions as follows:
>
> ---
>
> ### **Key Question 1**
>
> We acknowledge that the baseline naming could be clarified. To be specific, "Direct 1×" uses a *just-directly-ask* prompting approach (e.g., *“Given this question, answer with the appropriate answer format.”*) without explicit prompt engineering techniques.
>
> "ARGORA-like (modified CoT) 1×" uses the exact same task prompt template that is provided to ARGORA's expert instances during discussion. This prompt includes CoT-like step-by-step reasoning instructions (e.g., *"REASONING: [step-by-step reasoning] / FINAL ANSWER: [appropriate answer format]"*) and is identical in wording to what each ARGORA expert receives. We called it "modified CoT" since it is not the verbatim prompt format used in the original CoT paper, but modified slightly for compatibility with ARGORA.
>
> So as the reviewer has mentioned, this is the closest proxy that we can give for the actual expert prompts, and separates prompt-level gains from framework-level gains.
>
> ---
>
> ### **Key Question 2**
>
> We have conducted the requested ablation and report standalone accuracy for the individual expert configuration used inside ARGORA for the same experimental result reported in our paper:
>
> | Dataset                        | ARGORA | Best Exp. | Avg Exp. | Gain (Best) | Gain (Avg) |
> |--------------------------------|--------|-----------|----------|-------------|------------|
> | MMLU-Pro                       | 0.638  | 0.606     | 0.593    | +5.3%       | +7.6%      |
> | MedQA                          | 0.812  | 0.804     | 0.793    | +1.0%       | +2.4%      |
> | TruthfulQA                     | 0.882  | 0.870     | 0.854    | +1.4%       | +3.3%      |
> | GPQA Diamond                   | 0.450  | 0.434     | 0.415    | +3.7%       | +8.4%      |
> | MuSR (Murder Mystery)          | 0.664  | 0.644     | 0.637    | +3.1%       | +4.2%      |
> | MuSR (Team Alloc.) [GPT-5-mini]| 0.780  | 0.744     | 0.731    | +4.8%       | +6.7%      |
>
> Our results show that ARGORA outperforms even the best standalone expert. For MuSR Team Allocation, we additionally incorporate our GPT-5-mini results from the appendix (5 experts) to demonstrate that this pattern holds across different model families and expert counts. We will incorporate this ablation in our revision.
>
> ---
>
> ### **Key Question 3**
>
> The reviewer's understanding is correct, and we elaborate on this point in greater detail in our response to Reviewer cVAc (Weakness 1), which we invite the reviewer to consult. That being said, we would be more than happy to clarify our causal claim in our revision.
>
> ---
>
> ### **Key Question 4**
>
> We appreciate this concern and want to clarify that our main evaluation is also done using the more capable GPT-5-mini model as detailed in the Appendix section. We elaborate on this point in our response to Reviewer YxZ9 (Weakness 1), which the reviewer may feel free to consult. ARGORA shows consistent gains across both model families, and in several cases the stronger backbone amplifies ARGORA's benefits, and the results are generally consistent with the gains observed with GPT-4o-mini in the main paper.
>
> We also want to address that we would have preferred to include evaluations with even stronger models such as the full-scale GPT variants rather than the mini versions; but given the scale of our benchmark suites, the API costs for higher-capability models become nonnegligible.
>
> ---
>
> ### **Key Question 5**
>
> We believe single-edge interventions are sufficient in practice for ARGORA's graph structure.
>
> ARGORA's discussion graphs are structurally designed to be bounded-depth rooted trees. This graph topology allows us to use single-edge interventions for a wide range of counterfactual scenarios: removing the influence of individual supporting or attacking claims or entire discussion subchains, and blocking targeted rebuttals against a particular argument. The expressiveness of single-edge interventions is therefore a direct consequence of the topology of the QBAF-SCM graph constructed with ARGORA.
>
> Formally, as discussed in Appendix E (Remark E.10 on page 35), single-edge interventions suffice for any query of the form *"was argument a (and its edge e) necessary for outcome o?"* in our underlying SCM. The three counterfactual query families we define address the primary explanatory questions for ARGORA's decision process and do not require more complex interventions to be informative.
>
> That said, we agree that an extended family of interventions like simultaneous multi-edge deletions are a natural extension and could be valuable for deeper or more complex graph topologies. We believe this could be an interesting approach for future work.

---

> > ### Author Rebuttal · Reviewer_ipPa · 2026-04-03
> >
> > Please update the papers with the changes.

---

> > > ### Author Response · Authors · 2026-04-03
> > >
> > > We thank the reviewer for the acknowledgement. We will update the paper to ensure that the additional evaluations and clarifications discussed during the review process are clearly reflected.

---

### Official Review · Reviewer_cVAc · 2026-03-13

**Soundness:** 2
**Presentation:** 3
**Significance:** 2
**Originality:** 2
**Overall Recommendation:** 3
**Confidence:** 3

**Summary:**

The paper introduces ARGORA, a framework that organizes discussions among multiple expert LLM agents into structured argumentation graphs and evaluates them through causal analysis to determine which arguments influence the final decision. The study addresses a central question: how multi-agent LLM reasoning can become more interpretable and reliable than simple aggregation methods. In particular, the manuscript explores how combining structured argumentation, counterfactual diagnostics, and an external override mechanism can improve both decision quality and transparency in multi-LLM reasoning systems across several benchmarks.

**Compliance With Llm Reviewing Policy:**

Affirmed.

**Final Justification:**

During rebuttal, the authors address some of the concerns; however, they do not resolve the central issue that the LLM-derived graph may fail to capture meaningful causal dependencies. I would keep the current score leaning toward rejection.

**Key Questions For Authors:**

The proposed method involves multiple agents and several rounds of reasoning and revision. Could the authors provide more details on how the computational cost (e.g., token usage) compares to the baselines?

**Limitations:**

Yes

**Strengths And Weaknesses:**

**Strengths**
1. The paper addresses an important problem in current multi-agent LLM systems.
2. The writing is clear and well-organized, with sufficient details that make the paper easy to follow.

**Weaknesses**
1. Although the paper interprets QBAF evaluation as a deterministic structural causal model (SCM), this does not necessarily imply that the resulting graph corresponds to a genuine causal graph. In standard causal modeling, SCMs are typically used to represent mechanisms underlying a data generating process. In contrast, the SCM here appears to be largely a reformulation of the deterministic aggregation rule used to compute argument strengths, rather than a model of the underlying data generating process of the task. Moreover, the parent–child relations in the argument graph are derived from LLM-generated arguments, which may not faithfully reflect the actual dependencies relevant to the problem. As a result, the causal interpretation may be better understood as describing the internal reasoning aggregation mechanism of the system rather than true causal structure.
2. The empirical evaluation relies primarily on relatively simple baselines, including single-model prompting and majority voting, as shown in Table 1. While these provide a useful reference, the comparison does not include several stronger or more closely related multi-agent reasoning frameworks discussed in the related work (e.g., multi-agent debate or mixture-of-agents style systems). As a result, it is difficult to isolate whether the improvements come specifically from the proposed argumentation graph and causal analysis components, or simply from multi-expert sampling and aggregation. In addition, the gains over the majority vote baseline are relatively modest on several datasets, and in some cases the baseline performs comparably or even better. A broader set of competitive baselines or stronger compute-matched reasoning methods would help clarify the empirical contribution of the proposed framework.

---

> ### Author Rebuttal · Authors · 2026-03-28
>
> We thank the reviewer for taking the time to read and give a thorough assessment of our work.
>
> ---
>
> ### **Weakness 1**
>
> We agree that the SCM does not model the data-generating process of the underlying task, nor the internal reasoning of the LLMs. Our claim can be stated more precisely as follows: *by casting the deterministic QBAF evaluation procedure as an SCM, ARGORA supports a formally well-defined family of counterfactual attribution queries within the system’s decision mechanism itself*. This is the level at which our causal interpretation is intended.
>
> Within that scope, the induced SCM is formally justified (Prop. 4.1; Lemma C.12) and enables analyses that are unavailable in prior multi-agent LLM frameworks (Def. 4.4 to Def. 4.6). We also believe this is practically meaningful as shown in our cybersecurity use case, where counterfactual explanations help localize decision-relevant factors behind the final verdict and provide useful diagnostic value in high-stakes settings. To avoid misinterpretations of our causal claim, we are happy to revise the wording to make this scope explicit.
>
> ---
>
> ### **Weakness 2**
>
> We respectfully note that our baseline design was intentional: the majority-vote baseline matches ARGORA's expert count in terms of independent answer samples, specifically to test whether simply generating multiple answers and voting is sufficient to explain the gains, and we show that ARGORA's improvements are attributable to its specific components rather than generic multi-expert sampling.
>
> First, our ablation studies in the appendix directly address this concern. Appendix H.2 (Table 13) shows that replacing orchestrator-initialized QBAF base scores with uniform initialization consistently degrades both accuracy and NRE across benchmarks. This confirms that the QBAF scoring mechanism is a critical driver of performance.
>
> Second, as detailed in our response to Reviewer ipPa (Key Question 2), we have conducted a new ablation reporting the standalone accuracy of each individual expert configuration. ARGORA outperforms the best-performing individual expert on every benchmark, which demonstrates that the gains stem from structured aggregation rather than prompt specialization or sampling effects.
>
> We also conducted a comparison of ARGORA against ArgLLM (AAAI '25), arguably the closest prior work that combines argumentation frameworks with LLMs. We evaluated on their specific claim-verification dataset variants (MedClaim / TruthClaim), and find that ARGORA achieves noticeable improvements over ArgLLM's reported results:
>
> | Dataset   | ArgLLM | ARGORA |
> |----------|------------------|--------|
> | MedClaim | 0.71             | 0.84   |
> | TruthClaim | 0.78           | 0.85   |
>
> That said, we acknowledge that a more comprehensive comparison against other multi-agent LLM systems would further strengthen the evaluation. As discussed in Section 5.3 and Table 3, these systems differ substantially in the intermediate representations and outputs which makes strictly fair end-to-end comparisons difficult without re-implementing each method under identical settings. We will try our best to incorporate additional comparative results in the final revision.
>
> ---
>
> ### **Key Question**
>
> Runtime per datapoint is already reported in our contextual orthogonality pruning ablation (Appendix H.1, Figure 8), but we additionally report the following per-instance token and time costs averaged across our benchmark evaluations:
>
> | Metric                  | Single LLM | ARGORA (3 exp) | Ratio |
> |------------------------|----------------|--------------------|-------|
> | Output tokens (avg)    | ~650           | ~8,000             | ~12×  |
> | Wall-clock time (avg)  | ~14s           | ~130s              | ~9×   |
>
> The wall-clock time ratio (9×) being lower than the token ratio (12×) is a result of optimization at implementation level, which allows ARGORA expert discussions to run in parallel. We also report the breakdown of inference calls per component in a single ARGORA discussion instance as follows:
>
> | Role          | Inference calls per instance |
> |---------------|-----------------------------|
> | Orchestrator  | 17                          |
> | Per expert    | 6 ~ 9                       |
>
> We note that multi-expert LLM frameworks are inherently more expensive in token usage than single-model baselines, and this is a fundamental characteristic shared by all multi-agent systems, not specific to ARGORA. We believe this cost is justified by the capabilities ARGORA provides beyond raw accuracy such as mechanistic explainability, which are not available from single-model or majority-vote baselines. This is also why we emphasize that ARGORA is particularly well-suited for high-stakes, detail-sensitive applications such as the cybersecurity use case (Section 6), where the interpretability and diagnostic value of the argumentation structure outweigh the additional inference cost.

---

> > ### Author Rebuttal · Reviewer_cVAc · 2026-04-03
> >
> > While the clarification narrows the scope to decision-level counterfactuals, it does not address the central concern that the LLM-derived graph may fail to capture meaningful causal dependencies. I would keep the current score.

---

> > > ### Author Response · Authors · 2026-04-03
> > >
> > > We thank the reviewer for the acknowledgement. That being said, we would like to clarify our position once more by addressing the reviewer's central concern that ARGORA fails to capture meaningful causal dependencies. We wish to emphasize that **capturing task-level causal dependencies is not a goal of this work, nor does the paper claim otherwise**.
> > >
> > > We believe the concern arises because ARGORA casts its argumentation graphs as SCMs, whereas SCMs are often associated with mechanisms underlying a data-generating process.
> > >
> > > However, in the literature, causal-intervention formalisms are also used to analyze internal computational mechanisms of learned systems, rather than only external data-generating processes. For example, Geiger et al. (JMLR 2025) [1] explicitly position causal abstraction as a theoretical foundation for mechanistic interpretability. Likewise, Meng et al. (NeurIPS 2022) [2] use causal interventions to identify internal activations that are decisive for factual recall in GPT, and Wu et al. (NeurIPS 2023) [3] apply a causal-abstraction-based method to identify an interpretable causal mechanism underlying an LLM's behavior on a reasoning task. Our use of an induced deterministic SCM is intended in this narrower sense: not as a claim about external ground-truth causality, but **as a formalization of intervention-defined effects within ARGORA’s explicit decision procedure**.
> > >
> > > Finally, whether the induced argumentation graph captures structure that is useful for ARGORA’s purpose is not left as an assumption. It is evaluated empirically through the positive NRE on multiple benchmarks, the positive correctness margins, and the decision-relevant counterfactual explanations in the cybersecurity use case (Sections 5–6 and Appendix J). The results support the narrower claim that the induced structure is decision-relevant and diagnostically informative for the system defined by ARGORA.
> > >
> > > We hope this clarifies the scope and formal grounding of our causal claims, and addresses the reviewer's remaining concern.
> > >
> > > **References**
> > >
> > > [1] Geiger, A., et al. (JMLR 2025). *Causal abstraction: A theoretical foundation for mechanistic interpretability*.
> > >
> > > [2] Meng, K., et al. (NeurIPS 2022). *Locating and editing factual associations in GPT*.
> > >
> > > [3] Wu, Z., et al. (NeurIPS 2023). *Interpretability at scale: Identifying causal mechanisms in Alpaca*.

---

### Official Review · Reviewer_YxZ9 · 2026-03-18

**Soundness:** 3
**Presentation:** 4
**Significance:** 3
**Originality:** 3
**Overall Recommendation:** 4
**Confidence:** 3

**Summary:**

The paper argues that existing multi-expert LLM systems gather diverse perspectives but combine them through simple aggregation, which obscures which arguments drove the final decision. The authors propose ARGORA to address this by organizing multi-expert discussions into explicit argumentation graphs (Quantitative Bipolar Argumentation Frameworks) that encode support and attack relations among arguments.

The framework further treats these argument graphs as causal models, enabling counterfactual analysis by removing individual arguments or edges and recomputing outcomes. This allows ARGORA to identify which arguments are necessary for a given decision and to assess how outcomes change under targeted modifications, providing a form of causal attribution over reasoning chains.

Overall, I think ARGORA is a technically solid and conceptually interesting paper. However, the authors could improve the empirical section of the paper. Especially, showing how their framework works for different models and providing a solid ablation study beyond hyperparameters, for example, what is the effect of the override policy?

**Compliance With Llm Reviewing Policy:**

Affirmed.

**Key Questions For Authors:**

Please check the weaknesses

**Limitations:**

yes

**Strengths And Weaknesses:**

Strengths:

1. ARGORA produces an inspectable argument graph whose final score is computed by a deterministic formula.
Treating the argument graph as a causal model so you can actually ask 'would the answer have changed if this argument weren't there?' is something I think is novel.

2. The evaluation is also more comprehensive than typical accuracy-centric benchmarks. In addition to task performance, the authors consider reasoning efficacy, argumentative utility, and structural stability, offering a more holistic view of system behavior. The inclusion of failure cases, particularly on MuSR subtasks, further strengthens the empirical analysis.

Weakness:
1. All experiments use GPT-4o-mini exclusively, making it hard to distinguish framework-level failures from model-level limitations.

2. The system has at least five interacting components (expert elicitation, QBAF construction, orthogonality pruning, SCM counterfactuals, override policy), yet none are ablated in isolation. It is difficult to determine which components drive the reported gains.

---

> ### Author Rebuttal · Authors · 2026-03-28
>
> We thank the reviewer for taking the time to read and give a thorough assessment of our work. The main concerns in this review seem to focus on the experimental side of things, like evaluations for different models and the missing ablation for each component of the ARGORA framework in the main sections. However, we do note that all of the concerns stated above have been addressed in the Appendix section of our paper. Because our work on ARGORA contains many components and evaluations, we unfortunately had to move a substantial amount of these sections from the main paper to the appendix due to page limit constraints. Specifically, we address the two weaknesses brought up by the reviewer as follows.
>
> ---
>
> ### **Weakness 1**
>
> Our evaluation actually spans three distinct model families:
>
> - (1) GPT-4o-mini — the model used to evaluate on the benchmark suite in the main paper (Table 1, 7 benchmarks)
> - (2) GPT-5-mini — full quantitative evaluation on 5 of the same benchmarks from the benchmark suite (Appendix G.4, Tables 8–9), this time using 5 experts
> - (3) Qwen3-14B (an open source model, 14B parameters) — utilized by all 3 experts in the cybersecurity use case (Section 6) in the main paper
>
> We also note some interesting key findings with the results on GPT-5-mini: on parts of the MuSR dataset, we observe that the net reversal efficiency (NRE) jumps from +0.014 (p=0.500) with GPT-4o-mini to +0.169 (p=0.001) with GPT-5-mini, demonstrating that ARGORA’s argumentative utility scales with base model capability (this also further addresses the failure case mentioned by the reviewer in the strengths section of our paper). We hope that these evaluations help answer the concerns about framework-level failures and model-level limitations as mentioned by the reviewer.
>
> ---
>
> ### **Weakness 2**
>
> We appreciate this concern, and we want to clarify that component-level ablations for the key accuracy-affecting components are also present in our paper (again, primarily in the Appendix due to page constraints). Specifically, we ablate the following:
>
> - **QBAF construction**
>   Appendix H.2 (Table 13) directly compares orchestrator-initialized base scores against a neutral baseline in order to isolate the effect of task-aware scoring on the final winner. We find that orchestrator initialization yields consistently higher accuracy across all seven benchmarks and maintains positive NRE on most datasets, whereas neutral initialization frequently produces negative NRE, indicating that the LLM-based QBAF scoring is a necessary component for stable corrective behavior rather than an optional enhancement.
>
> - **Orthogonality pruning**
>   Appendix H.1 (Figure 8) conducts a full sweep of the pruning threshold \rho_sim from 0.1 to 0.9 on GPQA Diamond, measuring both accuracy and runtime. This ablation shows that overly aggressive pruning removes informative arguments and slightly degrades accuracy, while overly permissive pruning causes context window saturation and increases runtime.
>
> - **Override policy**
>   Appendix G.6 (Tables 10–12) provides an extensive ablation of the override mechanism, sweeping across four distinct cost formulations and tradeoff weight (λ) value variations. We also ablate the winner-confidence gate by comparing gated versus ungated override policies. The results demonstrate that the gating is the dominant safety factor, and that gated overrides achieve non-negative net gain in 88.2% of configurations versus only 57.5% for ungated, and the gate correctly blocks all harmful override attempts on datasets like MuSR Team Allocation where internal consensus is more reliable than the external signal.
>
> We acknowledge that these results should be made more prominent, and we will revise the main paper to better convey these ablations for the reader.

---

> > ### Author Rebuttal · Reviewer_YxZ9 · 2026-04-03
> >
> > I thank the authors for the detailed response. The additional model evaluations and component-level ablations address my concerns, and the scaling result on MuSR with GPT-5-mini is a meaningful finding.
> >
> >
> > I strongly encourage the authors to follow through on their commitment to make these ablations more prominent in the main paper, and not bury critical validation experiments in the appendix.
> >
> > I will keep my score.

---

> > > ### Author Response · Authors · 2026-04-03
> > >
> > > We are glad that our response helped address the main concerns. As for the evaluations in the appendix, we agree that these validation results deserve greater prominence in the main paper rather than the appendix, and we regret that page constraints limited their visibility in the submitted version.
> > >
> > > We will make full use of the additional page allowed in the revised version to bring these experiments to the main paper.

---

### Decision · Program_Chairs · 2026-04-30

**Decision:**

Reject

**Comment:**

This paper argues that existing multi-expert LLM systems gather diverse perspectives but combine them through simple aggregation, which obscures which arguments drove the final decision. it proposes ARGORA to address this by organizing multi-expert discussions into explicit argumentation graphs (Quantitative Bipolar Argumentation Frameworks) that encode support and attack relations among arguments. It further treats these argument graphs as causal models, enabling counterfactual analysis by removing individual arguments or edges and recomputing outcomes. This allows ARGORA to identify which arguments are necessary for a given decision and to assess how outcomes change under targeted modifications, providing a form of causal attribution over reasoning chains.